# Future land-use pattern projections and their differences within the ISIMIP3b framework

Edna Johanna Molina Bacca[1,2], Miodrag Stevanović[1], Benjamin Leon Bodirsky[1], Jonathan C. Doelman[3,4], Louise P. Chini[5], Jan Volkholz[1], Katja Frieler[1], Christopher P.O. Reyer[1], George Hurtt[5], Florian Humpenöder[1], Kristine Karstens[1,2], Jens Heinke[1], Christoph Müller[1], Jan Philipp Dietrich[1], Hermann Lotze-Campen[1,2], Elke Stehfest[3], and Alexander Popp[1,6]

[1]Potsdam Institute for Climate Impact Research, Member of the Leibniz Association, Potsdam, Germany
[2]Department of Agricultural Economics, Humboldt-Universität zu Berlin, Berlin, Germany
[3]PBL Netherlands Environmental Assessment Agency, The Hague, The Netherlands
[4]Copernicus Institute of Sustainable Development, Utrecht University, Utrecht, The Netherlands
[5]Department of Geographical Sciences, University of Maryland, College Park, MD, USA
[6]Faculty of Organic Agricultural Sciences, University of Kassel, Witzenhausen, Germany.

**Correspondence:** Edna Johanna Molina Bacca (mbacca@pik-potsdam.de)

**Abstract.**

Land use is a key human driver affecting Earth's biogeochemical cycles, hydrology, and biodiversity. Therefore, projecting future land use is crucial for global change impact analyses. This study compares harmonized land-use and management trends, analyzing uncertainties through a three-factor variance analysis involving socioeconomic-climate scenarios, land-use models, and climate models. The projected patterns are used as human-forcing inputs for the Intersectoral Impact Model Intercomparison Project phase 3b (ISIMIP3b) and multiple impact modeling teams. We employ two models (IMAGE and MAgPIE) to project future land use and management under three socioeconomic-climate scenarios (SSP1-RCP2.6, SSP3-RCP7.0, and SSP5-RCP8.5), driven by impact data like yields, water demand, and carbon stocks from updated climate projections of five global models, considering $CO_2$ fertilization effects. On the global level, there is strong agreement among land-use models on land-use trends in the SSP1-RCP2.6 scenario (low adaptation and mitigation challenges). However, significant differences exist in management-related variables, such as the area allocated for second-generation bioenergy crops. Uncertainty in land-use variables increases with higher spatial resolution, particularly concerning the locations where cropland and grassland shrinkage could occur under this scenario. In SSP5-RCP8.5 and SSP3-RCP7.0, differences among land-use models in global and regional trends are primarily associated with grassland area demand. Concerning the variance analysis, the selection of climate models minimally affects the variance in projections at different scales. However, the influence of the socioeconomic-climate scenarios, the land-use model, and interactions among the underlying factors on projected uncertainty varies for the different land-use and management variables. Our results highlight the need for more intercomparison exercises focusing on future spatially explicit projections to enhance understanding of the intricate interplay between human activities, climate, socioeconomic dynamics, land responses, and their associated uncertainties on the high-resolution level as models evolve. It also underscores the importance of region-specific strategies to balance agricultural productivity, environmental conservation, and sustainable resource use, emphasizing adaptive capacity building, improved land-use management, and targeted conservation efforts.

# 1 Introduction

Land-use and land-use change substantially and directly impact the Earth's biogeophysical and biogeochemical processes and systems (Luyssaert et al., 2014). Among others, land-use changes perturb the interactions between the terrestrial biosphere and the atmosphere, including the hydrological and carbon cycles and other processes (Foley et al., 2005). For example, land-use change, which could have affected up to 32% of the world's land between 1960 and 2019 (Winkler et al., 2021), has caused net changes in $CO_2$, $CH_4$, and $NO_x$ fluxes (Kim and Kirschbaum, 2015). These disturbances on biogeochemical and biogeophysical processes can lead, in turn, to local and global alterations of surface water and groundwater levels, soil quality, species richness and evenness (biodiversity), other ecosystem services, the spread of diseases and pests, and weather and climate (Roy et al., 2022; Lambin et al., 2001; Oliver and Morecroft, 2014).

Recently, land cover changes have been driven predominantly by human land-use activities, particularly by managing and expanding agricultural land (cropland and pastures) into forests and other natural vegetation (Lambin and Meyfroidt, 2011). This trend has been linked, on global and local scales, to various factors such as shifts in population (affecting food demand), changes in dietary patterns due to growing incomes, advancements in agricultural yields (technological and intensification changes), growing demand for bioenergy in recent decades (Alexander et al., 2015), and climate change (Mendelsohn and Dinar, 2009). The evolution of these factors in the future has been explored using the Shared Socioeconomic pathways (SSPs) (Popp et al., 2017), which indicate that projections based on inequality (with highly unproductive agricultural land in low-income countries), rapid population growth, or high demand for agricultural commodities may lead to further agricultural land expansion. Conversely, a more sustainable demand for agricultural products, achieved through dietary changes and a decline in population growth, could lead to decreased agricultural land use and support mitigation measures like afforestation and forest protection, allowing for the restoration of natural vegetation.

Future projections of land-use and agricultural management indicators are crucial for different impact assessments that take into account the effects of socioeconomic and climate change on the Earth system (e.g., greenhouse gas emissions (GHG) resulting from land-use changes) (Pongratz et al., 2018), water quality (e.g., issues stemming from fertilizer and nutrient leakage into lakes and rivers) (Schindler, 2006), energy demand (e.g., considerations related to urban development and associated heating/cooling demands) (Nazarian et al., 2022), to name a few. Thus, there have been different previous efforts in the land-use modeling community to define, harmonize, and evaluate climate and socioeconomic development scenarios and their impacts. For this purpose, various frameworks and models have been utilized to project and compare future land-use and food system-related variables focusing on crop and livestock production, food prices, use of resources, and changes in land-use areas, among others, under different scenarios (Sörgel et al., 2024; Weindl et al., 2024; Doelman et al., 2022; Rose et al., 2022; Lèclere et al., 2020; Hasegawa et al., 2018; Popp et al., 2017; Nelson et al., 2014; Popp et al., 2014b). At the same time, studies have evaluated different land-use model types, including partial and computable general equilibrium models, within specific scenarios to understand the main factors affecting land-use projections and food availability, the models' responses to those factors, and their associated uncertainties on global, regional and spatially explicit resolutions (Schmitz et al., 2014; Stehfest et al., 2019; Alexander et al., 2017; Prestele et al., 2016). Although these studies have pointed out and agreed that variance and spread of

results come from differences in inputs, variable definitions, parametrization, and sensitivity to change, no study has assessed the level of agreement and the role of variance using a set of harmonized high-resolution land-use and land-use management projections under different scenarios including $CO_2$ fertilization effects on yields.

This study compares the harmonized land-use and agricultural management patterns generated as climate-human forcing data by two land-use models (LUMs) for the ISIMIP framework phase 3b (more details about ISIMIP can be found in Appendix C1). We aim to inform about the differences in trends and the level of agreement among projections in different resolutions and to point out differences with previous estimations. Specifically, the comparison is made on three resolutions: on the global level, for five world regions, and at the grid level. Specifically, we compare the land-use and land-use change patterns generated by the Integrated Model to Assess the Global Environment (IMAGE) (Stehfest et al., 2014; Van Vuuren et al., 2021), and the Model of Agricultural Production and its Impact on the Environment (MAgPIE) (Dietrich et al., 2019) under assumptions for three different socioeconomic-climate scenarios and climate impact data generated using five Coupled Model Intercomparison Project Phase 6 (CMIP6)-biased corrected global climate models (GCMs). The global trends of the LUMs projections under the different scenarios were compared to the Land-Use Harmonization 2 (LUH2) dataset (Hurtt et al., 2017) of future land-use projections, which has commonly been used for impact analyses in global and regional studies (Yu et al., 2019; Qiu et al., 2023; Hoffmann et al., 2023). In addition to comparing the projections, we consider the variance of contributing factors to identify differences in the land-use model outputs and the locations where the variation among the projections is driven by factors different from the socioeconomic-climate scenarios assumptions, e.g., where differences among land-use model dynamics, the interaction among factors, or the uncertainty from the climate impact data play a more prominent role in explaining the variance. Our work differs from previous studies in the intercomparison of aggregated and high-resolution land-use data for a consistent set of scenarios; the consideration of climate impacts on biophysical constraints (crop yields, water availability and demand, and carbon densities) considering CMIP6 biased-corrected climate data and $CO_2$ fertilization effects; and that the output data was harmonized in the historical period of the time series (1995-2015). Besides cropland, grassland, forest, and other natural vegetation land types, our analysis focuses on second-generation bioenergy cropland areas, irrigated areas, and synthetic nitrogen fertilizer use to which we refer to as 'land-use management variables' in the text.

The paper is structured as follows: In section two, the methodology and the concepts used throughout the text are described and explained; section three includes the results, where regional trends of the LUMs are analyzed and compared to the LUH2 data set (section 3.1); grid-level projections and hotspots of uncertainty are assessed (section 3.2); and sources of variance in the different resolutions are identified (section 3.3). Finally, section four contains a discussion of the results and the conclusion.

## 2 Methods

### 2.1 Land-Use Models

This study used data from two land-use models that reported data sets for the ISIMIP 3b round. Although their approach and parametrization of biogeochemical, biogeophysical, and socioeconomic processes differ, both models represent the global land

system in detail through land-use modules capable of representing and allocating land types and management systems under different global change scenarios on the spatially explicit level.

The Integrated Model to Assess the Global Environment (IMAGE) framework (Stehfest et al., 2014; Van Vuuren et al., 2021) is developed by the Netherlands Environmental Assessment Agency (PBL) to understand changes in environmental conditions and sustainability issues driven by changing socioeconomic development, such as economic and population growth, over time. For this purpose, the IMAGE framework combines different submodels describing the energy system, agricultural and land-use sectors (26 world regions), and biophysical and biogeochemical conditions (grid level). The MAGNET Computable General

Equilibrium (CGE) model represents the agricultural economy, projecting, e.g., demand, production, and trade in agricultural commodities. The IMAGE-land model allocates crop, livestock, and timber production on the grid level based on regional information regarding food production and demand, animal feed, fodder, grassland, bioenergy, timber, and local climatic and geographic properties. Demand for bioenergy production aligns with climate policies and is determined by the energy system model TIMER. The TIMER energy model defines bioenergy demand based on land supply, biomass productivity, input costs,

and learning dynamics, which influence bioenergy prices. Climate policies in the IMAGE framework are designed to meet long-term climate targets by establishing global emission pathways. These pathways determine carbon tax prices and mitigation costs, which, in turn, affect bioenergy prices and demand (as detailed by Doelman et al. (2018)). An in-house version of the Lund-Potsdam-Jena managed Land (LPJmL) dynamic global vegetation model, used to calculate crop yields, soil characteristics, and other biophysical constraints, is dynamically coupled to IMAGE. Regarding disaggregation of land-use patterns to

the grid level, IMAGE relies on gridded potential yields from LPJmL, data from the simulation's previous time step, a regional management factor, and an empirical allocation algorithm. The process begins with calculating potential cropland and crop production data in the current time step using the patterns from the previous step. If production is insufficient to meet demand, less productive areas are abandoned, whereas cropland expansion employs the empirical algorithm that evaluates cropland and grassland allocation. More information is available in (Doelman et al., 2018).

The Model of Agricultural Production and its Impact on the Environment (MAgPIE) (Dietrich et al., 2019) (Version 4.4.0 for this study) is hosted at the Potsdam Institute for Climate Impact Research (PIK). MAgPIE is a recursive partial equilibrium optimization model of the agricultural and forestry sectors. It integrates demographic and economic development with agricultural commodities and timber production under different land-use management and land-based mitigation policies, aiming to minimize global production costs. As outputs, the model reports, among others, land-use patterns, technological

change needed to maintain production, GHG emissions, and total cost of agricultural production. The model uses PIK's hosted LPJmL-generated spatially explicit data of potential yields, carbon stocks, and blue water availability and demand for agriculture (Müller, 2024; Müller, 2024). For this application, MAgPIE uses exogenous inputs from the REMIND model, which is a multiregional energy-economy general equilibrium model that considers long-term macroeconomic growth. Specifically, REMIND provides information on GHG pricing and the demand for second-generation bioenergy crops (lignocellulosic feed-

stocks). REMIND determines this demand by considering the supply, trade, and conversion of biomass feedstocks through the value chain while accounting for the energy sector market conditions and regulatory frameworks in each socioeconomic growth scenario (as detailed by Merfort et al. (2023)). Since these scenarios are aligned with specific climate change path-

ways, bioenergy demand, for example, is intrinsically linked to the emissions budgets and carbon taxes required to achieve particular warming targets. Compared with ISIMIP2b (Frieler et al., 2017; Popp et al., 2014a), MAgPIE was run using a new forestry module Mishra et al. (2021) and a module for the accounting of "sticky" on-farm capital stocks, giving some inertia to the relocation of production and improving spatially explicit outputs. For more details regarding MAgPIE's 4.4.0 version and modules, refer to (Dietrich et al., 2021). In MAgPIE, land-use disaggregation is based on the patterns of the previous time step, available cropland, and a mapping between the high and low resolutions. At each time step, starting with cropland, changes in land use from the clusters are disaggregated using expansion and reduction weights and information about land availability and suitability. Detailed information can be found in the interpolateAvlCroplandWeighted function from the R library luscale developed by the MAgPIE team (Dietrich et al., 2024).

The spatially explicit analyses in this study were conducted at a 0.5°×0.5° resolution, although harmonized land-use projections are reported at 0.25°×0.25° since ISIMIP reports the set at the 0.5°×0.5° resolution. Considering that MAgPIE and IMAGE perform simulations using different regions, we selected five mega-regions to illustrate regional trends. Specifically, we used the so-called SSP regions, which have been widely applied in studies involving the Shared Socioeconomic Pathways (SSPs) and climate change, e.g., in Popp et al. (2017); Bauer et al. (2017); Meinshausen et al. (2020); Fu et al. (2021) (see Appendix Figure B1 for a map of the regions).

## 2.2 Scenarios

Following ISIMIP's 3b protocol (https://protocol.isimip.org/#ISIMIP3b/agriculture), the land-use patterns analyzed in this study represent three main socioeconomic-climate scenarios (also called only scenarios through the document): The first, SSP1-RCP2.6, corresponds to an increasingly sustainable world (SSP1) characterized, in the land-use context, by land regulation, a shrinking population after the second half of the century, an increase of productivity in developing economies, healthier diets (less animal products), less waste, and a globalized economy. It also assumes carbon prices for land-use emissions. SSP1 was matched to RCP 2.6, representing a mitigation pathway that limits global warming to +1.8°C (with a very like range of [+1.3°C,+2.4°C]) (Popp et al., 2017; IPCC, 2023) at the end of the century relative to 1850-1900. Secondly, the SSP3-RCP7.0 pathway describes a world with a growing population and regions focused on internal energy and food security issues, with hardly any cooperation due to regional rivalry. Land-use change is no further regulated compared with existing policies, the trade of agricultural commodities is reduced, livestock products dominate diets, and food waste is high. RCP7.0 represents a medium to high-end emissions pathway, with a warming increase of +3.6°C ([+2.8°C,+4.6°C]) (Popp et al., 2017; IPCC, 2023). The third, SSP5-RCP8.5, displays a globalized economy developed and driven by fossil fuels exploitation and international trade. Regarding land use, no additional protection policies are considered, and, as for SSP3, diets based on livestock products and high waste dominate. For RCP8.5, a high warming scenario, a +4.4°C ([+3.3°C,+5.8°C]) global mean surface temperature increase compared with pre-industrial levels is expected at the end of the century (Popp et al., 2017; IPCC, 2023). Specific details about how the narratives were incorporated into the different land-use models can be found in Table A1 of the Appendix.

Each simulation utilized biophysical data that captured the impacts of the different climate change pathways (RCPs) on cropland and pasture yields, water demand and availability, and carbon stocks—changes in carbon stock data applied to natural

vegetation and planted forests. The impact data was derived from internal (IMAGE) or external (MAgPIE) LPJmL computations and was generated using five GCMs: GFDL-ESM4 (Dunne et al., 2020), IPSL-CM6A-LR (Boucher et al., 2020), MPI-ESM1-2 (Müller et al., 2018), MRI-ESM2-0 (Yukimoto et al., 2019), and UKESM1-0-LL (Sellar et al., 2019) (See Figure

1 for a graphical depiction of the modeling workflow). These GCMs were selected based on the completeness of their data across all ISIMIP sectors, their performance during the historical period, and their representation of key processes, among other criteria (Lange, 2021).

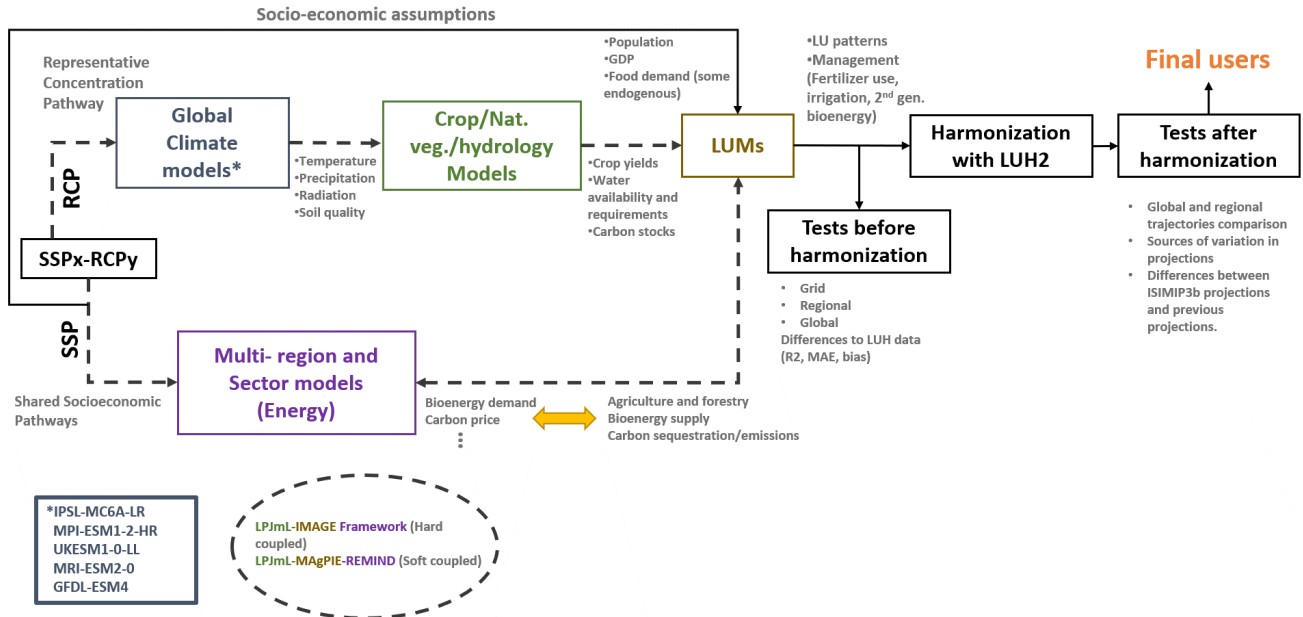

**Figure 1. Modeling protocol.** The flow diagram depicts the modeling workflow starting with the global climate models, which feed the crop/natural vegetation/hydrology models, which in turn generate the input data used by the land-use models (LUMs) together with the multi-region and sector models data used to build the assumptions and constraints of the different SSPx-RCPy scenarios. Black boxes represent processes (decision of scenarios and post-processing tests and steps), the purple represents the multi-region and multi-sector models, the gray the climate models, the green the crop/natural vegetation/hydrology models, and the light brown the land-use models. The dotted line represents the data transfer among models

Although the three main scenarios are the focus of this work, four counterfactuals were generated for the ISIMIP3b phase.

Three corresponded to projections based on the SSP trajectories without climate impacts (SSPx-NoAdapt). In these scenarios, socio-economic development trajectories were considered; however, biophysical constraints impacted by climate change (yields, water demand and availability, and soil and natural vegetation organic carbon) remained at 2015 values during the projections' horizon (2015-2100). The fourth counterfactual corresponded to a sensitivity experiment including SSP5-RCP8.5 forcing effects without $CO_2$ fertilization (SSP5-2015$CO_2$) based on impact data derived using the GFDL-ESM4 GCM. A brief

analysis of these scenarios can be found in Appendix D1.

## 2.3 Harmonization

A harmonization step was carried out in ISIMIP3b to facilitate a continuous transition between reconstructed gridded historical land use and the projected land-use and agricultural management patterns generated by the LUMs. The last historical year was 2015. This step also ensured a consistent format for the land-use data across all models.

The harmonization was done following the Land-Use Harmonization (LUH2) methodology (Hurtt et al., 2020) developed for the CMIP6 scenarios and used previously for ISIMIP2b (Frieler et al., 2017). This step was essential due to the variations observed in the definitions (e.g., criteria for distinguishing forest and other types of natural vegetation), resolutions, processes parameterization, and input sources among the different LUMs. Specifics of the harmonization can be found in Appendix C2.

## 2.4 Statistical analysis

### 2.4.1 Aggregation of raw data

For the present study, the harmonized data was then aggregated from $0.25°\times0.25°$ to the global, the five world regions, or $0.5°\times0.5°$ resolutions. The land-use types (cropland, grasslands, forest, other natural vegetation, and second-generation bioenergy crop areas), which were reported as fractional patterns (fraction of a grid cell) in the harmonized ensemble of projections, were multiplied by the size of each grid cell and then aggregated based on the respective mappings. For fertilizer use, reported in kg-per-hectare-per-crop type on the grid level, the value at the different resolutions was calculated by multiplying each grid-cell value by the fraction of the specific crop type and the grid-cell area. These values were then aggregated to the specific resolution using the respective mappings.

For the global and regional trend analyses, the average per SSP$x$-RCP$y$ and LUM was calculated using the simulations based on the five different GCMs.

### 2.4.2 Grid-level mean and coefficient of variation

To evaluate the resulting $0.5°\times0.5°$ projections by the different LUMs and their uncertainty, the mean and coefficient of variation (CV) were calculated per grid cell. For this purpose, the mean value, per scenario (SSP$x$-RCP$y$), of the land-use types and management variables was calculated for each grid cell, considering the simulations based on the two LUMs and the five different GCMs. The mean per grid cell was then based on ten simulations (2 LUMs $\times$ 5 GCMs) for each SSP$x$-RCP$y$ scenario.

Similarly, to evaluate the dispersion among the LUMs$\times$GCMs patterns per grid cell with a standardized measure, the coefficient of variation (CV, Equation 1) was calculated for each scenario using ten simulations (2 LUMs $\times$ 5 GCMs).

$$CV_j = \frac{\sigma_j}{\mu_j} \tag{1}$$

where the index $j$ represents a grid cell, $\sigma$ the standard deviation and $\mu$ the mean among the ten simulations. The CV was selected to ensure that grid cells with very different values of the analyzed variable were comparable.

Once the mean and the CV were calculated per grid cell, the cells were grouped per region, socioeconomic-climate scenario, and analyzed variables. The median and spread of the grouped cells for both indicators (mean and CV) were then analyzed and depicted in boxplots to identify regions where the different variables had larger or smaller values per grid cell, areas with large allocation of the variables, and uncertainty hot spots.

### 2.4.3 Variance analysis

Similar to previous studies, e.g., (Nishina et al., 2015; Hattermann et al., 2018; Vetter et al., 2015), to decompose the sources of variation in the variables of this study (land-use types, second-generation bioenergy cropland area, synthetic nitrogen fertilizer use, and irrigated cropland), a multi-factor variance analysis was performed at the global, regional, and grid scales.

This analysis aims to evaluate the level of agreement among LUMs informing about the primary sources of variation of the land-use and land-use-related projections of ISIMIP3b on different scales. In other words, this analysis is used to identify the locations and variables where variations can be explained by the differences among the scenarios' assumptions rather than by differences among land-use model dynamics, impact data, or their interactions. Given the level of detail reported by the land-use models, we focus on the factors corresponding to the categorical variables available. Our central assumption is that the primary sources of variance in the data stem from (1) distinctions among scenario trajectories, (2) differences in the processes, inputs, and modeling approaches of the various land-use models, and (3) uncertainties in the GCMs used to generate the impact data. Incorporating additional variables would require re-running the models and conducting further tests. However, as the primary aim of this study is to evaluate data presented rather than to, e.g., comprehensively analyze differences among land-use models, such tests fall outside the scope of this work. Schmitz et al. (2014) and Nelson et al. (2014) delve more deeply into differences among land-use models. Table A2 in the Appendix provides additional information on initial inputs and model processes relevant for calculating the land-use types and management variables.

Then, first, three factors were considered: the Land-use model (LUM) having two levels (IMAGE and MAGPIE); secondly, the Global Climate Model (GCM) with five levels (GFDL-ESM4, IPSL-CM6A-LR, MPI-ESM1-2, MRI-ESM2-0, and UKESM1-0-LL); and the Scenario with three levels (SSP1-RCP2.6, SSP3-RCP7.0, and SSP5-RCP8.5).

The total sum of squares, which represents the total variation of the set, can be denoted as the individual factors' sum of squares plus the sum of squares of the residual (Equation 2).

$$SS_{total,v,t} = SS_{LUM,v,t} + SS_{GCM,v,t} + SS_{Sce,v,t} + SS_{Int} \qquad (2)$$

where $SS$ indicates the Sum of Squares, and the indexes $total$ the overall sum of squares, $LUM$ the SS explained by the LUMs, $GCM$ by the GCM-based impact data, $Sce$ by the Scenario, and $Int$ the interactions among factors. Finally, the indexes $v$ denote the land-use variable and $t$ the time step under consideration. The fraction of the variation each factor explains was then calculated by dividing the individual factors' $SS$ by $SS_{total}$. On the grid scale, the variance analysis was performed on each cell.

The residual term—$SS_{Int}$ in Equation 2 —represents the portion of variance the independent variables (GCMs, RCPs, LUMs) cannot explain. This interpretation, where residuals are equivalent to the interactions, is particular to this type of study due to the deterministic nature of our data (the LUM models are deterministic). Since the total ($SS_{total,v,t}$) and factors' variance ($SS_{LUM,v,t} + SS_{GCM,v,t} + SS_{Sce,v,t}$) can be derived from the data, the factor that reflects the effect of the interactions $SS_{Int}$ can be calculated as the difference between the total and the factor's variance. This component captures the non-additive or nonlinear contributions to the variance.

Similarly, an additional variance analysis was performed, including the harmonization factor, to elucidate the locations where the effect of harmonization was strongest on the spatially explicit level. The unharmonized LUMs' outputs were used together with the harmonized. This means an additional factor ($Harm$) with two levels (harmonized and unharmonized) was added to Equation 2:

$$SS_{total,v,t} = SS_{LUM,v,t} + SS_{GCM,v,t} + SS_{Exp,v,t} + SS_{Harm,v,t} + SS_{Int} \tag{3}$$

We performed the variance analyses using the anova() function of the rstatix package of the R software (R Core Team, 2021).

## 2.5 Land-Use Harmonization 2 - CMIP6 dataset (LUH2)

To evaluate differences among the LUM's outputs for the ISIMIP 3b round with existing land-use and land-use management-related projections, we used the Land-Use Harmonization 2 (LUH2) data set developed by Hurtt et al. (2017) and used for CMIP6, which comprises the years from 2015 to 2100.

Using this data set offers multiple advantages, including the same format and similar historical trends to which the ISIMIP 3b-LUM's projections are harmonized, the same land-use and land-use management variables as the ones generated by the LUMs, and the three climate-human forcings evaluated in this study. The LUH2 projections include eight SSP*x*-RCP*y* combinations derived from five different Integrated Assessment Models (IAMs). Each SSP*x*-RCP*y* land-use projection reported is based on one IAM. Specifically, the SSP1-RCP2.6 LUH2 projection was based on the IMAGE 3.0 modeling framework; the SSP3-RCP7.0 on the Asia-Pacific Integrated assessment Model/Computable General Equilibrium mode (AIM/CGE) coupled with a land allocation model (Fujimori et al., 2012, 2014, 2017; Hasegawa et al., 2017); and the SSP5-RCP8.5 on the REMIND–MAgPIE integrated assessment modeling framework.

LUH2 data used for CMIP6 differs from the ISIMIP3b data in that it does not account for $CO_2$ fertilization. In crop models such as LPJmL, $CO_2$ fertilization has a positive effect (yield growth), leading, e.g., to lower required cropland areas to achieve the same production levels. Additionally, LUH2 used for CMIP6 combines outputs from multiple land-use models for different scenarios, introducing variability in dynamics based on the models used. In contrast, for ISIMIP3b, each land-use model simulated each SSP-RCP combination covered in this study. Another key difference lies in the inputs of the LUH2 harmonization algorithm, as the historical datasets used in ISIMIP3b have been updated compared to those in LUH2 for CMIP6. Additionally, a new representation of protected lands to better match the IAM assumptions was included, affecting patterns related to natural vegetation. There are also notable differences in the versions of the models employed. For MAgPIE, the

version used for CMIP6 simulations was 3.0, while ISIMIP3b utilized version 4.4.0. The latter (starting from MAgPIE 4.0) introduces several enhancements, most notably, a food demand model that accounts for detailed dietary composition, food waste, and demographic characteristics. MAgPIE's current version also improves spatially explicit outputs by incorporating the accounting of capital stocks and their depreciation and a more detailed representation of the forestry sector. Similarly, for IMAGE, the version used for ISIMIP3b was 3.3, whereas version 3.0 was used for LUH2. IMAGE 3.3 includes more crop categories, and advancements in bioenergy, deforestation, land-based mitigation, and water use modeling.

## 3 Results

### 3.1 Global and regional harmonized projections

#### 3.1.1 Land-use dynamics

On the global scale, harmonized land-use projections of the LUMs agree on the direction and rate of change for the different land-use types in SSP1-RCP2.6 (Figure 2a) over the modeling time horizon, with the largest land-use changes occurring in grasslands. However, although the LUMs agree with the direction of change in most of the land-use types for the different regions in 2050 in SSP1-RCP2.6, there are disagreements in cropland in Latin America (LAM) and other natural vegetation in the Middle East and Africa (MAF) (Figure 3a). In 2100, LUMs also agree with the direction of change for most land-use types, except for cropland in LAM (Figure 3b).

In SSP3-7.0 and SSP5-8.5, projections show different trends among LUMs and land-use types on the global and regional aggregation levels, most notably for grasslands. Specifically, there is a higher demand for grasslands in IMAGE compared with MAgPIE during the analysis time horizon (Figure 2a). In SSP3-RCP7.0, IMAGE's grasslands grow globally, mostly in LAM and MAF, compared with 2015 values, while for MAgPIE grasslands decrease, with most reductions occurring in the OECD countries, LAM, and the Asian countries excluding those that were part of the former USSR (ASIA).

Concerning cropland, for SSP1-RCP2.6, ISIMIP3b's projections for both LUMs display expansion until mid-century compared to 2015 and then a decrease. This decline in cropland is likely associated with a decrease in population and a change to more sustainable diets in SSP1-RCP2.6, which reduces the demand for agricultural commodities for food and feed (Popp et al., 2017). For SSP3-7.0 and SSP5-8.5, although both LUMs estimate that cropland expands, projections differ in terms of the size of the increase after 2050. Under SSP3-RCP7.0, MAgPIE projects larger cropland expansion than IMAGE. LUMs agree, however, that this expansion would occur mostly in MAF. In SSP5-8.5, cropland projections at the global scale almost overlap for both LUMs throughout the century. The LUMs also agree that MAF, ASIA, and LAM experience the highest growth in cropland and that the reforming economies that used to be part of the URSS (REF) undergo a slight decrease in 2050 and 2100 compared with 2015.

Regarding forest (primary and secondary) and other natural vegetation, an increase is expected in SSP1-RCP2.6 by the two models. In contrast, LUMs agree that forests and other natural vegetation areas steadily decline globally under SSP3-7.0, especially in LAM, MAF, and ASIA. However, there is a broad difference between LUM trends in SSP5-8.5's forest and other

natural vegetation projections on the global scale. While those land-use types stagnate after 2015 in MAgPIE, there is a large decline in forest and natural vegetation for IMAGE, mostly in MAF and LAM, related to competition for grasslands in this scenario in the affected regions.

Urban land projections between IMAGE and MAgPIE projections across different SSP$x$-RCP$y$ are virtually the same because this land type is an exogenous parameter in MAgPIE, derived from the last LUH2 data set, which is based on IMAGE's LUH2

projections for CMIP6 for urban land.

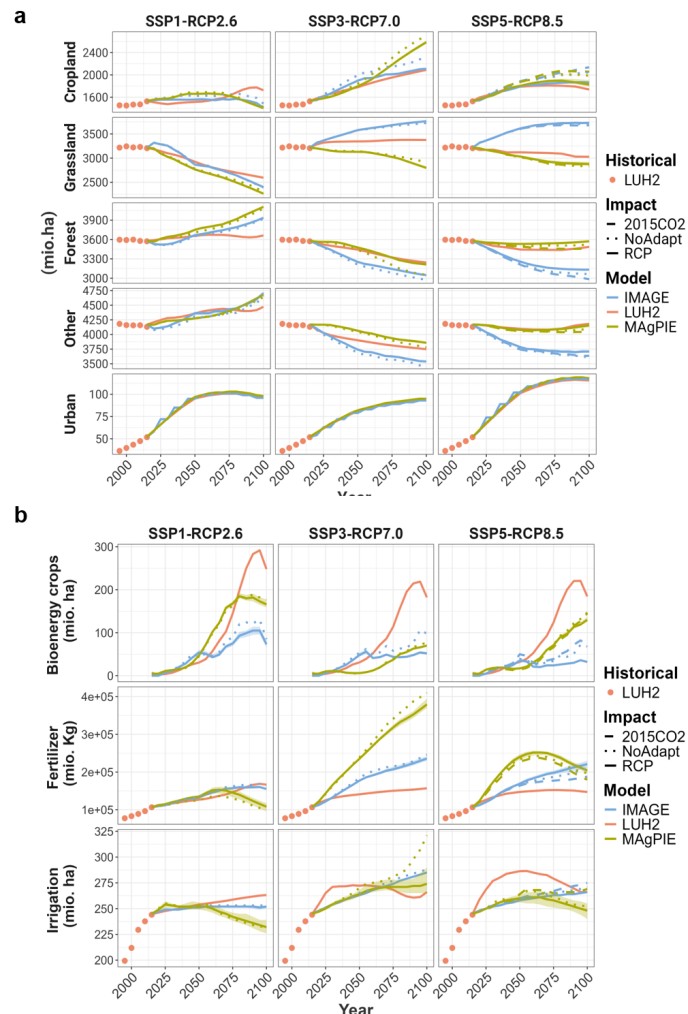

**Figure 2. Global harmonized data from two different land-use models (LUMs) for the ISIMIP (3b) round under three socioeconomic-climate scenarios (SSP1-RCP2.6, SSP3-RCP7.0, and SSP5-RCP8.5) and three impact types (RCP, NoAdapt and 2015CO$_2$ for SSP5).** a) shows the harmonized projections for five different land-use types areas (Cropland, Grassland, Forest, Other (Natural vegetation), and Urban areas in units of million hectares (mio. ha) and b) shows harmonized projections for two different management-related variables: synthetic nitrogen fertilizer use (Fertilizer) in million kilograms (mio. Kg); and irrigated cropland (Irrigation) in units of mio. ha. Additionally, it reports the area used for second-generation bioenergy crops (Bioenergy crops) in mio. ha. The lines in green and blue correspond to the average of the projections of each LUM based on impact data derived from five GCMs. The ribbon represents the upper and lower projections per LUM of the five GCMs-based impact data. The dashed line represents the counterfactual where no climate impact is considered (SSP$x$-NoAdapt), and the dotted line is the counterfactual where CO$_2$ fertilization is not included (SSP5-2015CO$_2$) in the yield projections used by the LUMs (only available for SSP5-RCP8.5). The orange line depicts LUH2 future projections for CMIP6 global climate model simulations. Finally, the circular orange dots are the LUH2 historical values to which the ISIMIP3b projections were harmonized

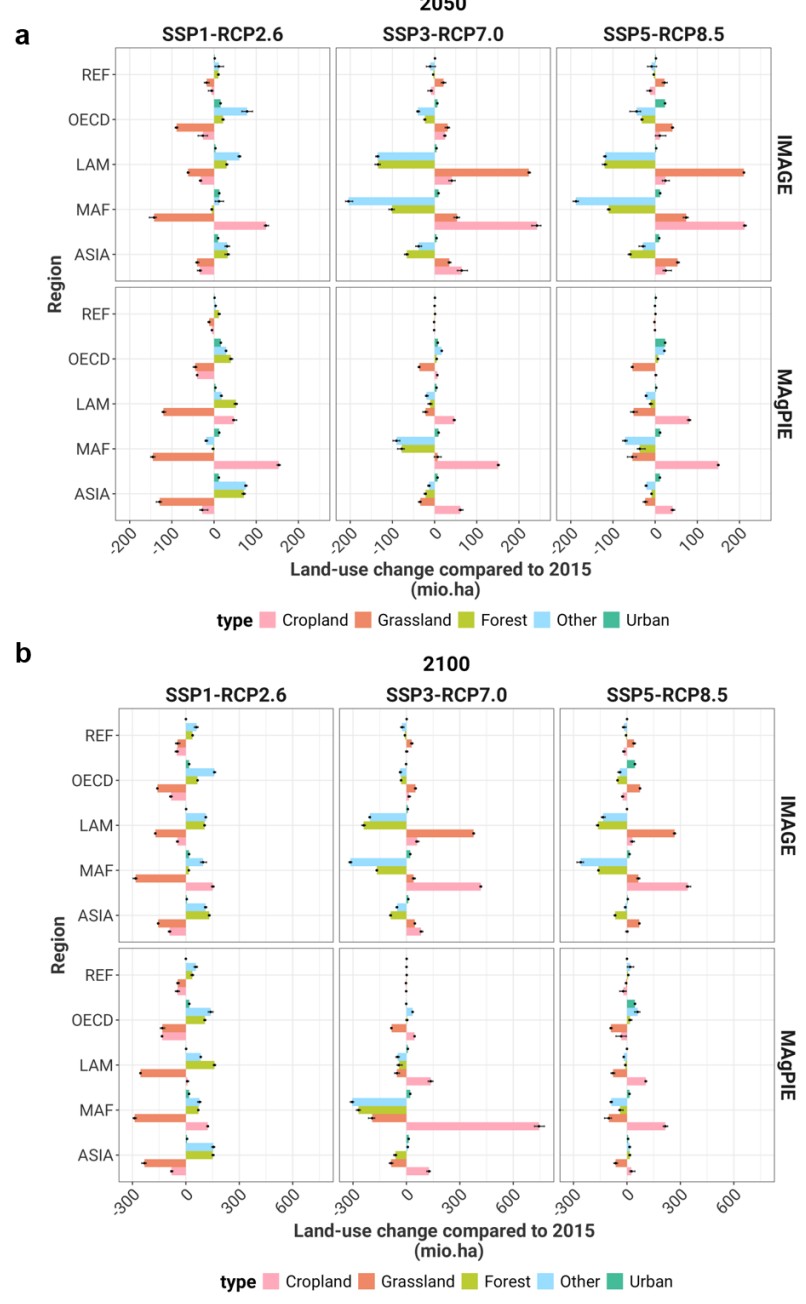

**Figure 3. Land-use change projections per region under the three socioeconomic-climate scenarios (SSP1-RCP2.6, SSP3-RCP7.0, and SSP5-8.5).** a) shows the difference in 2050 compared with 2015 of Cropland (pink), Grassland (orange), Forest (light green), Other natural vegetation (blue), and Urban area (dark green) in million of hectares (mio. ha), and b) the difference in the year 2100 compared

with 2015 based on harmonized LUMs future projections. Bars represent the average value of LUM projections under impacts based on five GCMs, and the extremes of the error bars are the minimum and maximum values of the LUM-specific ensemble

### 3.1.2 Land-use management variables

Regarding land-use management variables (second-generation bioenergy crop areas, nitrogen fertilizer use, and irrigated land),
LUMs generally agree on the trends in SSP1-RCP2.6. However, although both LUMs project a peak of crop area destined for second-generation bioenergy crops around 2090 (Figure 2b) for SSP1-RCP2.6, the rate of increase is different for the LUMs, starting in 2050, and leads to the largest difference in the year 2075. In MAgPIE projections, the increase of the second-generation bioenergy cropland area occurs primarily in the ASIA and REF regions, while in IMAGE projections, most of the bioenergy crops area is supplied by OECD countries and MAF (Figure B2 in the Appendix). These differences among
LUMs likely relate to the models' bioenergy crop yield proxies, regional demand for bioenergy, emissions reduction potential, allocation algorithms or trading patterns. In SSP3-RCP7.0, IMAGE projections display a bigger growth than MAGPIE for second-generation bioenergy crops until 2050, while MAgPIE estimates become larger than IMAGE's after 2075. On the regional scale, the LUMs agree that the ASIA region displays the largest area destined for second-generation bioenergy crops in 2100 under SSP3-RCP7.0. In SSP5-RCP8.5, global second-generation bioenergy cropland grows steadily for both LUMs.
However, after 2060, the growth rate becomes higher in the MAgPIE projections.

    Substantial differences emerge after 2065 for fertilizer in SSP1-RCP2.6, related to a reduction in cropland in MAgPIE in this period. In SSP1-RCP2.6, on the regional scale, IMAGE estimates higher fertilizer use than MAgPIE except for the OECD region throughout the century, and both LUMs agree that ASIA has the highest fertilizer application over the modeling time horizon. Both LUMs show increased synthetic nitrogen fertilizer use in the SSP3-7.0 scenario, with MAgPIE global fertilizer
use projections growing steeper than IMAGE's. Regionally, the distribution of the fertilizer use increase differs among the LUMs, but it is mostly concentrated in the MAF, OECD, and ASIA regions. In SSP5-RCP8.5, fertilizer application increases for both LUMs until 2065, with a higher growth rate for MAgPIE. However, after 2065, MAgPIE's fertilizer use projections decrease while IMAGE's steadily increase. Under SSP5-RCP8.5, the largest difference among estimations occurs in 2050.

    Similar to the fertilizer use patterns in SSP1-RCP2.6, MAgPIE projects higher reductions in projected irrigated areas, fol-
lowing the decrease in cropland in the second half of the century. In SSP3-RCP7.0 and SSP5-RCP8.5, irrigation global and regional trends among LUMs are similar to those in the low-emission scenario. In SSP3-RCP7.0, IMAGE's irrigated land is larger than MAgPIE projections, and in SSP5-RCP8.5, MAgPIE's global irrigated land projections decline slightly after 2070, opposite to IMAGE's behavior.

### 3.1.3 Differences between LUM's ISIMIP3b projections and LUH2-CMIP6

To assess how ISIMIP3b projections differ from existing estimates up to the generation of ISIMIP3b's land-use data, we compared aggregated global dynamics with the LUH2 data set of projections used for CMIP6 simulations (Hurtt et al., 2017).

    In SSP1-RCP2.6, ISIMIP3b harmonized projections show a larger reduction of grasslands globally than in the LUH2 data set, especially after 2050. Regarding cropland, opposite to ISIMIP3b projections, LUH2 projections decrease until 2050 compared

to 2015 and then increase. The different dynamics in cropland and grasslands lead to a larger increase in forest area than previously reported in LUH2, most notably after the second half of the century (Figure 2a). As for the global second-generation bioenergy cropland area under SSP1-RCP2.6, estimates are considerably lower in the ISIMIP3b projections than in LUH2. For example, in 2090, IMAGE's ISIMIP3b projections in SSP1-RCP2.6 are only a third of LUH2's. This drop in demand for second-generation bioenergy crops is related to changes in the mitigation assumptions of SSP1-RCP2.6, which involves updated impacts on yields. Fertilizer-use trends seem similar between LUH2 and IMAGE's ISIMIP3b projections. Finally, projected irrigated cropland areas start differing more strongly between LUMs and LUH2 after 2050, with MAgPIE projections being considerably higher than those of LUH2.

Compared to ISIMIP3b's SSP5-RCP8.5 and SSP3-RCP7.0 socioeconomic-climate scenarios, LUH2's land-use projections fall between the range of outputs reported by the LUMs for the different land-use types. Regarding land-use management variables, second-generation bioenergy cropland peaks around 2070 in LUH2 projections in SPP5-RCP8.5 and SSP3-RCP7.0. ISIMIP3b global average projections grow steadily, with a slightly steeper rate for SSP5-RCP8.5. The growing rates of ISIMIP3b projections in these socioeconomic-climate scenarios are notably flatter, with no peak than the LUH2 data set, showing lower demand for cropland areas destined for second-generation bioenergy crops in ISIMIP3b. Concerning synthetic nitrogen fertilizer use in SSP5-RCP8.5 and SSP3-RCP7.0, ISIMIP3b projections, especially MAgPIE's, show higher values than LUH2, which could be related to a slightly higher cropland area in ISIMIP3b's MAgPIE estimates. Finally, for irrigated land, the trends are completely distinct in SSP3-RCP7.0 among the ISIMIP3b and LUH2 projections. For SSP5-RCP8.5, LUM projections show a smaller irrigated area during the time horizon than LUH2.

## 3.2 Spatially explicit intercomparison and uncertainty hot-spots

### 3.2.1 Cropland

In the grid-cell level analysis across LUMs×GCMs per scenario, ASIA displays the highest median value of cropland per grid cell (Figures 4 and B4 in the Appendix). In contrast, REF displays the lowest in 2050 in all scenarios, similar to 2015's regional cropland distribution (see Figure B3 in the Appendix). Regarding scenario differences, SSP3-RCP7.0 displays a larger median than the other two scenarios in ASIA. In other regions, such as LAM, MAF, or the OECD, SSP3-RCP8.5, and SSP5-RCP8.5 have similar values of median cropland areas per grid cell in 2050. In 2100, ASIA remains the region with the highest median allocation of cropland per grid cell only for SSP1-RCP2.6, while MAF becomes the region with the highest median for the SSP5-RCP8.5 and SSP3-RCP7.0, being SSP3-RCP7.0 median value larger than that of SSP5-RCP8.5. In 2100, although the SSP5-RCP8.5 assumes a population reduction in the second half of the century and, consequently, demand for agricultural products, SSP1-RCP2.6 remains, in all regions, as the scenario with the lowest median.

Appendix B5 includes maps of land-use types in 2015 (LUH2 values) and average values per grid cell across LUMs×GCMs for each socioeconomic-climate scenario in 2050 and 2100 (Figure B7).

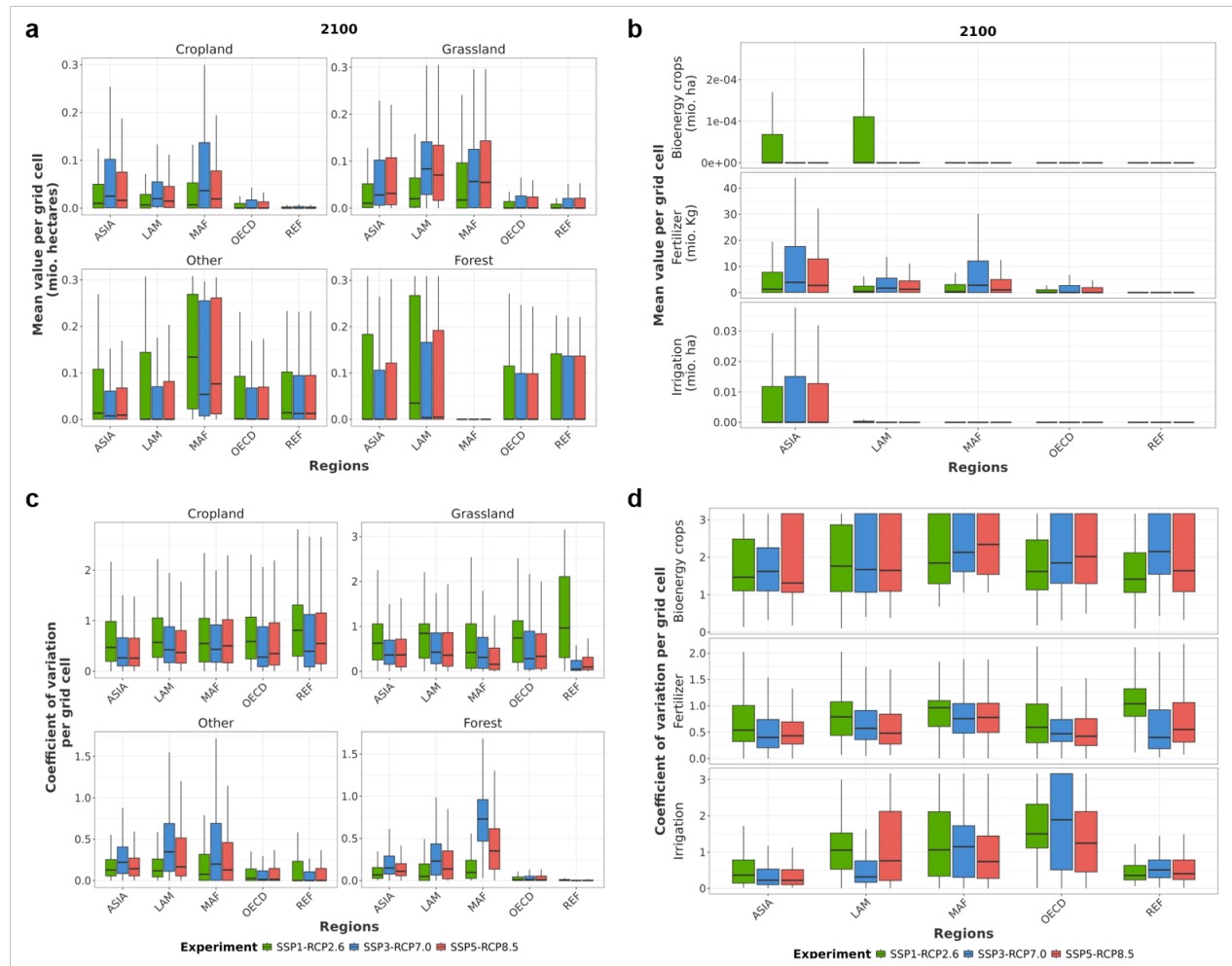

**Figure 4. Boxplot representation of grouped cells per region, variable, and socioeconomic-scenarios in 2100** a) Displays the distribution of average land-use types area per grid cell and b) the distribution of second-generation generation bioenergy crop area (Bioenergy crops), synthetic nitrogen fertilizer use (Fertilizer), and irrigated cropland (Irrigation). c) Shows the distribution of the coefficient of variation of land-use types area per grid cell calculated based on ten simulations (2 land-use models x impact data based on five global climate models) and d) the distribution of second-generation bioenergy crop area, synthetic nitrogen fertilizer use, and irrigated cropland

Regarding the distribution of the coefficient of variation per grid cell across LUMs×GCMs and within the regions, MAF shows the largest median value in 2050 in all scenarios (Figure 4c). In 2100, REF has the largest median CV for SSP1-RCP2.6 and SSP5-RCP8.5 and MAF for SSP3-RCP7.0. REF's behavior is related to its small allocation of cropland per grid cell, while MAF's is due to different allocation dynamics among LUMs. Although cropland area demand is the lowest in all regions in SSP1-RCP2.6, compared to other scenarios, the median CV per region and grid cell is larger than in other socioeconomic-climate scenarios and increases between 2050 and 2100. Despite similar trends in projections from LUMs on the aggregated

level (global and regional) in SSP1-RCP2.6, the large CV in this scenario indicates major differences in allocation and impact distribution between the LUMs on the grid level.

Additionally, it can also be observed that in highly concentrated cropland areas, the coefficient of variation is lower than in more dispersed cropland areas for all scenarios, which holds for the other land-use types. This behavior can be seen, e.g., in India in Figures 5a and 6a, one of the largest crop producers in ASIA and the world (Food And Agriculture Organization Of The United Nations, 2024). Finally, although cropland uncertainty hot spots vary for the different scenarios, East Africa, Australia, and Central Asia consistently display high coefficients of variation in the cropland area for the three SSP*x*-RCP*y* across LUMs×GCMs and years (Figure B8 in the Appendix).

The supplementary Figure B8 in the Appendix provides a visual global representation of the coefficient of variation per grid cell based on LUMs×GCMs simulations and for each SSP*x*-RCP*y* in 2050 and 2100.

### 3.2.2  Forests

In 2050 and 2100, the median forest area per grid cell is highest under SSP1-RCP2.6 compared to SSP3-RCP7.0 and SSP5-RCP8.5 and increases over time (Figure 4 and B4 of the Appendix), reflecting the protection policies associated with the SSP1-RCP2.6 narrative. Specifically, LAM (Amazon rainforest, Figure 5b), followed by ASIA (Southeast Asian rainforests), has the largest median forest area per grid cell in all socioeconomic-climate scenarios. Conversely, MAF has the lowest median forest area per grid cell (close to zero) and the highest median coefficient of variation across all regions and scenarios. Uncertainty is particularly high in the African tropical rainforests (ATR) and the SSP3-RCP7.0 scenario.

### 3.2.3  Grassland

While MAF continues to have the highest median grassland area per grid cell in all regions under SSP1-RCP2.6 in 2050 compared to 2015, a shift to LAM is observed in SSP3-RCP7.0 and SSP5-RCP8.5. This shift results in a higher median grassland area per grid cell in LAM compared to other regions across all scenarios by 2100.

Among the scenarios, SSP1-RCP2.6 has the lowest median grassland area per grid cell across all regions in 2050 and 2100. Although in SSP1-RCP2.6, global and regional aggregated LUMs×GCMs projections agree with a reduction of grasslands, and on the rate of change, the median CV per grid cell is the largest in all regions. This suggests differences among LUMs on the locations where grasslands could be reduced under sustainable scenarios for afforestation or reforestation, exemplified by the fact that in SSP1-RCP2.6, northern hemisphere boreal forests, and the Amazon rainforest are hot spots of uncertainty for grasslands. The median grassland area and coefficient of variation per grid cell are similar between SSP3-RCP7.0 and SSP5-RCP8.5 in most regions in 2050 and 2100, with slight differences in the MAF and OECD regions (Figures 4 and B4). Hot spots of uncertainty include Central and East Europe (Figure 5c and 6c).

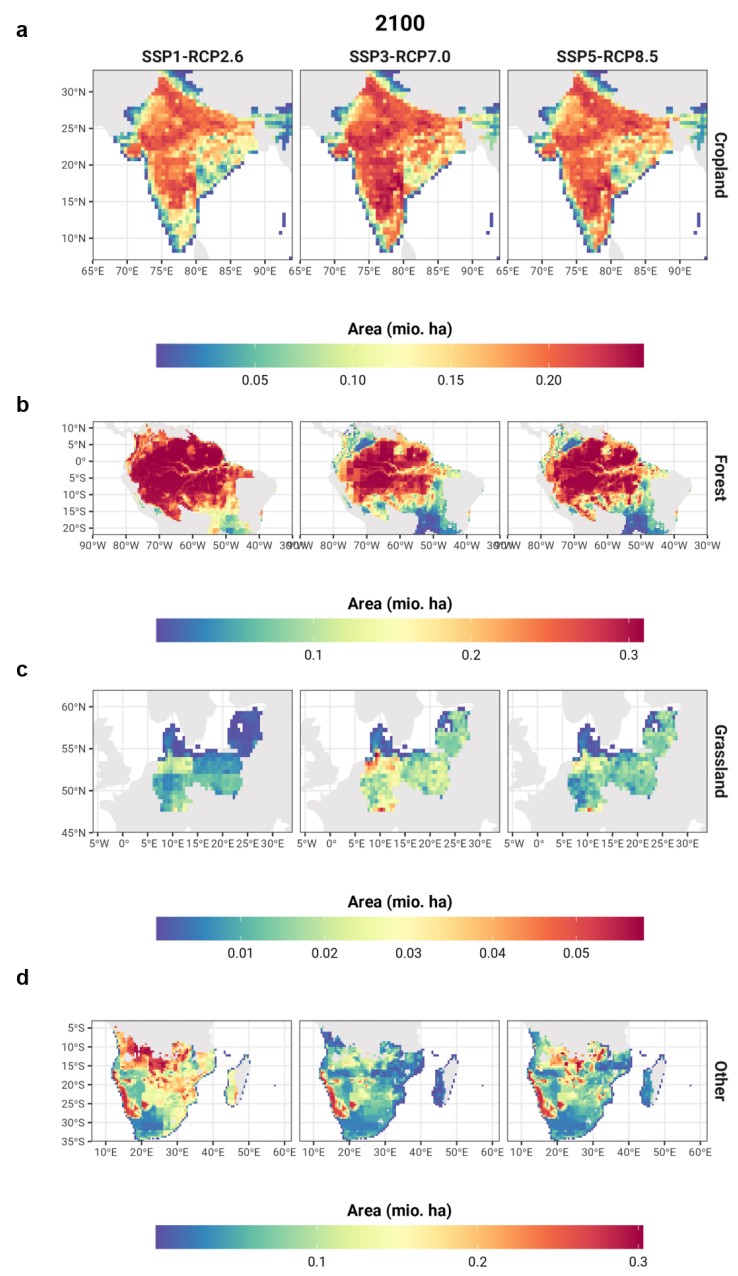

**Figure 5. Mean area calculated per grid cell for different land-use spatially explicit projections for different areas of interest under three socioeconomic-climate change scenarios, SSP1-RCP2.6, SSP3- RCP7.0, SSP5-RCP8.5.** a) Shows grid cell cropland projections for the subcontinent of India in units of millions of hectares (mio. ha), b) the Amazon rainforest, c) Grassland area in central and east Europe,

and d) Other natural vegetation in Southern Africa. The mean was calculated using ten simulations (two land-use models x impact date based on five climate models) per SSP*x*-RCP*y*

### 3.2.4 Other natural vegetation

Due to the extensive size and the difficulty of converting the Sahara subregion to other land uses, MAF consistently shows the largest median area of other natural vegetation per grid cell across all regions, scenarios, and years. In the SSP1-RCP2.6 scenario, the median area of other natural vegetation is higher than that of other socioeconomic-climate scenarios and increases over time. This trend is observed in MAF and other regions as well. In contrast, for SSP3-RCP7.0 and SSP5-RCP8.5, the median of other natural vegetation area per grid cell declines over time, with SSP5-RCP8.5 having a slightly higher median than SSP3-RCP7.0 across all regions.

Regarding the CV, its median is highest in all regions for SSP3-RCP7.0 over time. For example, southern Africa, a region with rich and diverse ecosystems, exemplifies this trend (Figures 5d and 6d). Key high-uncertainty regions for the LUMs×GCMs ensemble include Southeast South America, the Sahel, and the east coast of Australia.

### 3.2.5 Second generation bioenergy

Second-generation bioenergy crops (Figures B6, B9-B12) are generally allocated in concentrated and highly fertile areas across all scenarios. These areas primarily include the west coast of Australia, southern Brazil, Eastern European Plain (especially in SSP1-RCP2.6), Southeast Asia, southern China, and West Africa. The SSP1-RCP2.6 scenario has the largest median second-generation bioenergy crop areas per grid cell in 2050 and 2100 across regions, corresponding to the higher demand seen on global and regional levels.

Despite high uncertainty for bioenergy crops (median CV greater than one across all regions over time) (Figure 4 and B4), specific allocation sites show high agreement among LUMs. These sites include parts of the Atlantic forest in southeast Brazil, southern China and the North China Plain, mainland Southeast Asia (Indo-Burma region), and the West African forest, which are also biodiversity hotspots (Myers et al., 2000).

Unlike other land-use variables, LUMs do not include initial maps of second-generation bioenergy cropland for the historical period. Thus, differences in allocation among LUMs arise from the absence of historical data on dedicated second-generation bioenergy cropland locations and the distinct allocation rules of each LUM. Both models allocate bioenergy crops based on biophysical suitability. However, in MAgPIE, bioenergy crops must compete with other land uses and crop types. Since REMIND determines regional demand and trade flows, each region must fulfill its requirements in the land-use model. In contrast, in IMAGE, cropland dedicated to bioenergy is confined to abandoned agricultural lands or, when insufficient, to natural grasslands.

### 3.2.6 Irrigation and synthetic nitrogen fertilizer use

Across all scenarios for 2050 and 2100, irrigated areas (Figures B6, B9-B12) still correspond to historically irrigated areas. They are primarily located in ASIA along the Ganges and Indus rivers, along main river basins in China (e.g., Hai He, Huang

rivers), and the Arvand River in Iran. The low CV in these regions indicates strong agreement among the LUMs in all scenarios. The median of projected irrigated areas per grid cell is highest in the SSP5-RCP8.5 and SSP3-RCP7.0 scenarios for both 2050 and 2100, with SSP3-RCP7.0 showing slightly higher irrigation utilization across all regions, which could be related to higher

cropland area demand in these scenarios. The median coefficient of variation per grid cell for the LUMs×GCMs ensemble is highest in SSP1-RCP2.6 in most regions, reflecting reduced irrigation due to lower agricultural commodity demand. High uncertainty areas include Northern Europe and Australia (OECD countries).

While the SSP3-RCP7.0 and SSP5-RCP8.5 scenarios indicate a higher nitrogen fertilizer use per grid cell, China consistently exhibits the highest usage, followed by India, the American Corn Belt, and Brazil in all scenarios for 2050 and 2100 (Figures

B9 and B10). Throughout regions, fertilizer use is lowest under SSP1-RCP2.6 and decreases over time, resulting in a higher median CV as time progresses. The regions with the largest uncertainty include northern Australia and East Africa.

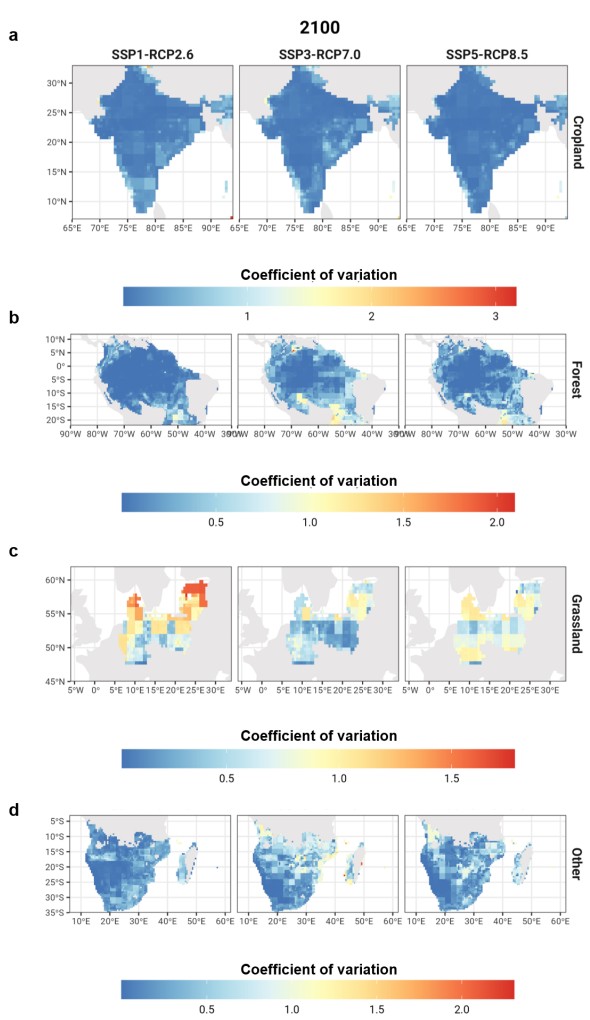

**Figure 6. Coefficient of variation calculated per grid cell for different areas of interest under three socioeconomic-climate change scenarios, SSP1-RCP2.6, SSP3- RCP7.0, SSP5-RCP8.5.** a) Shows the coefficient of variation calculated for cropland for the subcontinent of India, b) forest area in the Amazon rainforest, c) Grassland area in central and North-East Europe, and d) Other natural vegetation in Southern Africa. The coefficient of variation was calculated using ten simulations (two land-use models x impact data based on five climate models) per SSP*x*-RCP*y*

## 3.3 Variance analyses

### 3.3.1 Global and regional projections

Generally, the variance, measured as the total sum of squares (Figure B13 in the Appendix), starts at zero and increases with time for all variables and regions for the harmonized datasets. After 2030, the variance analysis shows that the variance of the different land-use and land-use management projections through the century can be explained mainly by the differences among the scenarios rather than by the LUMs or the interactions among factors (Figure 7) on the global level. The GCM factor has little or no share in explaining the variance among projections. GCMs only make a small difference for irrigation on the global

level and for the REF region, where this factor explains a small share of the variance for cropland and other natural vegetation until the first half of the century (Figure B14 in the Appendix).

Differences in the LUMs largely contribute to variance in the projections, particularly for other natural vegetation, forests, and grassland, before 2030, where variance is lower (Figure B14). This also holds true globally and in regions such as ASIA, MAF, and the OECD, even though differences are small among the scenarios on the global level. This is in line with the climate

and socioeconomic (population, income, diet, and others) assumptions, where the largest differences start taking place around 2030 and start diverging more strongly in the second half of the century (Figure 7 and Figure B13 in the Appendix) (Popp et al., 2017; Müller et al., 2021).

Scenario differences contribute most significantly to the overall variance in second-generation bioenergy crop projections, both globally and regionally ,especially in ASIA and the OECD, around the 2060-2070 period. Afterward, LUMs and/or the

460 Interactions factor have a higher share of explaining the variance than the other factors. The differences among LUM models regarding second-generation bioenergy projections suggest challenges for long-term bioenergy with carbon capture and storage (BECCS) and related mitigation policy on the global and local levels since, under the same scenario, LUMs display different second-generation demand and production sites. In the case of fertilizer use, although the Scenario factor has a higher impact on variance, the shares of the Interactions (at the global scale and for LAM and MAF) and LUM (OECD and REF) factors

contribution to variance are individually comparable to those of the Scenario factor.

LUM and Scenario are the two factors that have the highest influence on variance for grasslands globally throughout the century. Specifically, differences in LUM dynamics have the strongest influence until 2050, when the Scenario becomes the factor with the highest share of the variance. This behavior is similar for the ASIA, the OECD, and MAF regions. For LAM, LUM explains the variance for grassland until almost the end of the century.

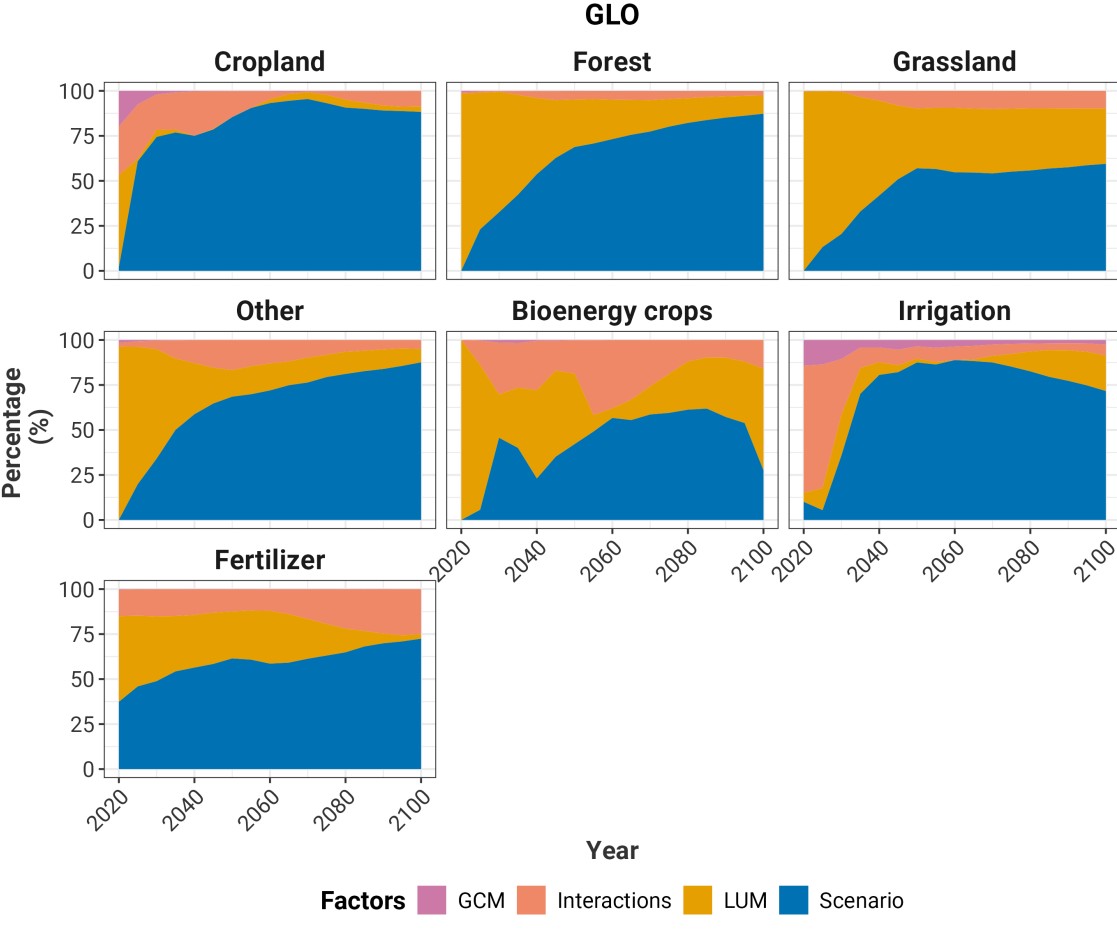

**Figure 7. Fraction of variance explained by the specific factors for the harmonized global land-use and land-use management projections.** GCM stands for the global climate models used to generate the climate impact inputs used by the Land-Use Models (LUMs). The Scenarios factor relates to the different SSP*x*-RCP*y* scenarios. Finally, the Interactions factor refers to the residual, assumed here as the interactions between the different factors

## 3.3.2  Grid-level analysis and harmonization effects

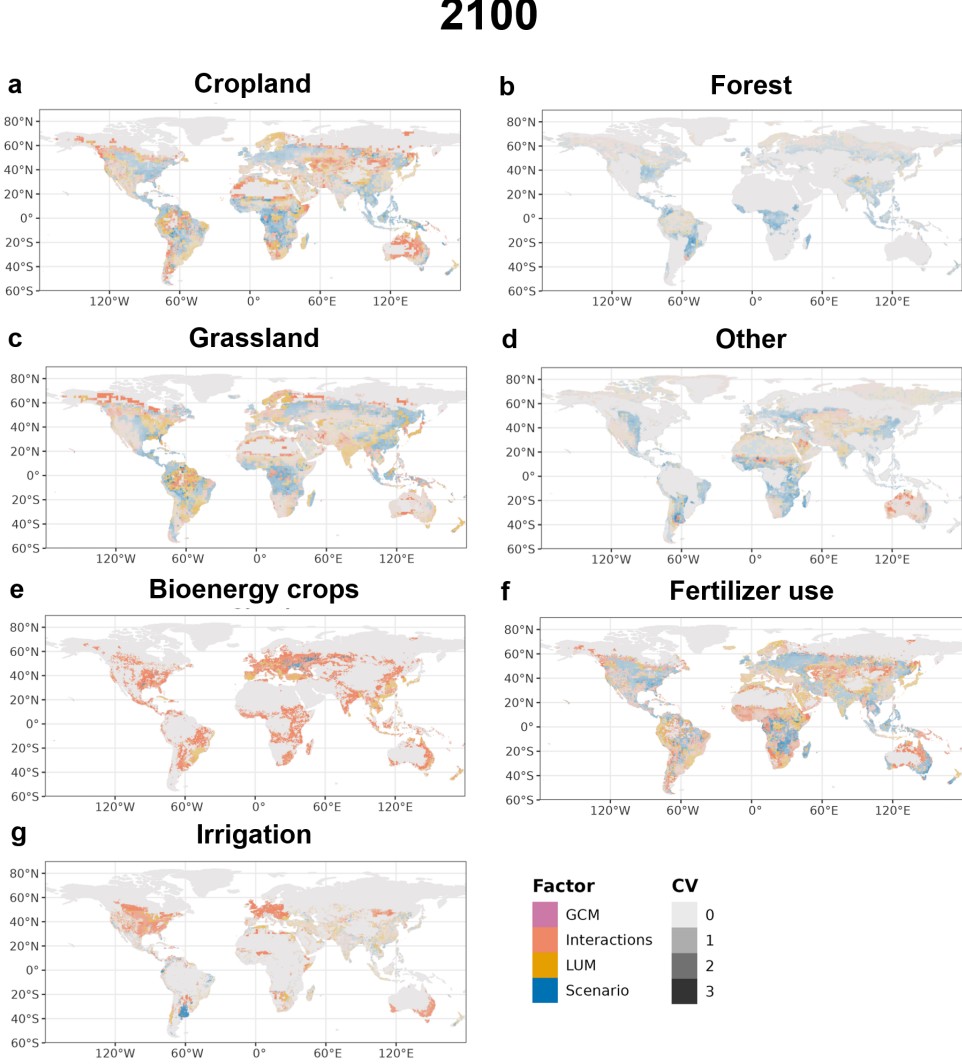

**Figure 8. Highest fraction of variance explained by the specific factors for the harmonized spatially explicit land-use and land-use management projections in 2100.** GCM stands for the global climate models used to generate the climate impact inputs used by the Land-Use Models (LUMs). Scenario relates to the different SSP*x*-RCP*y*. The Interactions factor refers to the residual, assumed here as the interactions between the different factors. In the maps, the color represents the factor (LUMs, GCMs, Scenario, and Interactions) that explains the highest share of the variance in each cell, and the opacity (lower values correspond to more transparent colors) depicts the coefficient of variance of each cell calculated based on 30 simulations (two LUMs x five GCMs x three SSP*x*-RCP*y*)

By 2100, compared to 2050, the Scenario becomes the factor with the highest share explaining variance in most grid cells for cropland, other natural vegetation, and forests (Figures 8 and B15). Specifically for cropland in high-producing regions within the USA, South East Asia, and Europe, the variance per grid cell can be explained to a large extent by the Scenario factor in 2100, which points toward large differences among impacts under different climate and socioeconomic pathways in these regions, a better agreement between LUMs dynamics, and/or better data availability on these areas. In the case of forests, the level of agreement among LUMs is related to the fact that they are large and highly concentrated (compared to cropland or grassland) and, in the case of natural vegetation, are hard to convert to other land types (e.g., the Saharan desert).

As in the regional and global analyses, the GCM factor can explain the variance to a greater extent only in a few cells of the different land-use and land-use management variables.

For grassland, fertilizer use, irrigation, and especially second-generation bioenergy crops, the Interactions factor explains the variance for most grid cells in 2050 and 2100.

The fact that the Interactions factor is significant compared to the other factors highlights the complexity of the relationships between land-use patterns and the GCMs, LUMs, and scenarios studied. Equation 2 in the methods section simplifies highly complex systems, spanning climate, crop, energy, and land-use models, as the workflow diagram shows (Figure 1). Therefore, a significant contribution from the Interaction factor highlights the varying sensitivities and complexity of modeling different land-use variables and the effect that climate impacts and socioeconomic growth assumptions have on them.

In the case of irrigation, other factors have the highest share in a few regions. Particularly regarding the Pampas in South America, the Scenario factor has the highest contribution to variance. For grasslands and fertilizer use, the picture is mixed. In grassland, while in some regions within China, the Scenario makes the largest difference in variance, in others like South Brazil, India, and the USA, LUMs differences have a higher influence. For fertilizer, for a large user such as China, for example, LUMs and Scenario explain a similar number of cells' variance compared to the Interactions factor. However, the Scenario factor explains the variance in most cells in other regions, such as India, the USA, or Indonesia.

Finally, the effects of harmonization (Figure 9) on high-resolution projections are evaluated through an additional analysis of variance considering high-resolution harmonized and raw projections (unharmonized projections reported by the land-use models). Harmonization greatly impacts fertilization use and forest spatially explicit projections. Specifically for forests in central and east Europe and northeast Russia, harmonization has the largest contribution to variance. One of the primary explanations for the effect of harmonization on forests is the difference between the LUMs' reference datasets and LUH2 historical maps used in harmonization, especially in areas with intermediate tree cover.

Table A2 in the Appendix shows the differences between the starting or reference maps, the modeling dynamics, and details regarding definitions among the land-use models for the different variables here studied. Especially for forests, definitions (e.g., based on potential standing stock thresholds), inputs or reference data sets, and calculation methods differ, explaining the large effect of harmonization on forest patterns. These differences in definitions are not only seen in the context of LUMs. For example, global forest areas in 2000 ranged between 3600 and 4300 million hectares among different satellite sources and FAO (Ma et al., 2020).

# 2100

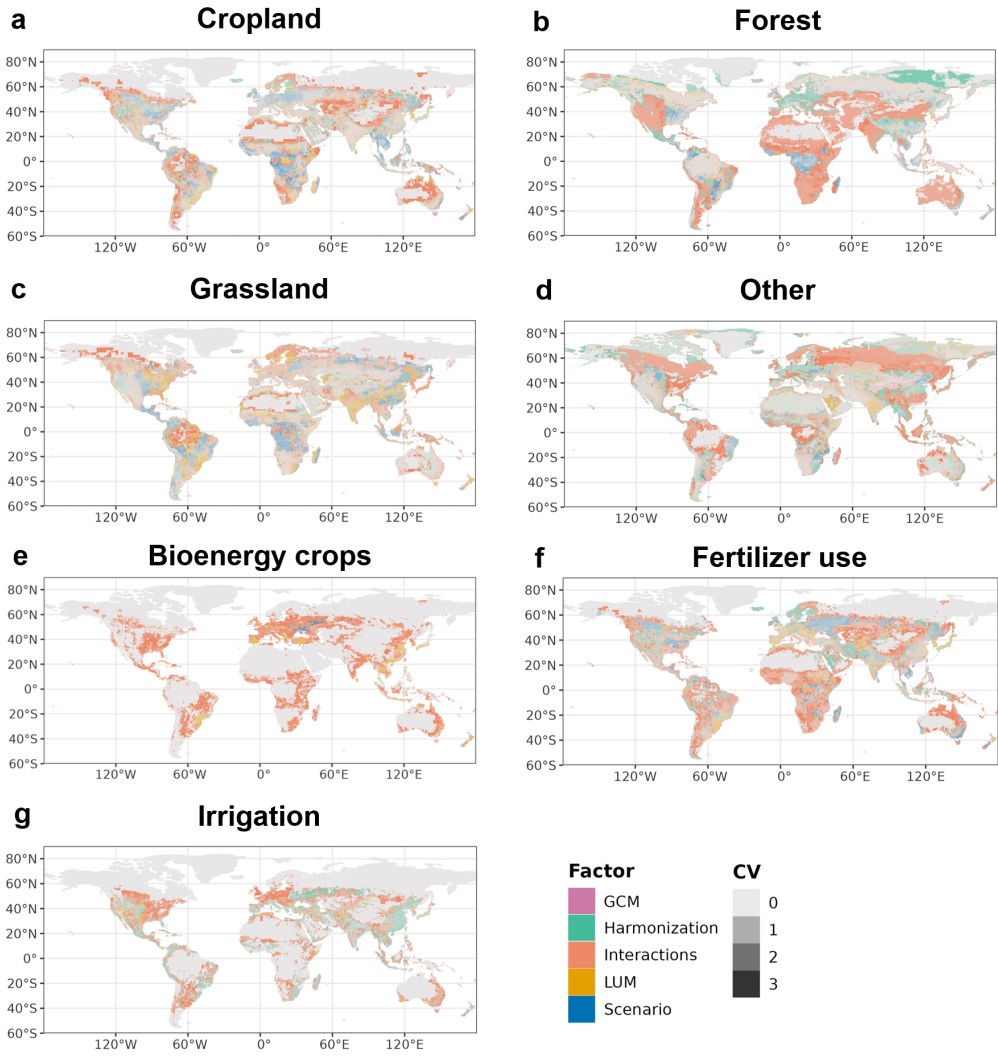

**Figure 9. Highest fraction of variance explained by the specific factors for the harmonized and raw spatially explicit land-use and land-use management projections.** GCM stands for the global climate models used to generate the climate impact inputs used by the Land-Use Models (LUMs). Scenario relates to the different SSP*x*-RCP*y*. The harmonization factor represents the variance associated with the harmonized and unharmonized sets. Finally, the Interactions factor refers to the residual, assumed here as the interactions between the different factors. In the maps, the color represents the factor (LUMs, GCMs, Scenario, and Interactions) that explains the highest share of the variance in each cell, and the opacity (lower values correspond to more transparent colors) depicts the total sum of squares

## 4 Discussion and conclusion

This paper compares and assesses the land-use and land-use management projections generated by two land-use models as direct human forcing input to ISIMIP3b, and their uncertainties for multiple spatial resolutions (global, regional, and $0.5° \times 0.5°$).

For the SSP1-RCP2.6 scenario, we found that global trends of different land-use types are very similar across the LUMs×GCMs ensemble. However, we found some differences regarding the regional and local distribution of land-use change, specifically in cropland for the LAM region. This is most likely due to a higher demand for bioenergy crops in this area in MAgPIE compared to IMAGE. For SSP5-RCP8.5 and SSP3-RCP7.0, global and regional trends disagree regarding the direction of change of grassland area, which leads to differences in forests and natural vegetation. A possible explanation for this behavior is the expected increase in livestock products in the SSP5-RCP8.5 and SSP3-RCP7.0 scenarios. Higher demand for meat and dairy products leads to a greater need for grasslands and crops used as animal feed. Both models account for the feed mix required to meet animal energy needs, considering factors like production systems types and feed conversion. However, how these demands and shares of the feed mix are estimated differs between the models, which can lead to varying projections for grassland use. On the one hand, in MAgPIE, grassland intensification and reliance on crop-based feed sources reduce the need for grassland expansion in scenarios with high demand for livestock products. On the other hand, although IMAGE moves to more intensive livestock systems as well, the share of grass in the feed mix stays relatively high—especially in SSP3-RCP6.0—resulting in a grassland expansion. For information on livestock systems modeling in IMAGE, refer to Bouwman et al. (2005); Lassaletta et al. (2019), and for MAgPIE to Weindl et al. (2017a, b). In this case, LAM is one of the regions most affected by the disagreement in grassland projections. Latin America is one of the regions with high economic inequality and biodiversity concentration, which could be highly vulnerable to climate change impacts, and mitigation due to its large potential for re/afforestation and BECCS (Hirata et al., 2024; Kim and Grafakos, 2019; Calvin et al., 2014; Reyer et al., 2017). In general, differences in land-use projections are expected to directly affect the impact models that use this data as input. For instance, grasslands are among the ecosystems with the highest wildfire frequencies (Donovan et al., 2017). Therefore, uncertainty in LUMs×GCMs grassland projections could influence the identification of fire hotspots due to human-induced effects (Thompson and Calkin, 2011). Uncertainty propagation stemming from land-use patterns could also impact, e.g., the calculation of emissions from land-use transitions (Neuendorf et al., 2021), shifts in biomes (Alexander et al., 2017), the assessment of ecosystem services, habitat intactness, and biodiversity (Yang et al., 2024), among others.

While both MAgPIE and IMAGE simulate the land-use system by accounting for future socioeconomic, biogeochemical, and biogeophysical changes, they differ in their setups. These differences may partly explain discrepancies in global projections for cropland and grassland areas under some scenarios, as well as the significant influence of the LUM factor on variance for certain variables at spatially explicit levels. A key distinction lies in the economic modeling approach. MAgPIE is a partial equilibrium model focused on the agricultural sector, whereas the IMAGE framework uses the CGE model MAGNET, which accounts for the entire economy. Additionally, MAgPIE's cropland allocation is based on minimizing production costs and local biophysical constraints, while IMAGE's approach relies on a constant elasticity of transformation function, which associates land supply responsiveness with changes in yields and prices (Schmitz et al., 2014). Previous studies, e.g., Alexander et al.

(2015), have shown that CGE models often project lower cropland areas. This outcome is likely due to factors such as input substitutability, interactions between agriculture and other economic sectors, and their effects on prices, demand, and supply of agricultural commodities and inputs. Another major difference involves the use of the LPJmL model. MAgPIE employs LPJmL outputs as exogenous inputs, while IMAGE integrates LPJmL dynamically. As Doelman et al. (2022) highlighted, the dynamic coupling of crop, hydrological, and vegetation models can influence estimates, leading to variations in projected biophysical conditions on the spatially explicit level under similar scenarios. Finally, the approach to technological change (TC) is another critical factor. TC directly impacts yields for cropland and grassland, which, in turn, affects land demand and competition, contributing to variations in land-use projections. For further details on the key processes modeled in IMAGE and MAgPIE, please refer to Tables A1 and A2 in the Appendix.

The difference among LUMs regarding land-use change and agricultural management for the different socioeconomic-climate scenarios also highlights the importance of model and data set development. Due to impacts and model dynamics updates, regional and national studies in integrated assessment models (IAMs) are needed as much as periodical model inter-comparison exercises. On the one hand, for example, LUMs have been used to conduct region-specific studies. For instance, MAgPIE has performed assessments focused on China (Wang et al., 2023) and India (Singh et al., 2023), while IMAGE has examined the European Union (Veerkamp et al., 2020). These studies have involved further development and validation of the models' outputs for these regions. It is important to note that China, India, and Europe are among the largest producers of agricultural commodities—often referred to as 'breadbaskets'—and have received considerable attention from the scientific community studying the agricultural and food systems. In our study, as shown in Figures B8, B11, and B12, the coefficient of variance in these regions, particularly for cropland area, fertilizer use, and irrigation, is relatively low compared to other areas. This remains true even under scenarios such as SSP3-7.0 and SSP5-8.5 toward the end of the century. These findings highlight the importance of expanding research to less-studied regions and land-use variables. On the other, the identification of uncertainties to better understand land-use and land-related dynamics on different resolutions among LUMs is key, e.g., for climate change mitigation and adaptation decision-making and to reduce, as much as possible, incompatibility among sustainability targets (e.g., growing second-generation bioenergy crops in biodiversity hotspots).

Regarding second-generation bioenergy crops, we found an agreement among LUMs regarding the peak period with the highest crop area for the low-emissions scenario, which is congruent with mitigation targets. Nonetheless, the peak size differed between MAgPIE and IMAGE, with MAgPIE being almost double that of IMAGE. Previous studies, as in Popp et al. (2014b), suggest that such differences among models on the global and regional level can be associated with bioenergy prices, energy deployment levels and make-ups, crop yields, assumptions about economic and technological growth, biomass resources, and sensitivities to other variables. However, compared to the LUH2 projections used in CIMIP6, ISIMIP3b MAgPIE's peak is considerably lower. This lower needed second-generation bioenergy crop area than previously calculated in the SSP1-RCP2.6 scenario could imply lower environmental impacts of bioenergy crop deployment due to less water consumption, conversion of land, or soil erosion (Wu et al., 2018; Calvin et al., 2021).

Respecting other land-use types, the larger reduction rate of grasslands and larger increase rate of forests during the century than LUH2, in the SSP1-RCP2.6 scenarios could, for example, impact previous estimations related to water resources (Shah

et al., 2022) and biodiversity indicators based on species adapted to open (e.g., grasslands) or closed ecosystems (e.g., forests) (Bond, 2021), among others.

Concerning variance, although input uncertainty increases as emissions grow (Molina Bacca et al., 2023; Jägermeyr et al., 2021), there is also high uncertainty in spatially explicit outputs for the SSP1-RCP2.6 scenario among the LUMs. This behavior likely occurs due to the LUMs' different land-use allocation, intensification dynamics, and interpretation of socioeconomic development narratives. The shrinkage of grassland, forests, or fertilizer use in sustainable development scenarios can happen in different regions and socioeconomic or ecological contexts, which are interpreted differently by the LUMs based on the model type, inputs, substitution elasticities, and assumptions made in processes such as trade (Schmitz et al., 2014). This supports the importance of considering local impact studies to complement global studies for informed decision-making on different government and cooperation levels. Particularly, cropland uncertainty hot spots include countries and regions such as East Africa (Somalia) and Central Asia, which are under a critical food insecurity risk - due to limitations derived from their geopolitical, socioeconomic, geographical, landscape (e.g., delicate ecological systems) and climatic impact contexts, supporting previous works, which highlights the vulnerability of these regions (Su et al., 2024; Boitt et al., 2018).

For the spatially explicit projections of grasslands, forests, other natural vegetation, and second-generation bioenergy crops, we identified forest areas such as the African and Amazon rainforests, boreal forests of the northern hemisphere, the Brazilian Atlantic forest, and the Indu-Burma Region as key regions of uncertainty for the LUMs×GCMs ensemble. The uncertainty in these areas for multiple land-use types and second-generation bioenergy crop areas pinpoint the tight link between the food demand, biodiversity protection, and climate impacts (Behnassi et al., 2022). For example, given that most mitigation pathways rely heavily on BECCS (Calvin and Fisher-Vanden, 2017), the uncertainty and the specific allocation of second-generation bioenergy cultivation sites could represent challenges for global and local mitigation policy-making and biodiversity protection (Hirata et al., 2024). Finally, at the spatially explicit level, Australia was an uncertainty hot spot for cropland, natural vegetation, second-generation bioenergy, fertilizer use, and irrigation area projections. This result agrees with previous work from Prestele et al. (2016) that uses a different methodology and set of projections and where Australia is also a hot spot of uncertainty for cropland area projections. The LUM factor explains almost a third of the variance in this case.

Besides modeling dynamics and assumptions, another source of uncertainty in the high-resolution patterns reflected in the LUM factor of the variance analysis are the different downscaling procedures used by the models. Disaggregation of LUMs outputs to high-resolution levels is critical in determining spatially explicit land-use patterns and could contribute to uncertainty if different algorithms are used. During the harmonization process, the original gridded data reported by the LUMs is aggregated to a $2° \times 2°$ resolution and subsequently harmonized and disaggregated to $0.25° \times 0.25°$ using the approach described in Hurtt et al. (2020). However, the different algorithms the LUMs use to disaggregate their outputs introduce uncertainty on where the reduction or expansion of cropland, or other land types, occurs, affecting fertilizer and irrigation patterns on the spatially explicit level.

The uncertainties observed in land-use variables at different resolutions arise from error propagation throughout the modeling workflow, as well as from scenario narrative modeling approaches and other factors. These uncertainties highlight the need for conscientious use of the reported data, carefully considering its limitations and assumptions. The objective of the data is to

provide a global overview of land and agricultural systems and their development under a set of socioeconomic and climate scenarios based on different assumptions.

While our study's primary focus is not to provide direct policy recommendations, it could offer some general insights. For example, our study suggests the need to promote sustainable grassland management practices and diversified feed mixes for livestock to balance ecological, environmental, and economic demands, particularly in regions like LAM and MAF, where grasslands are projected to grow, especially under the SSP3-RCP7.0 and SSP5-RCP8.5 scenarios in IMAGE's simulations. Also, building adaptive capacity could be key to addressing uncertainties in land-use changes and management projections. It would need to prioritize region-specific strategies that reconcile agricultural and environmental priorities. Key uncertainty hotspots in our study include the allocation of cropland in East Africa, Central Asia, and Australia; forest areas in the African tropical rainforest (ATR); grasslands in Central and Eastern Europe; and other natural vegetation in Southeast South America, the Sahel, and the east coast of Australia.

Another key point is the decline of forests and other natural vegetation in scenarios such as SSP3-RCP7.0 and SSP5-RCP8.5, especially in LAM, MAF, and ASIA. This emphasizes the urgency of prioritizing conservation efforts, monitoring, and dedicated policies to safeguard biodiversity-rich ecosystems.

Likewise, the differences in second-generation bioenergy crop allocation among models call for tailored regional strategies that support sustainable bioenergy expansion while considering local suitability and market demands. Developing a common framework for bioenergy crop allocation scenarios in land-use models (LUMs) could also help reduce uncertainty and suggest better methods for the sustainable allocation of bioenergy crops.

Projected increases in fertilizer use, particularly in Asia (notably China and India), Brazil, and the American Corn Belt due to their critical roles in food production, highlight the need for efficient fertilizer management practices. Building regional capacity to balance food security requirements while minimizing environmental impacts is essential. Finally, in the scenario with high agricultural demand (SSP3-RCP7.0), areas around rivers such as the Ganges, Indus, and Huang or Aravand rivers consistently appear as critical locations for irrigated cropland. Strengthening water management systems in highly irrigated regions will ensure sustainable irrigation practices and support long-term agricultural productivity.

In policy and management decision-making contexts, however, the data presented here should be seen as an overview of global trends. In other words, it is not intended to replace targeted assessments and actions specific to, e.g., country, local, or regional levels that include contextual requirements and knowledge—including input from communities and experts— that should be incorporated during the assessment and planning phases to ensure that proposed actions align with the actual needs of the stakeholders (Neuendorf et al., 2021).

This study differs from earlier studies because the harmonized land-use and land-use management future projections were based on impact data derived from bias-corrected CMIP6 climate model estimates, where $CO_2$ was considered under a standardized set of scenarios and climate models. Additionally, the analyses comprised cropland, forest, and grasslands and a set of land-use and land-use-related variables. However, one of the limitations of our work is that the analyses were performed using a small set of land-use models. This set was selected because the impact modeling teams' simulation capacities in the ISIMIP framework are limited and need to consider a wide range of factors other than land use, such as climate data from

a wide range of GCMs. Yet, despite these limitations, it is noteworthy that this is the first consistent land-use input data set from different LUMs for ISIMIP impact models, while in earlier rounds, only projections from one LUM were used (Frieler et al., 2017). Also, using projections from MAgPIE and IMAGE still gives options for variance assessments since they cover a large range of possible outcomes under the same scenarios compared to other land-use models (Stehfest et al., 2019). However, further analyses to evaluate, e.g., risks related to biodiversity protection and food security or variance of socioeconomic development-climate impacts on the agriculture, forestry, and other land-use sectors at different scales, would require a larger set of land-use models. Other limitations include that even though the projections were harmonized to LUH2 historical maps, different assumptions and inputs related to the SSP*x* narratives depend on each LUM team interpretation and sources of inputs, leading to important shifts due to harmonization (e.g., in forests in central and east Europe and north-east Russia). These shifts lead to mismatches between the original LUMs' crop and forest areas, their yields, and agricultural product demands, which drive land-use allocation decisions. Thus, future land-use model intercomparison exercises would greatly benefit from a standardized set of inputs and/or the interpretation of scenario narratives.

Our analysis revealed that land-use and land-use management projection uncertainty varies across resolutions and socioeconomic climate scenarios. Since these projections are crucial for networks such as ISIMIP, AgMIP, and GGCMI and are fundamental for assessing impacts, attribution, and decision-making across different scales related to climate change mitigation and adaptation in multiple sectors and disciplines, further analyses and intercomparisons at high-resolution levels are necessary. This will enhance our understanding of the socioeconomic drivers of land-use dynamics, the effects of climate-related policies on land use, and their associated uncertainties.

*Code and data availability.* Data sets and the scripts used for the analyses made in the study and creating the plots can be found at https://doi.org/10.5281/zenodo.12964394 and https://doi.org/10.5281/zenodo.12964533, respectively. MAgPIE version 4.4.0 documentation in https://rse.pik-potsdam.de/doc/magpie/4.4.0/ and code in https://github.com/magpiemodel/magpie/releases/tag/v4.4.0

# Appendix A: Supplementary tables

| Process | | IMAGE | MAgPIE |
|---|---|---|---|
| **Trade** | Detail of the process ⟶<br>Scenario ↓ | **Armington approach** | **Endogenous. Historical patterns of self-sufficiency until 2015 (FAO). After, trade barriers are relaxed based on the scenario.** |
| | SSP1-RCP2.6 | All tariffs are removed | Up to 20% of livestock and secondary products and 30% of all other traded commodities are freely traded in 2050, behavior stays until 2100 |
| | SSP3-RCP7.0 | Trade tariffs are increased by 10% | Up to 5% of livestock and secondary products, and 10% of all other traded commodities are freely traded in 2050, behavior stays until 2100 |
| | SSP5-RCP8.5 | All tariffs are removed | Up to 20% of livestock and secondary products and 30% of all other traded commodities are freely traded in 2050, behavior stays until 2100 |
| **Diets** | Detail of the process ⟶<br>Scenario ↓ | **Function of population and income** | **Driven by population, income, and demography details** |
| | SSP1-RCP2.6 | Preference for animal-based products decrease by 30% | Healthy and low mear diets, reduced food waste |
| | SSP3-RCP7.0 | Preference for animal-based products increase by 30% | Unhealthy and high meat consumption diets, high shares of food waste |
| | SSP5-RCP8.5 | Preference for animal-based products increase by 30% | Unhealthy and high meat consumption diets, high shares of food waste |
| **Management and Technological progress** | Detail of the process ⟶<br>Scenario ↓ | **Technological change based on FAO and GDP projections. Substitution between production functions** | **Endogenous decisions of irrigated production of crops. Endogenous intensification of inputs. Different levels of R&D and irrigation costs** |
| | SSP1-RCP2.6 | High development and management due to high GDP growth | low costs |
| | SSP3-RCP7.0 | Low management due to low GDP growth, stagnated technological development, limited diffusion of knowledge | high costs |
| | SSP5-RCP8.5 | High due to high population growth and strong technological development | low costs |
| **Protected Areas** | Detail of the process ⟶<br>Scenario ↓ | **Exogenous and based on WDPA database** | **Exogenous and based on WDPA database** |
| | SSP1-RCP2.6 | Increase to protection of 30% of all terrestrial area | WDPA |
| | SSP3-RCP7.0 | Decrease in protected areas to the areas that are strictly protected currently | WDPA |
| | SSP5-RCP8.5 | All currently protected areas without expansion | WDPA |
| **Bioenergy** | Detail of the process ⟶<br>Scenario ↓ | **Endogenously determined by the TIMER-energy model and based on land availability, the food demand balance, and decarbonization efforts** | **Based on bioenergy demand from REMIND** |
| | SSP1-RCP2.6 | Strong increase, mostly second generation after 2040 | Growing demand of second generation bioenergy peaking around 2070 |
| | SSP3-RCP7.0 | Continuation of current bioenergy use and modest uptake of the second generation in the second half of the country | Sustained growing demand (a lower rate than SSP5) |
| | SSP5-RCP8.5 | Continuation of current bioenergy use and modest uptake of the second generation in the second half of the country | Sustained growing demand (a lower rate than SSP1) |

**Table A1. Assumptions of the different land-use models for the different scenarios**

| Variable | Model | Reference starting point/maps of the models | General dynamics in the models | Details regarding historical sets/reference starting point |
|---|---|---|---|---|
| Cropland | IMAGE | HYDE 0.08°x0.08° (5 arc-min) | Crop allocation is driven by regional production, potential yields, intensity levels, and spatial suitability factors. Regional production aims to meet demand for agricultural commodities shaped by demographic and income changes. This process considers input factors (e.g., labor and land), technological advancements, income and price elasticities, as well as trade and environmental policies. | The HYDE database is based on FAO, subnational statistics, and ESA's land cover consortium maps. HYDE allocation rules for this version and land type include population density>0.1 cap/km2, areas with better soil suitability (according to the GAEZ FAO IIASA dataset), easy access, and temperatures higher than 0°. |
| | MAgPIE | LUHv2 (based on HYDE 3.2) 0.5°x0.5° | The model is driven by socioeconomic changes, such as population growth and income levels, which shape demand for agricultural commodities. Considering trade patterns, crop yields, and land-use competition, it optimizes cropland allocation, always aiming to minimize costs. | |
| Grassland | IMAGE | HYDE 0.08°x0.08° (5 arc-min) | As per crop production, grassland allocation is determined by regional requirements, yields, intensity, and suitability, primarily for livestock feed, which depends on socioeconomic changes and policies, production system types, feed conversion efficiencies, and animal productivity. | HYDE's grassland is based on FAO's and subnational statistics and ESA's land cover consortium maps. Allocation rules include removing urban and cropland areas from grid cells and information regarding population density, temperatures (above -10°), plant functional types, and climatic and soil properties. |
| | MAgPIE | LUHv2 (based on HYDE 3.2) 0.5°x0.5° | Pasture area depends on the demand for biomass from pastures to feed livestock and the intensity of pasture utilization ("pasture yield"). | |
| Forest | IMAGE | LPJmL (endogenously calculated) | Timber and biomass demand, conservation and afforestation policies, land competition, harvest efficiency, suitability, and carbon pools are the main determinants of forest allocation and area. IMAGE considers these management systems: 1) clear-cutting followed by natural or assisted regrowth, 2) selective logging of (semi) natural forests, and 3) forest plantations. | Forest extent is based on the potential biome map from IMAGE selecting all forest biomes (Stehfest et al., 2014). Forest growth productivity is determined by the LPJmL dynamic vegetation model. Historical patterns of forestry are determined endogenously by the model driven by region-based demand for timber from FAO and a process-based forestry model (Arets et al., 2011) |
| | MAgPIE | LUHv2 (own map) rescaled to match FAO country-level data (FRA 2015) | Forest modeling includes managed, primary, and secondary forests. The main feature of managed forests is afforestation for CDR, based on either NPI or NDC policies, and timber production. Drivers of change for primary and secondary forests, as for other natural vegetation land, include land competition, primarily with land for agricultural uses, area protection, and emissions policies. | LUHv2 has its own estimation of natural vegetation. From every grid cell, cropland, urban, water, and ice areas are removed, and whatever is left is assumed to be a natural vegetation area. Carbon stocks are calculated with the help of the MIAMI-LU global terrestrial model, and potential forest areas are assumed to be those where aboveground potential standing stocks are higher than 2KgC /m2. |

| Variable | Model | Reference starting point/maps of the models | General dynamics in the models | Details regarding historical sets/reference starting point |
|---|---|---|---|---|
| Other | IMAGE | LPJmL (endogenously calculated) | Changes in other land are the derivative from other model dynamics, predominantly the expansion of agricultural land as described above. | Other natural land is based on the potential biome map from IMAGE, where all non-forest biomes (Stehfest et al., 2014) are selected, and all agricultural land use areas are subtracted. |
| | MAgPIE | LUHv2 0.5°x 0.5° | See drivers for primary and secondary forest above. | See Forest details for MAgPIE above. Other natural vegetation areas are assumed to be those where aboveground potential standing stocks are lower than 2KgC/m2. |
| Urban | IMAGE | HYDE 0.08°x0.08° (5 arc-min) | Urban area increases over time is determined as a function of urban population (demographic changes) and a country- and scenario-specific urban density curve. | HYDE's built-up area calculation is based on historical satellite and country-level data of population and urban density. |
| | MAgPIE | LUHv2 (based on HYDE 3.2) 0.5°x 0.5° | Urban Land is an exogenous parameter based on the LUH2 -CMIP6 dataset based on IMAGE's projections varying with the SSP narratives. | |
| Bioenergy | IMAGE | IEA data on country level | The TIMER energy model defines bioenergy demand in IMAGE based on land supply, biomass productivity, input costs, and learning dynamics, which together influence bioenergy prices. Bioenergy prices are also influenced by carbon tax prices and mitigation costs based on scenario-specific emission targets. | Historical shares of bioenergy are based on IEA data and allocated endogenously in the model with a rule-based approach using land availability and productivity drivers at the grid level. |
| | MAgPIE | N.A. | MAgPIE's bioenergy crop patterns are based on bioenergy demand determined by the REMIND model, based on land supply, biomass productivity, input costs, and scenario-specific emission targets and bioenergy prices. | N.A. |
| Inorganic Fertilizer (Nitrogen) | IMAGE | FAO data on country level | The IMAGE framework includes a nutrient model that tracks nutrient flows (nitrogen and phosphorus) through wastewater discharges, soil nutrient budgets, and their environmental pathways, detailing the fate of soil nutrient surpluses. Fertilizer use is linked to crop production, considering the efficiency of nutrient uptake by crops. | Synthetic fertilizer use is based on FAO statistics and is allocated to the crop level using crop-specific nitrogen requirements (Beusen et al., 2015). |
| | MAgPIE | IFA data on world region | The demand for inorganic fertilizers is based on nitrogen flow balances made in each time step of the simulation in cropland and pasture soils using exogenous uptake efficiencies. | IFA's Inorganic fertilizer use database includes the most common fertilizers and industrial products derived from ammonia (NH3), for which the N content usually varies from 21 to 82%. |

| Variable | Model | Reference starting point/maps of the models | General dynamics in the models | Details regarding historical sets/reference starting point |
|---|---|---|---|---|
| Irrigated cropland (Area actually irrigated) | IMAGE | HYDE 0.08°x0.08° (5 arc-min) | Irrigated area expansion is exogenously prescribed based on historical trend extrapolation and scenario-specific assumptions (Doelman et al., 2018) | HYDE's area equipped for irrigation pattern is based on FAO and subnational statistics, Siebert et al. (2015), and MIRCA2000. Allocation rules include that the irrigated area should be inside cropland areas and surrounded by enough water availability (discharge maps). Areas with low aridity (precipitation/evapotranspiration) require more irrigation. |
| | MAgPIE | LUHv2 (based on HYDE 3.2) 0.5°x0.5° | The model endogenously determines investments in area equipped for irrigation (AEI). It considers that irrigated crop production is limited to areas with existing infrastructure and accounts for regional differences in unit costs per hectare for AEI expansion. The demand for irrigated cropland is determined by calculating cropland requirements based on the supply of demand for agricultural commodities and overall cost production minimization. (See above). | |

**Table A2. Details of the models' input sources and dynamics related to allocation and sizing of land-use types and management**

**Appendix B: Supplementary figures**

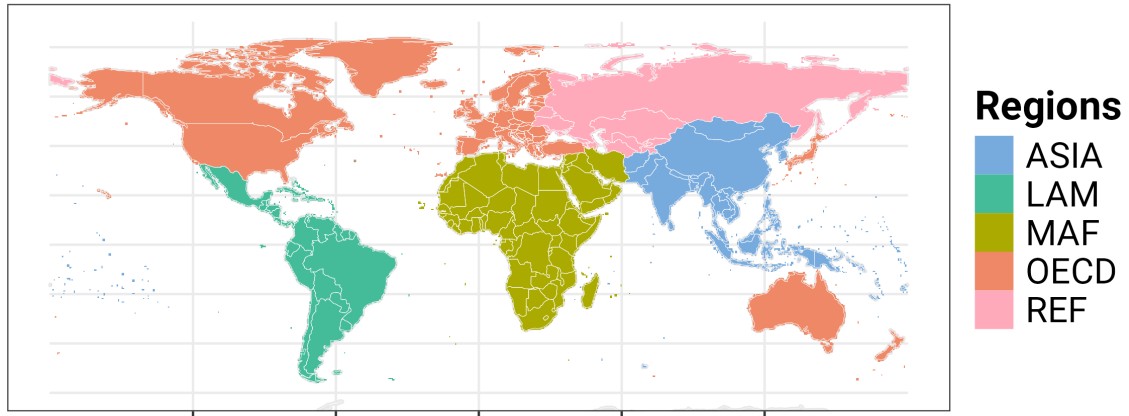

**Figure B1. Regions used in the regional analysis** ASIA stands for Asian countries not part of the former URSS, LAM for Latin American counties, MAF for Middle East and Africa, OECD the countries part of the Organisation for Economic Co-operation and Development (OECD), and REF for Reforming economies that were part of the URSS. These five regions correspond to the so-called SSP regions, which have been widely used in studies involving climate change and socioeconomic development

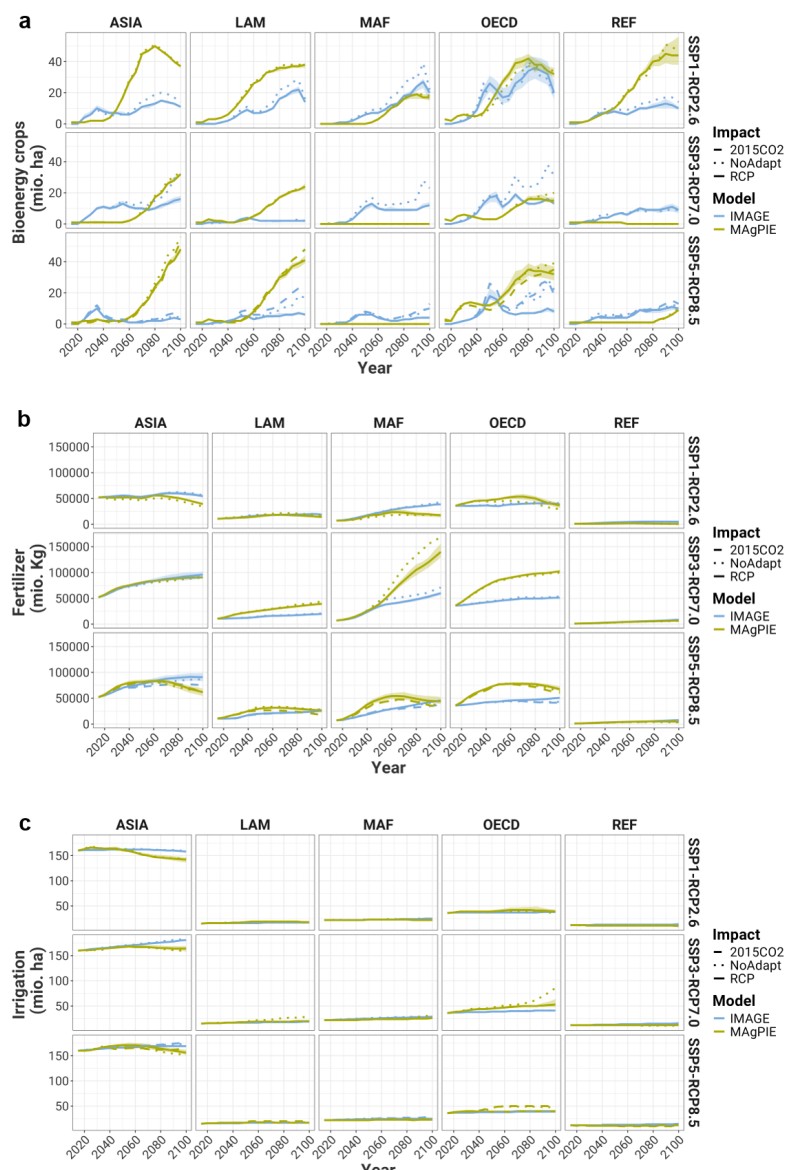

**Figure B2. Regional projections of land-use management related variables from different land-use models and for different climate and human forcings.** a) Shows second-generation bioenergy crops in units of million hectares, b) nitrogen fertilizer use in million kilograms, and c) irrigated crop area in units of million hectares. The lines in green and blue correspond to the average of the projections of each LUM, based on impact data derived from five GCMs under the scenario under consideration for the three SSPx-RCPy climate-human forcings (SSP1-RCP2.6, SSP3-RCP7.0, and SSP5-RCP8.5). The ribbon represents the upper and lower projections per LUM of the five GCMs-based impact data. The dashed line represents the counterfactual scenario where no climate impact is considered (SSPx-NoAdapt), and the dotted line is a scenario where $CO_2$ fertilization is not included (SSP5-2015CO2) in the yield projections used by the LUMs (only available for SSP5-RCP8.5)

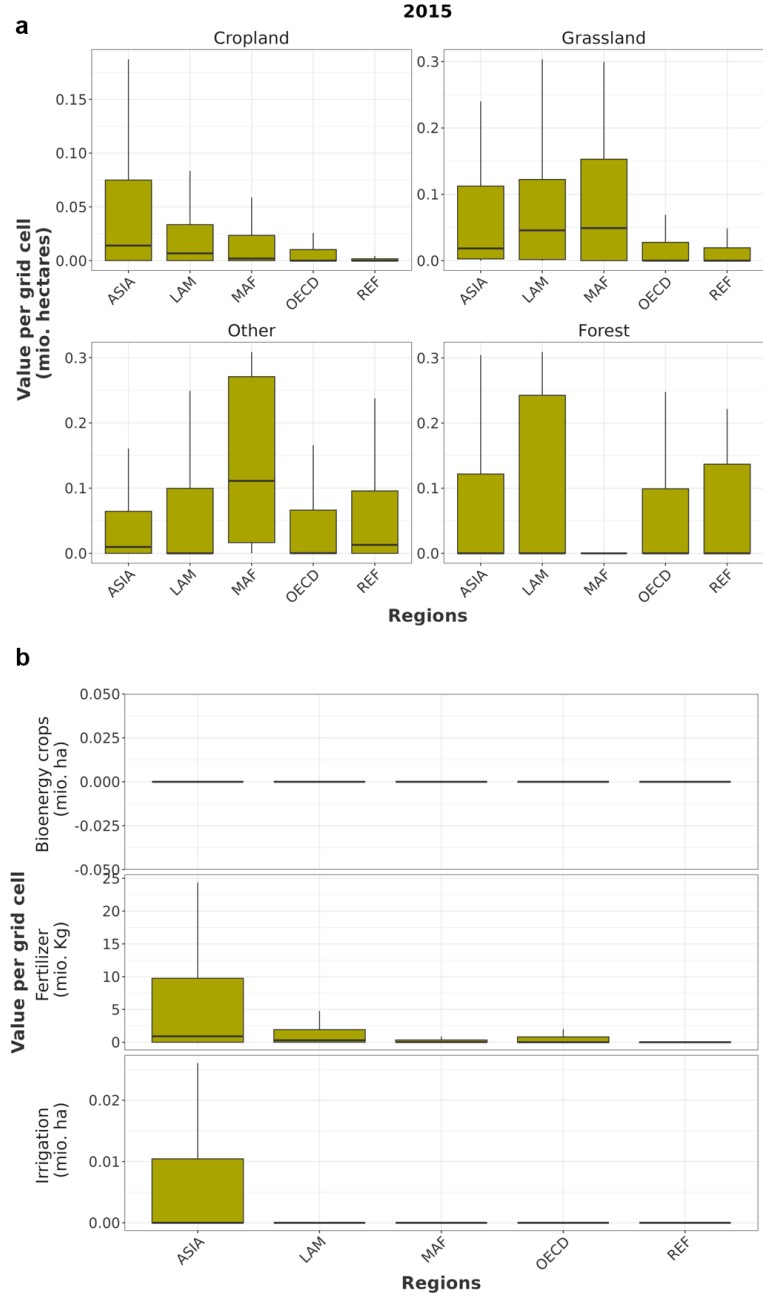

**Figure B3. Boxplot representation of grouped cells per region and variable in 2015** a) Displays the distribution of the land-use types and b) that of 2nd generation bioenergy crop area, synthetic nitrogen fertilizer use, and irrigated cropland. In the boxplots, the thicker horizontal line (usually close to the middle of the box) represents the median, the upper and lower sides of the box, the upper and lower quartiles, respectively, and the top of the vertical lines, the upper quartile plus 1.5 the interquartile range

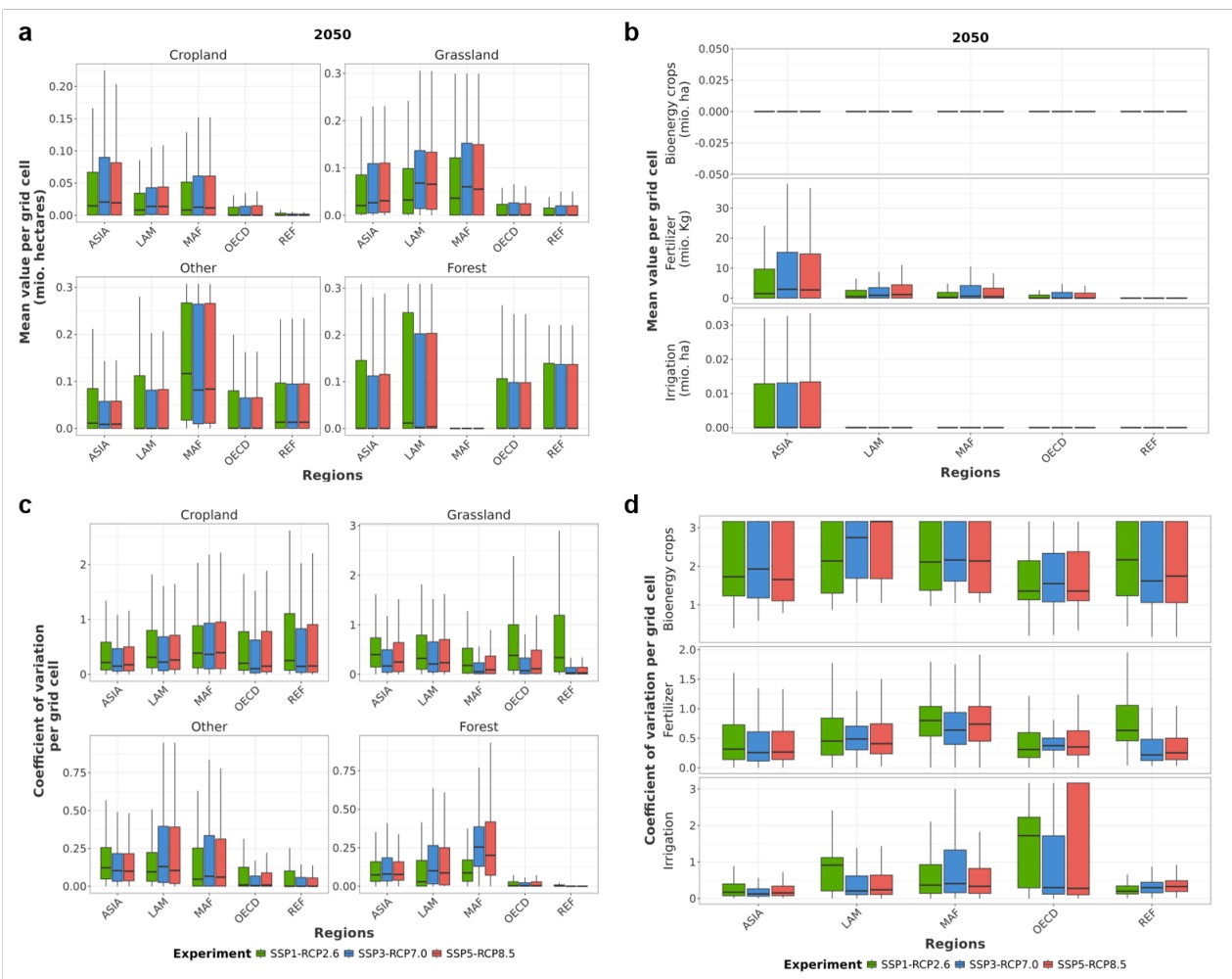

**Figure B4. Boxplot representation of grouped cells per region, variable, and SSP*x*-RCP*y* in 2050** On the one hand, a) Displays the distribution of average land-use types area per grid cell and b) that of 2nd generation bioenergy crop area, synthetic nitrogen fertilizer use, and irrigated cropland. On the other hand, c) shows the distribution of the coefficient of variation of land-use types area per grid cell calculated based on ten simulations (2 land-use models x impact data based on five global climate models) and d) that of 2nd generation bioenergy crop area, synthetic nitrogen fertilizer use, and irrigated cropland. In the boxplots, the thicker horizontal line (usually close to the middle of the box) represents the median, the upper and lower sides of the box, the upper and lower quartiles, respectively, the upper extreme of the vertical lines on the upper side of the box, the upper quartile plus 1.5 the interquartile range, while the lower extreme of the vertical lines on the bottom side of the box, the lower quartile minus 1.5 the interquartile range

**2015**

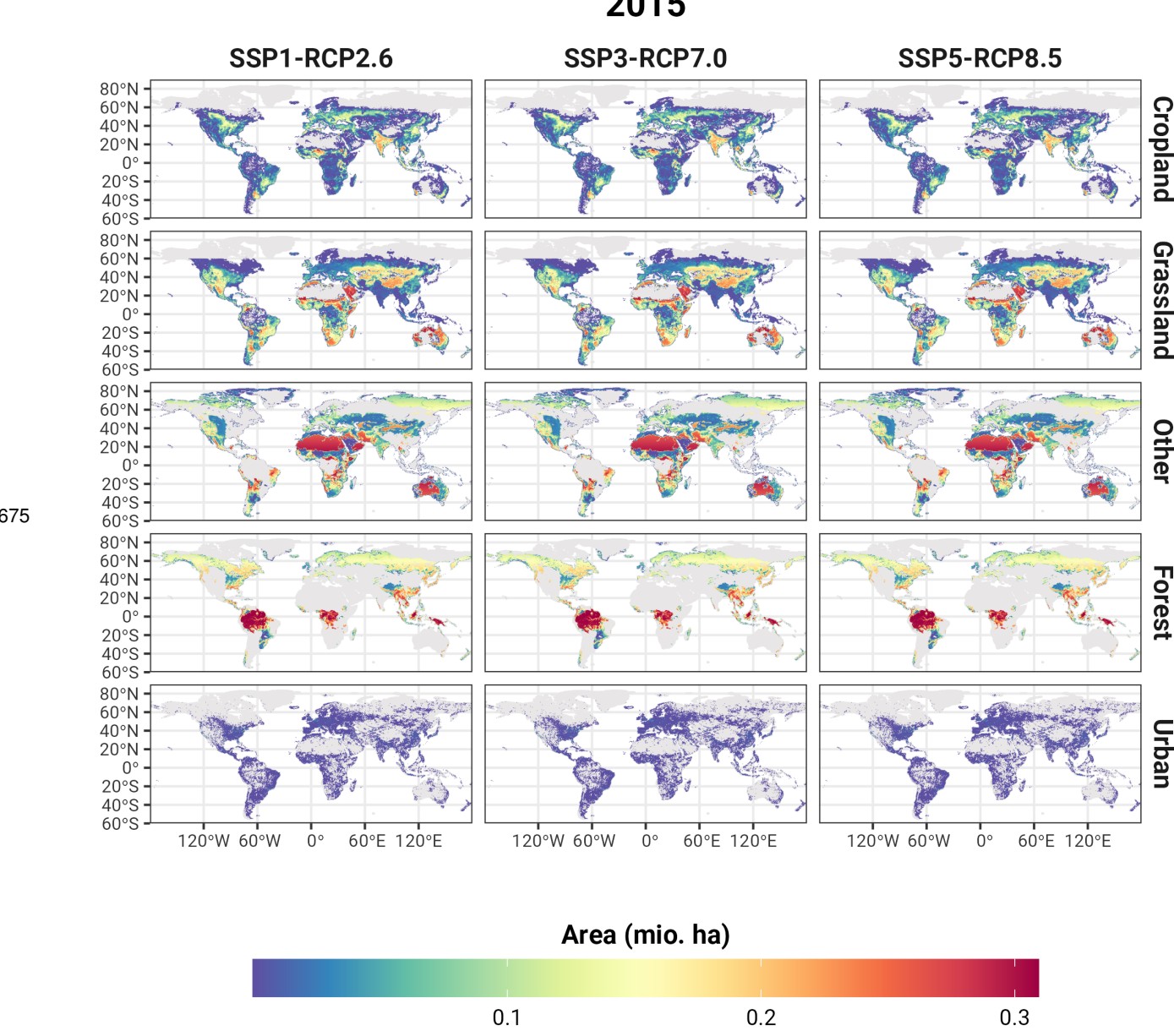

**Figure B5. Historical Land-use map (2015) to which the LUMs projections were harmonized**

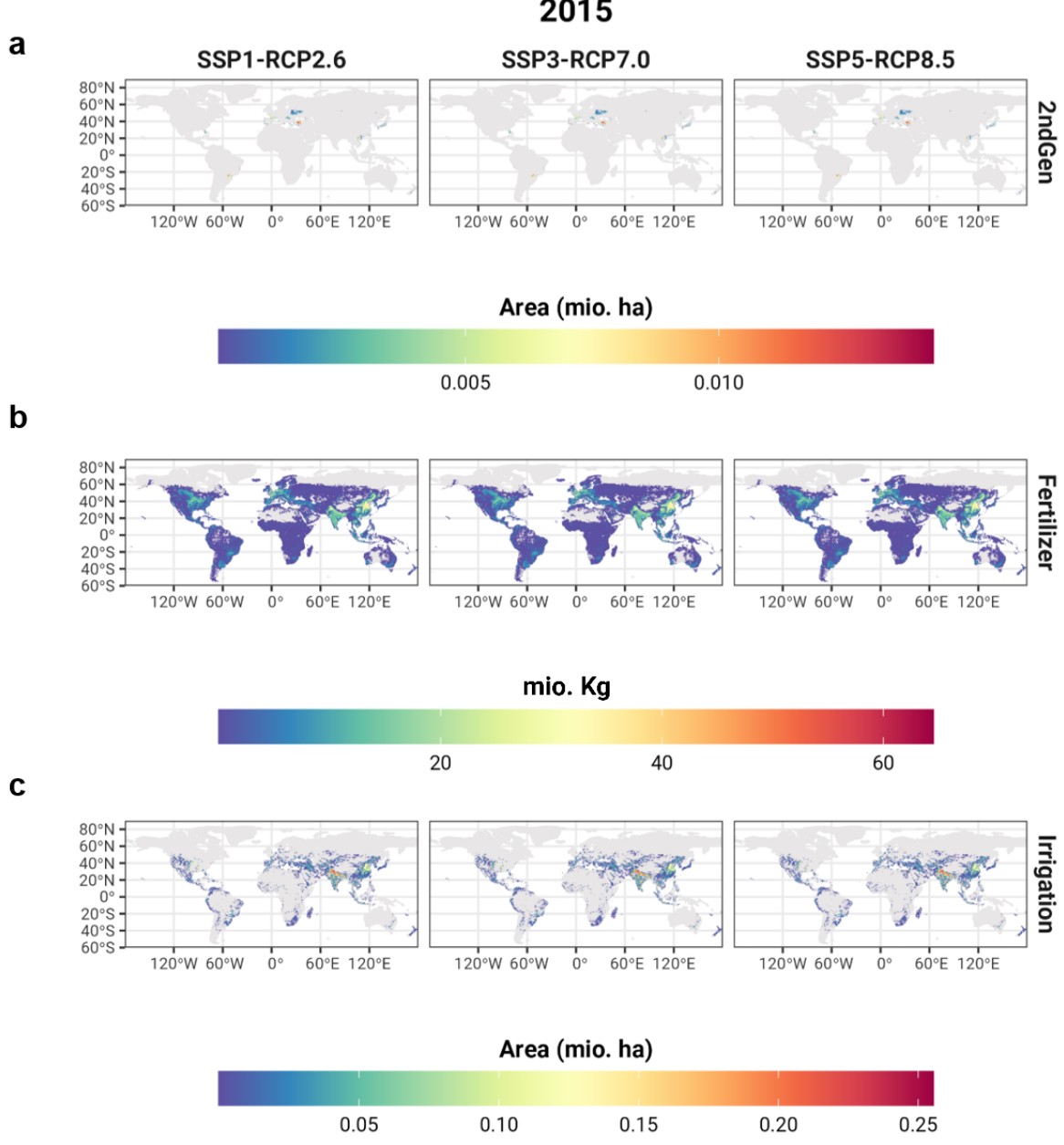

**Figure B6. Historical second generation, synthetic nitrogen fertilizer use, and irrigated cropland areas (2015) to which the LUMs projections were harmonized**

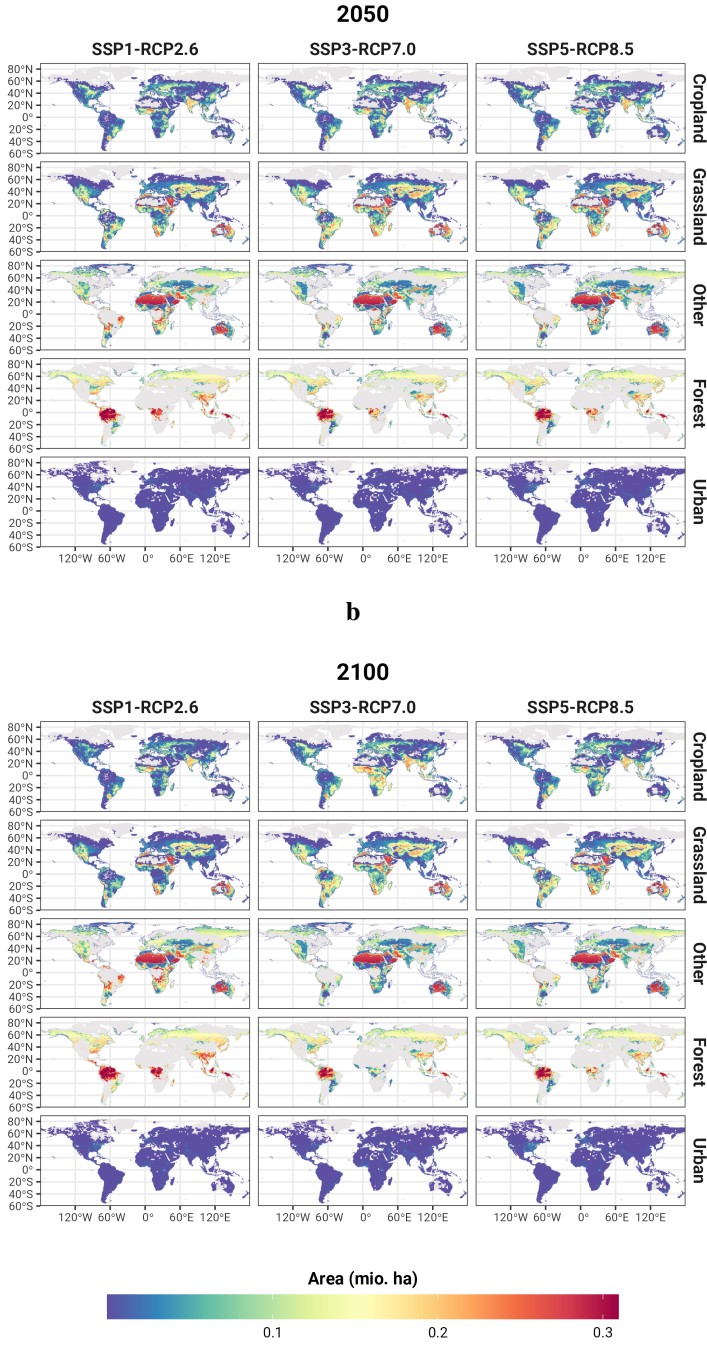

**Figure B7. Grid-level average of Land-use types for the LUMs-GCM ensemble under three socioeconomic and climate scenarios**. a) Depicts the year 2050, and b) the year 2100

**a**

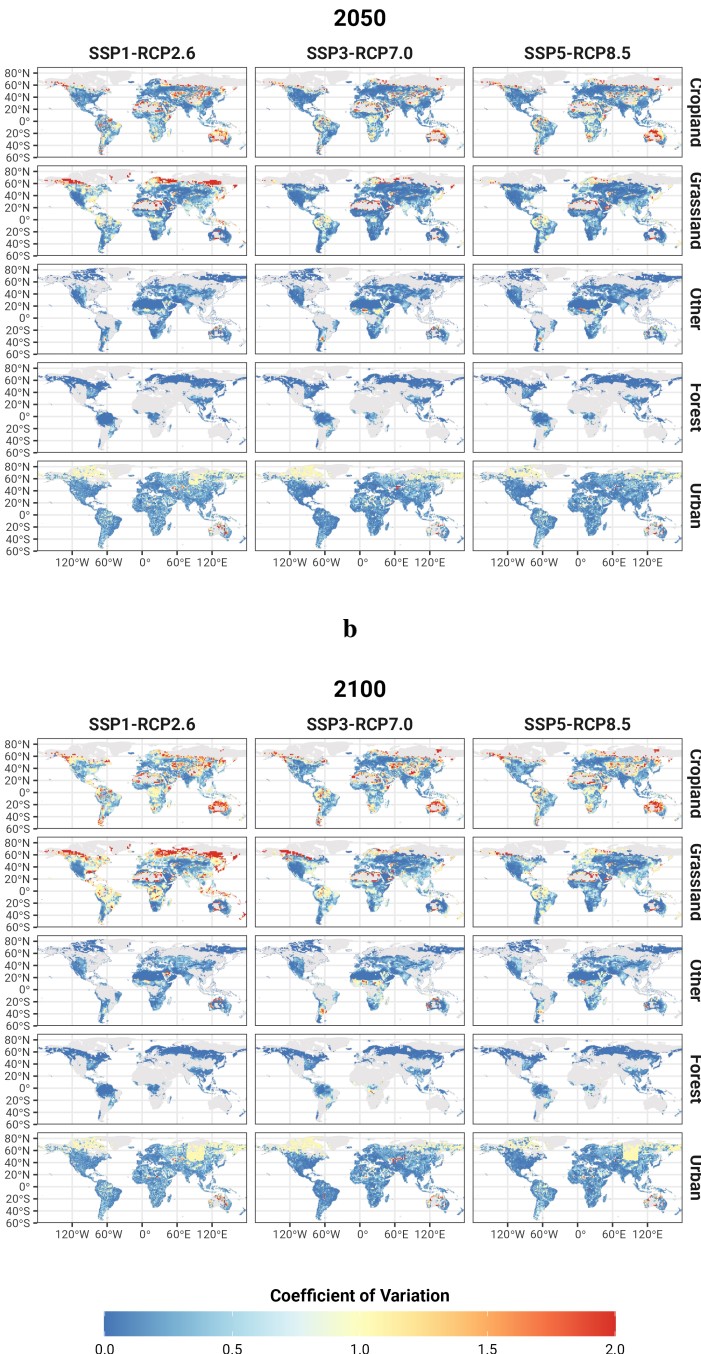

**Figure B8. Grid-level coefficient of variation of the different land-use types for the LUMs-GCM ensemble under three socioeconomic and climate scenarios**. a) Depicts the year 2050, and b) the year 2100

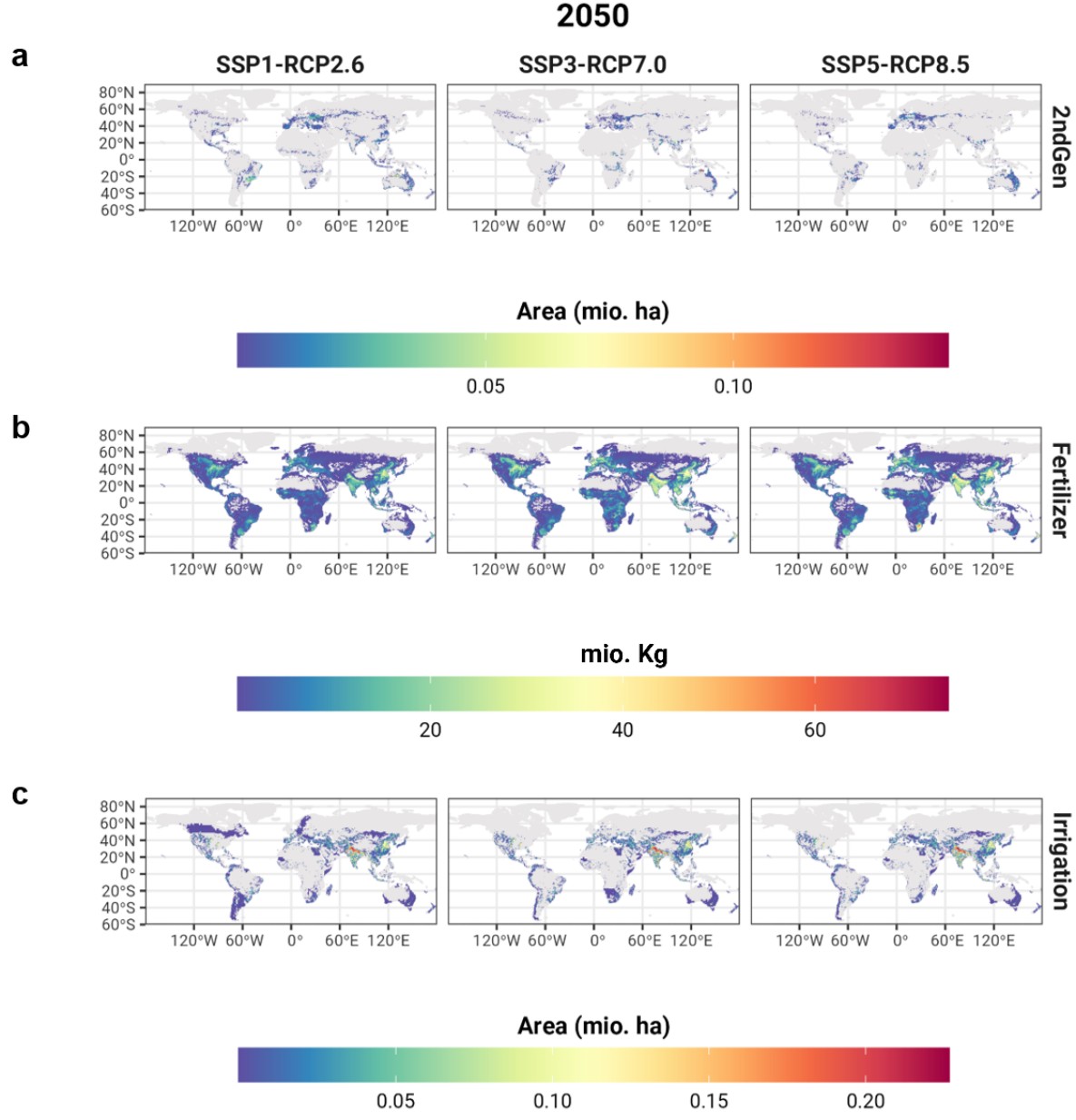

**Figure B9. Grid-level average of agricultural management variables for the LUMs-GCM ensemble under three socioeconomic and climate scenarios in 2050**

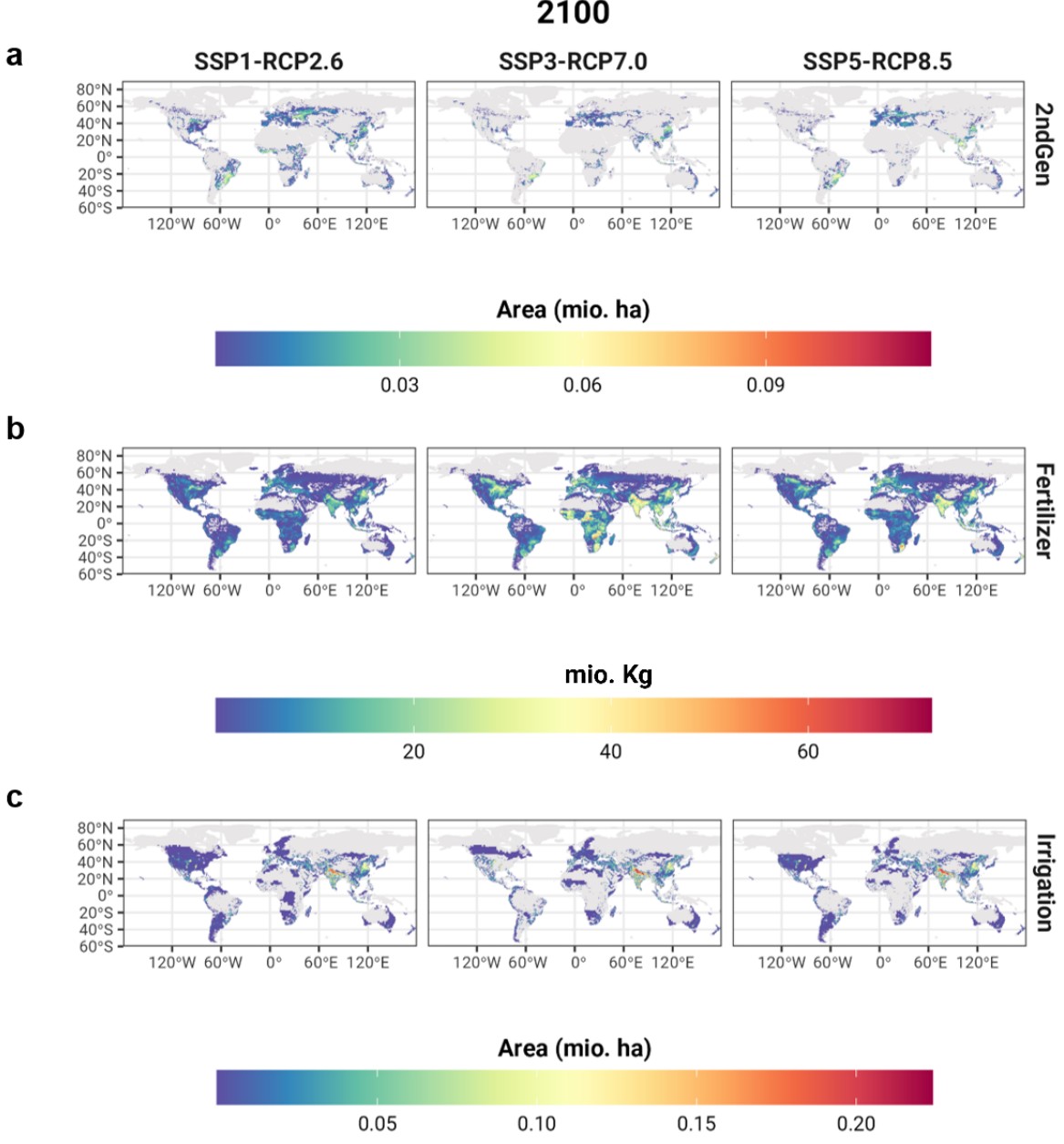

**Figure B10. Grid-level average of agricultural management variables for the LUMs-GCM ensemble under three socioeconomic and climate scenarios in 2100**

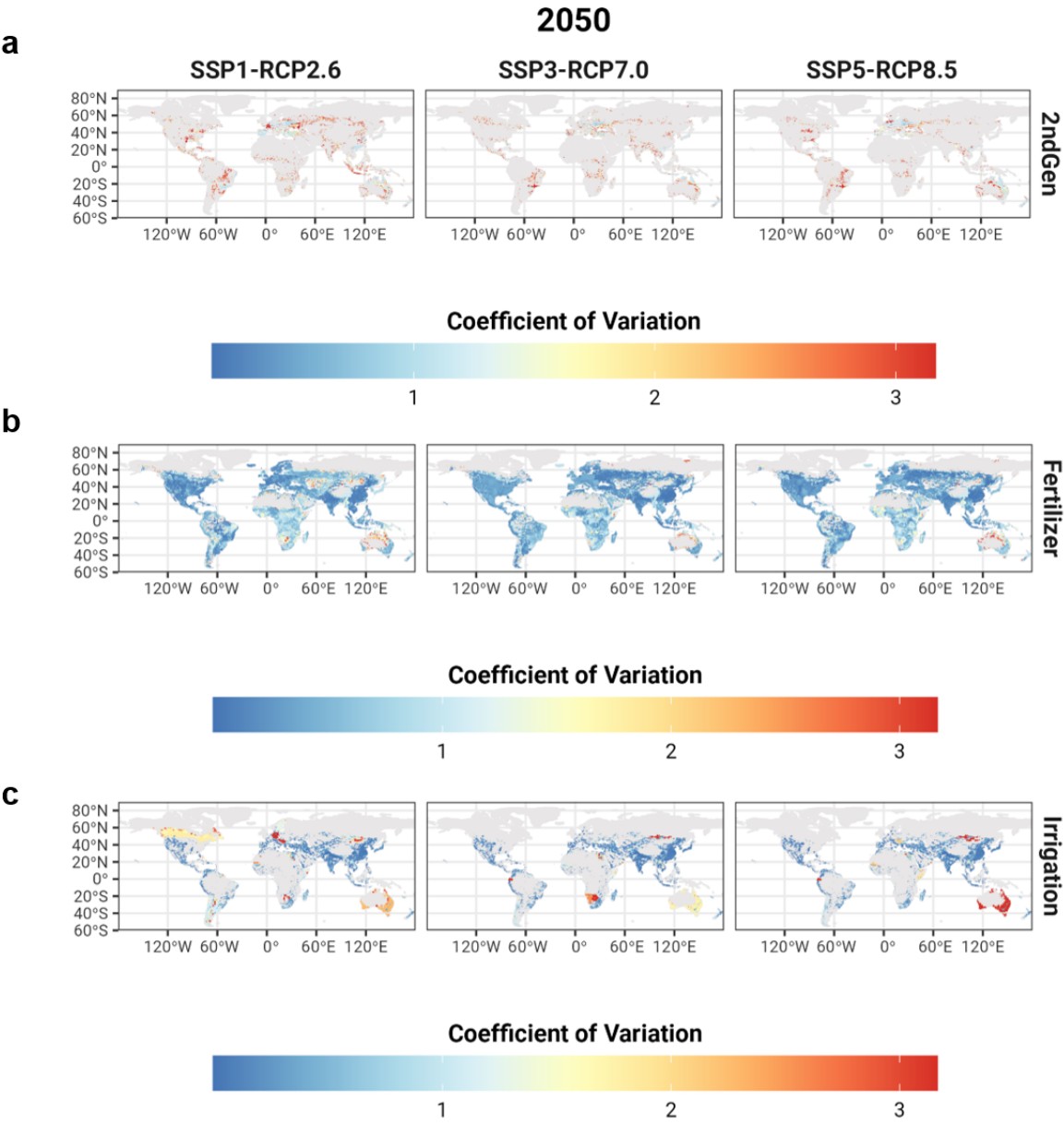

**Figure B11. Grid-level coefficient of variation of agricultural management variables for the LUMs-GCM ensemble under three socioeconomic and climate scenarios in 2050**

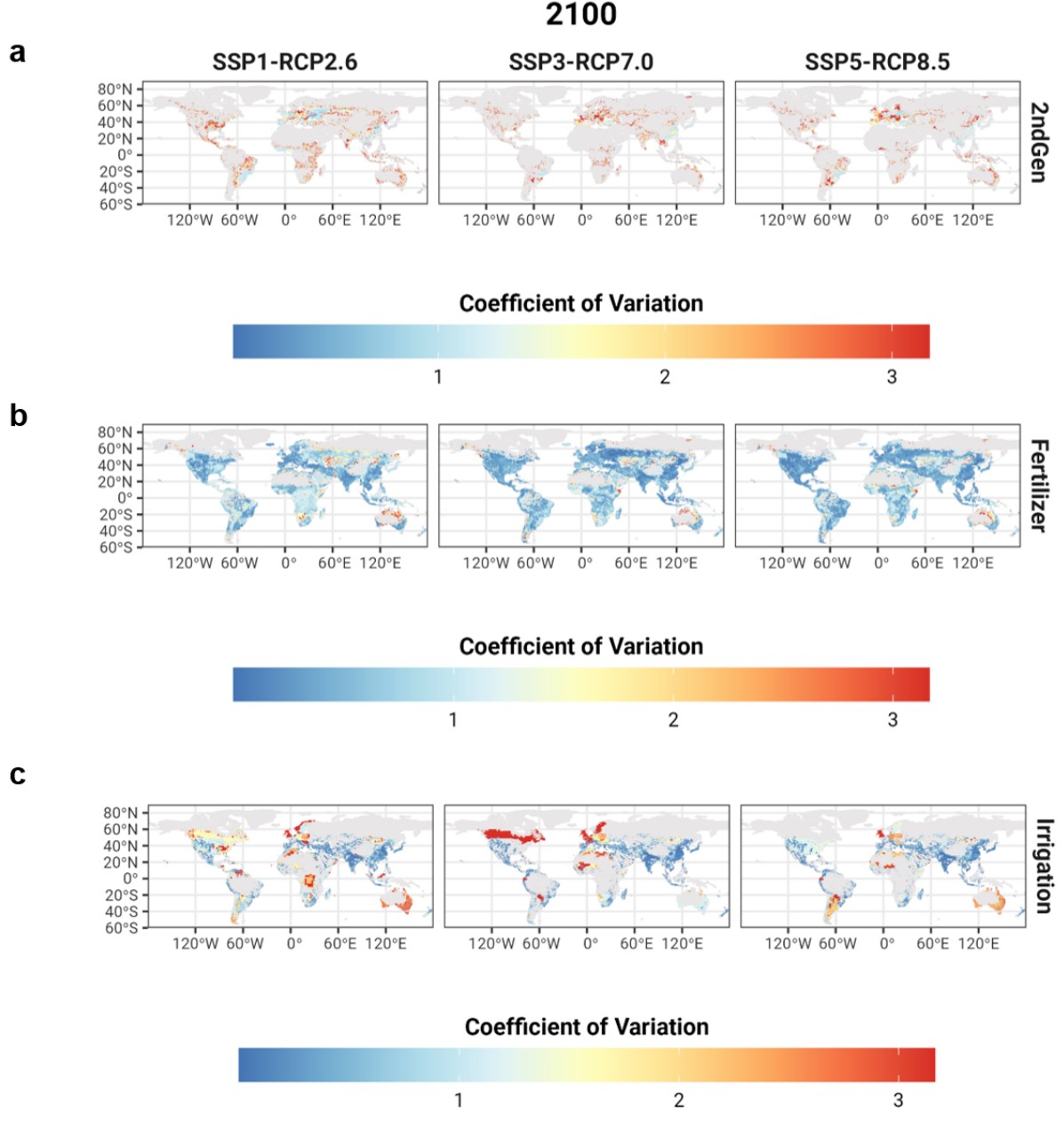

**Figure B12. Grid-level coefficient of variation of agricultural management variables for the LUMs-GCM ensemble under three socioeconomic and climate scenarios in 2100**

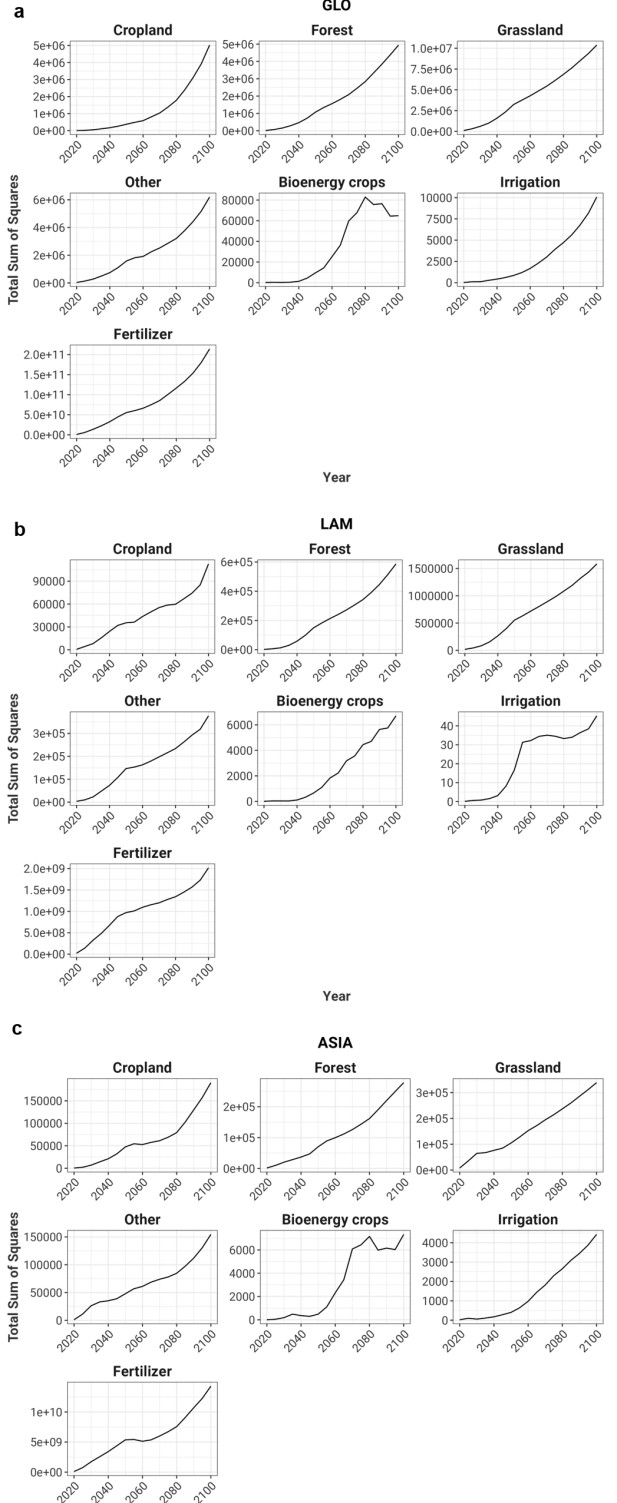

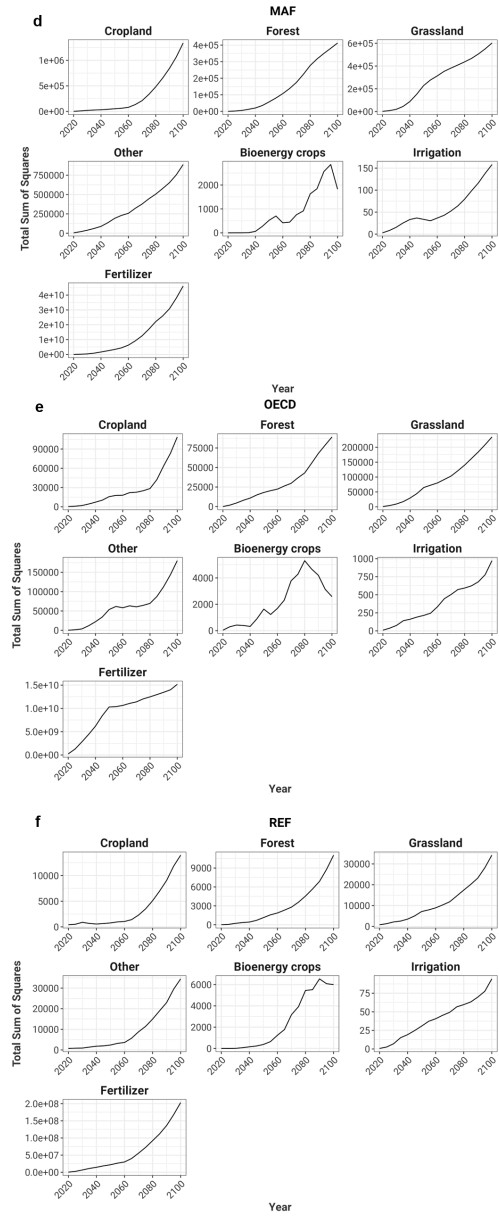

**Figure B13. Total regional variance based on four factors represented as the total sum of squares**

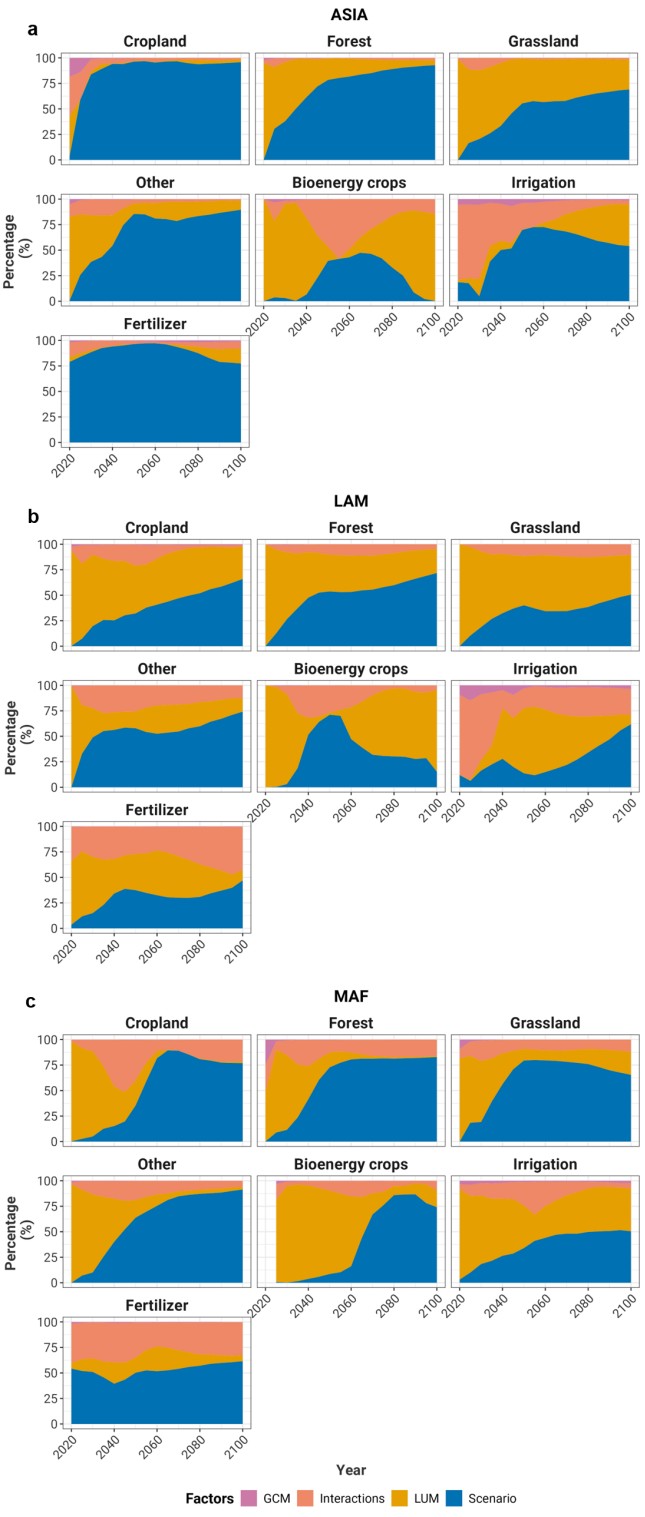

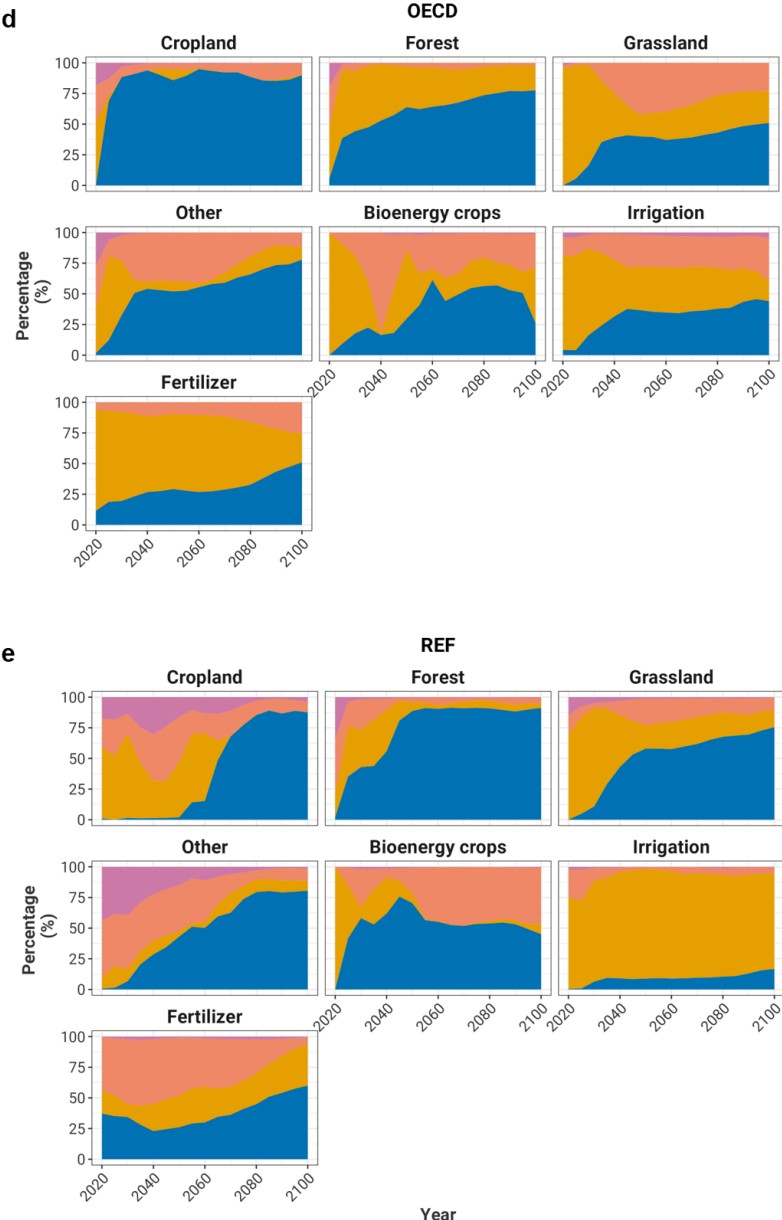

**Figure B14. Fraction of variance explained by the specific factors for the harmonized regional land-use and land-use management projections.** GCM stands for the global climate models used to generate the climate impact inputs used by the Land-Use Models (LUMs). Scenario relates to the different SSP*x*-RCP*y*. Finally, the Interactions factor refers to the residual, assumed here as the interactions between the different factors

# 2050

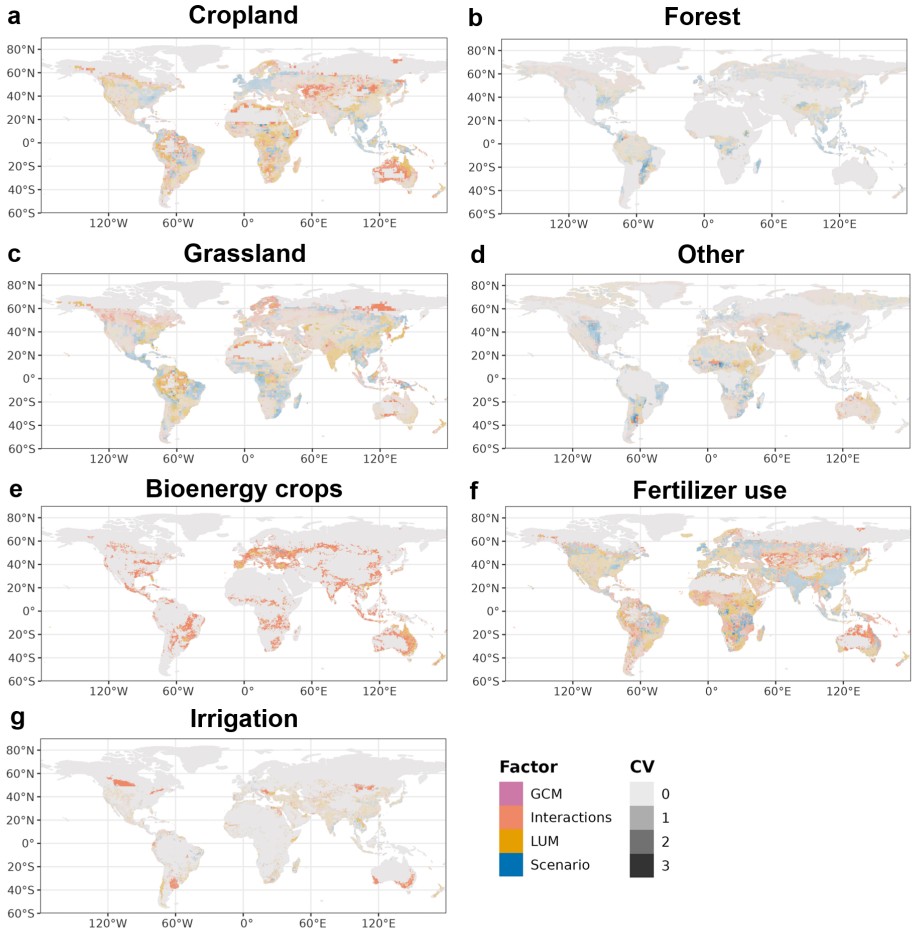

**Figure B15. Highest fraction of variance explained by the specific factors for the harmonized spatially explicit land-use and land-use management projections in 2050.** GCM stands for the global climate models used to generate the climate impact inputs used by the Land-Use Models (LUMs). Scenario relates to the different SSP*x*-RCP*y*. Finally, the Interactions factor refers to the residual, assumed here as the interactions between the different factors. In the maps, the color represents the factor (LUMs, GCMs, Scenario, and Interactions) that explains the highest share of the variance in each cell, and the opacity (lower values correspond to more transparent colors) depicts the coefficient of variance of each cell calculated based on 30 simulations (two LUMs x five GCMs x three SSP*x*-RCP*y*)

## Appendix C: Additional concepts and methods

### C1   ISIMIP

The Inter-Sectoral Impact Model Intercomparison Project (ISIMIP) provides harmonized input data and protocols for cross-sectoral global and regional climate impact model comparisons. Its primary objective is to add to the understanding of climate change impacts at different levels of warming across a wide range of sectors and impact models to assess model structural and input data uncertainties. ISIMIP aims to evaluate climate change's historical, current, and future effects on natural and human systems and to make impact data comparable across models and sectors (Rosenzweig et al., 2017).

Specifically, ISIMIP provides consistent climate and socioeconomic forcing data sets generated within established sectors and protocols, adhering to standardized formats, scales, and configurations. The collected data is openly accessible through a portal (https://data.isimip.org/). ISIMIP operates in a series of iterative rounds linked with the Coupled Model Intercomparison Project (CMIP) phases. The ISIMIP3b phase focuses on future projections (group III simulations) to examine future changes resulting from direct human influences across different sectors and climate change. Different land-use modeling teams contributed with projections following a joint set of assumptions and scenarios to provide future land-use projections from several LUMs as input for these ISIMIP3b group III simulations. The reported variables include cropland, forest, grasslands, natural vegetation, urban area, and their respective subtypes, which are relevant for all climate impact models that cover land-use dynamics, such as agricultural or land surface models. Additionally, the LUMs provided data on the distribution of bioenergy crops (second generation), irrigated crop areas, fertilizer use rates, and wood harvest, among others. These harmonized projections cover the period between 2015 and 2100 and are reported at a resolution of $0.5° \times 0.5°$. This study focuses on the four land-use types, second-generation bioenergy cropland area, irrigation, and nitrogen fertilizer use.

### C2   LUH2 harmonization

In the first step of the harmonization, the land-use data from the LUMs was standardized to a consistent spatial resolution of $0.25° \times 0.25°$ and interpolated to annual time steps in case the LUMs report at different resolutions and formatted as fractional patterns. The management data was also aggregated to national totals and converted to standard units. For the harmonization, the land-use data was then aggregated to a resolution of $2° \times 2°$ since the historical data and future projections had more consistency at this resolution, and it is also a common spatial resolution used for the land surface in climate models participating in CMIP6. Afterward, annual changes were computed from the LUMs' patterns and sequentially applied to the patterns from the previous time step, starting with the last year of the historical data set. This process was specifically carried out for cropland, grassland (pastures and rangelands), and urban land projections. The resulting harmonized patterns were then converted to the $0.25° \times 0.25°$ original resolution. Following this, the cropland and grassland were divided into five different crop functional types, as well as pastures and rangelands. Forests and other natural vegetation were later calculated as the remaining surface area not used for cropland, grazing, or urban areas. Further disaggregation into forests and other natural vegetation was based on LUH2's map of potentially forested areas, which was based on an empirically based ecosystem model and a climatology dataset. As the next step, similar to the land-use patterns, annual changes in the LUM's management data were calculated

and applied to the previous year's management gridded data, including irrigated areas, fertilizer inputs, and second-generation bioenergy crop areas. Bioenergy crop areas are harmonized separately from the cropland areas. Annual changes were calculated at the country level and applied to the corresponding grid cells within each country based on a pre-established mapping along with gridded data provided by the future projections. Detailed information on the harmonization and historical reconstruction of land-use and management patterns can be found in Hurtt et al. (2020).

## Appendix D: Additional analyses

### D1 Counterfacturals comparison

Global and regional projections of the counterfactuals without climate impacts (SSPx-NoAdapt) and without $CO_2$ fertilization (SSP5-2015$CO_2$), made as sensitivity analyses, show larger cropland areas than those with climate impacts for both LUMs. This effect particularly increases as emissions rise (SSP3-7.0 and SSP5-8.5) and aligns with modeling studies (Jägermeyr et al., 2021; Molina Bacca et al., 2023), supported by experimental evidence (Toreti et al., 2020), that show that introducing the $CO_2$ fertilization process to the global gridded crop models (GGCMs) could positively affect yields in some crops leading to lower future cropland. However, most GGCMs barely consider negative effects due to the redistribution of pests and diseases or compound climate effects, although there is ongoing work towards their inclusion, which could add additional local stresses to crop production (Fu et al., 2023; Jägermeyr et al., 2021).

On the global scale, regarding the management variables without the effect of climate change and without dynamic $CO_2$ fertilization effects on crop yields, we see slightly larger areas of second-generation bioenergy crops, with a larger effect in IMAGE, than the scenarios including impacts. This holds for all socioeconomic-NoAdapt scenarios. For irrigation, the scenarios without impacts (SSPx-NoAdapt), especially in SSP3-NoAdapt for MAgPIE, lead to considerably higher irrigated cropland demand towards the end of the century compared to SSP3-RCP7.0. Finally, fertilizer use in SSP1-NoAdapt and SSP5-NoAdapt compared to their counterparts, including climate change impacts, show little to no difference, while SSP3-NoAdapt is higher than SSP3-RCP7.0 due to considerably larger cropland areas in the scenario without impacts.

*Author contributions.* EMB drafted the manuscript supported by MS. EMB, MS, BLB, HLC, KF, CR, and AP designed the study. EMB, MS, JCD, and EST generated the raw data. Harmonization was done and based on the work of LPC and GH. Relevant core development of the models was performed by EMB, MS, JCD, ESF, FH, KK, and JPD. Scripts used for data manipulation, plotting, and analysis were done by EMB and FH. Data analysis was done by EMB, LPC, and JV. Biophysical impact data was generated based on CM's and JH's work. Collaboration interactions were led by KF and CR. All authors contributed to discussing results, writing, and/or reviewing the paper.

*Competing interests.* None of the authors reports competing interests

*Acknowledgements.* FH received funding from the European Union's Horizon Europe research and innovation program, project Wet Horizons (Grant No. 101056848). LPC and GH are founded by the NASA Carbon Monitoring System program (80NSSC21K1059). In general, this work has been supported by the COST Action CA19139 PROCLIAS (PROcess-based models for CLimate Impact Attribution across Sectors), funded by COST (European Cooperation in Science and Technology, https://www.cost.eu).

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
