# Peer review of "Future land-use pattern projections and their differences within the ISIMIP3b framework"

_EGUsphere, 2024_

## Author Comment (AC1)

| Comment | Response |
|---|---|
| **Line 46-48. Include more recent studies to support the frameworks and models to project and compare future land-use and land-use-related variables.** | In Lines 46 and 48, we referred to the most representative papers, which concerned land-use dynamics and intercomparison of multi-model outputs. However, we agree that additional papers are needed to provide more context and resources. Considering this, we will add newer intercomparison papers in lines 46-48 to tackle this point. The updated 46-48 lines would look like this:

"For this purpose, various frameworks and models have been utilized to project and compare future land use and food system-related variables focusing on crop and livestock production, food prices, use of resources, changes in land-use areas, among others under different scenarios (Sörgel et al., 2024; Weindl et al., 2024; Doelman et al., 2022; Rose et al., 2022; Lèclere et al.,2020; Hasegawa et al., 2018; Popp et al., 2017; Nelson et al., 2014; Popp et al., 2014b)" |
| **Lines 61-75 should be part of the methods section.** | We will summarize the information and leave the details in the methods section. |
| **Line 100. Not clear how the demand for bioenergy production aligned with climate policies was determined.** | For MAgPIE, REMIND provides information on GHG pricing and the demand for second-generation bioenergy crops (lignocellulosic feedstocks). REMIND determines this demand by considering the supply, trade, and conversion of biomass feedstocks through the value chain, while accounting for energy sector market conditions and regulatory frameworks in each socioeconomic growth scenario (as detailed by Merfort et al., 2023). Since these scenarios are aligned with specific climate change pathways, bioenergy demand is intrinsically linked to emissions budgets and carbon taxes required to achieve particular warming targets.

In IMAGE, the TIMER energy model defines bioenergy demand based on land supply, biomass productivity, input costs, and learning dynamics, which together influence bioenergy prices. Climate policies in the IMAGE framework are designed to meet long-term climate targets by establishing global emission pathways. These pathways determine carbon tax prices and mitigation costs, which, in turn, affect bioenergy prices and demand (as detailed in Doelman et al., 2018).

This explanation, including references, will be added to Table A1 in Appendix A. |
| **Given that the study's purpose was to identify uncertainties, the authors should discuss more in detail the implications of these models when used, particularly for policy making. More examples like the one presented in lines 454-456 are missing.** | Although the data and the study described in the manuscript were initially intended to inform impact modelers using our data as human forcing inputs within the ISIMIP context, this comment has brought to our attention that the data will be publicly available and could be utilized by users from various disciplines and sectors. In response to this, we have decided to add the following paragraphs to the discussion:

"Additionally, differences in land-use projections are expected to directly affect the impact models that use this data as input. For instance, grasslands are among the ecosystems with the highest wildfire frequencies (Donovan et al., 2017). Therefore, uncertainty in LUMs×GCMs grassland projections could influence the identification of fire hotspots due to human-induced effects (Thompson and Calkin, 2011). Uncertainty propagation stemming from land-use patterns could also impact, e.g., the calculation of emissions from land-use transitions (Neuendorf et al., 2021), shifts in biomes (Alexander et al., 2017), the assessment of ecosystem services, habitat intactness, and biodiversity (Yang et al., 2024), among others."

"The uncertainties observed in land-use variables at different resolutions arise from error propagation throughout the modeling workflow, as well as from scenario modeling approaches and other factors. These uncertainties highlight the need for conscientious use of the reported data, carefully considering its limitations. The objective of the data is to provide a global overview of land and agricultural systems and their development under a set of socioeconomic and climate scenarios based on different assumptions.In the context of policy and management decision-making, the data presented here should be seen as an overview of global trends. However, it is not intended to replace targeted assessments and actions specific to, e.g., country, local, or regional levels that include contextual knowledge—including input from communities and experts— that should be incorporated during the assessment and planning phases to ensure that proposed actions align with actual needs in the policy-making context (Neuendorf et al., 2021)." |

| Authors provide a rich set of results, however a summary key messages across land uses and global regions are missing, that is, messages that contextualise the value of the findings for decision making based on modelling outputs. This could be done in the abstract or in a conclusions section | We agree with the reviewer that the manuscript contains a substantial amount of information, and we acknowledge that both the text and the reader would benefit from a summary of the key messages. To address this, we will include summaries of the key messages in the suggested locations and at the end of each section. |
|---|---|

---

## Author Comment (AC2)

| Major comments | |
|---|---|
| **Comment** | **Response** |
| **Clarification of Figures 7 and 8: Could the author clarify why the gridded differences are primarily influenced by the specific factors highlighted in Figures 7 and 8, rather than other potential variables? On Lines 438–441, the manuscript states: "One of the primary explanations for the effect of harmonization on forests is the different inputs regarding forests among the LUMs and LUH2 historical maps used in harmonization, especially in areas with intermediate tree cover. For example, global forest areas in 2000 range among different satellite sources and FAO between 3600 and 4300 million hectares (Ma et al., 2020)." While this provides some explanations for the effect of harmonization on forests, more detailed explanations of the dominance of the factors shown in Figures 7 and 8 would be helpful for better understanding.** | Given the level of detail reported by the land-use models before and after harmonization, we focus on the factors corresponding to the categorical variables available. Our central assumption is that the primary sources of variance stem from (1) uncertainties in the GCMs used to generate the impact data, (2) differences in the processes, inputs, and modeling approaches of the various land-use models, and (3) assumptions underlying the scenario representations. Incorporating additional variables would require re-running the models and conducting further tests. However, as the primary aim of this study is to highlight the sources of uncertainty in the data presented rather than to comprehensively analyze differences among land-use models, such tests fall outside the scope of this work.

We will clarify this in subsection 2.4.3 (Methods) and subsection 3.3.2 (Variance analysis on the global scale), referring to the studies by Schmitz et al. (2014) and Nelson et al. (2014), which delve more deeply into differences among land-use models.

However, to further expand on the discussion in section 3.3.2 regarding differences among land-use models and factors—including inputs, definitions, and processes—we will add a supplementary table in Annex A. This table will detail the land-use variables explored in this study, clarifying their input sources, definitions, and calculation methods, for the two models and their treatment during harmonization. This addition will provide further transparency and context for our analysis. |
| **Comparison of ISIMIP3b and LUH2 Datasets: The manuscript compares the ISIMIP3b LUC with CMIP6 LUH2 data in various instances, such as on Lines 287–289: "This drop in demand for second-generation bioenergy crops is related to changes in the mitigation assumptions of SSP1-RCP2.6, which involves updated impacts on yields." This is informative, but could the authors provide a more detailed explanation of the core differences between the ISIMIP3b and LUH2 datasets, and explain how these fundamental differences contribute to the observed discrepancies? This additional context would help the reader better understand the significance of ISIMIP3b LUC and understand why it differs from CMIP6 LUH2.** | The following explanation will be added to the Methods section to clarify the differences between the LUH-CMIP6 and the ISIMIP3b datasets generated by the Land Use Models (LUMs).

LUH2-CMIP6 data differs from the ISIMIP3b data in that LUH2-CMIP6 does not account for $CO_2$ fertilization. Additionally, LUH2-CMIP6 combines outputs from multiple land-use models for different scenarios, introducing variability in dynamics based on the models used. Another key difference lies in the inputs of the LUH2 harmonization algorithm, as the historical datasets used in ISIMIP3b have been updated compared to those in LUH2-CMIP6. Additionally, a new representation of protected lands to better match the IAM assumptions was included. There are also notable differences in the versions of the models employed. For MAgPIE, the version used for CMIP6 simulations was 3.0, while ISIMIP3b utilized version 4.4.0. The latter (starting from MAgPIE 4.0) introduces several enhancements, most notably, a food demand model that accounts for detailed dietary composition, food waste, and demographic characteristics. MAgPIE's used version used in this study also improves spatially explicit outputs by incorporating the accounting of capital stocks and their depreciation, and a more detailed representation of the forestry sector.
Similarly, for IMAGE, the version used for ISIMIP3b was 3.3, whereas version 3.0 was used for LUH-CMIP6. IMAGE 3.3 includes more crop categories, advancements in modeling bioenergy, deforestation, land-based mitigation, and improvements in water use modeling. |

| | |
|---|---|
| **Explanation of Equation 2: Please explain how interaction is defined and how the interaction calculation is conducted.** | The residual term—"interactions" in Equation 2 for this study—represents the portion of variance the independent variables (GCMs, RCPs, LUMs) cannot explain. This interpretation, where residuals are equivalent to the interactions, is particular to this type of study due to the deterministic nature of our data (the LUM models are deterministic). Since the factors and total variance (as a sum of squares) can be derived from the data, the difference between the total variance and the variance explained by the factors reflects the effect of the residuals or interactions. This component captures the non-additive or nonlinear contributions to the variance.

 If the residual/interactions term is significant compared to the other factors, it highlights the complexity of the relationships between the dependent and independent variables. Equation 2 simplifies highly complex systems, spanning climate, crop, energy, and land-use models, as the workflow diagram shows. Therefore, a significant contribution from the interactions term highlights the varying sensitivities and complexity of modeling different land-use variables and the effect that climate impacts and socioeconomic growth assumptions have on them.

 We will add this clarification to the Methods section (2.4.3 Variance analysis) and in the results of the variance analysis in section 3.3 (Variance analyses). |
| **Uncertainty from Land Use Downscaling: The land-use downscaling process could introduce uncertainty into the gridded LUC. I suggest the authors could discuss this uncertainty in the discussion section** | We will add more information about how models disaggregate their data to spatially explicit levels and sources where detailed information can be found in the models' description in the methods section (subsection 2.1) and possible implications in section 4 (Conclusions and Discussion).

 **In methods:**
 Disaggregation of land-use patterns in IMAGE relies on gridded potential yields from LPJmL, data from the previous step, a regional management factor, and an empirical allocation algorithm. The process begins with calculating potential cropland and crop production data in the current time step using the patterns from the previous step. If production is insufficient to meet demand, less productive areas are abandoned, whereas cropland expansion employs the empirical allocation algorithm that evaluates cropland and grassland allocation. More information is available in (Doelman et al., 2018).
 In MAgPIE, land-use disaggregation is based on the previous step patterns, available cropland, and a mapping between the high and low resolutions. At each time step, starting with cropland, changes in land use from the clusters are disaggregated using expansion and reduction weights and information about land availability. Detailed information can be found in the interpolateAvlCroplandWeighted function from the R library Lucode developed by the MAgPIE team (Dietrich et al., 2024).

 **In the discussion section:**
 Disaggregation of LUMs outputs to high-resolution levels plays a critical role in determining spatially explicit land-use outputs in LUMs and could contribute to uncertainty. However, during the harmonization process, the original gridded data reported by the LUMs is aggregated to a 2°×2° resolution and subsequently disaggregated using the approach described in Hurtt et al. (2020). Adopting a consistent algorithm in this step could help minimize allocation differences in the higher-resolution harmonized data due to downscaling. |

| Minter comments | |
|---|---|
| **Comment** | **Response** |
| **Lines 447-448: "However, we found some differences regarding the regional and local distribution of land-use change, specifically in cropland for the LAM region." Please explain why this difference in cropland occurs.** | We would expand this sentence as: "However, we found some differences regarding the regional and local distribution of land-use change, specifically in cropland for the LAM region, due to a higher demand for bioenergy crops in this area in MAgPIE, as it can be seen in Figure B3". Further context to differences in bioenergy demand will be added as suggested for the Minor comment regarding differences in allocation and demand for bioenergy cropland area below. |
| **Lines 69-70: "... which has commonly been used for impact analyses in global and regional studies. (Yu et al., 2019; Qiu et al., 2023; Hoffmann et al., 2023)." Please check if the period before the parentheses needs to be removed.** | We will review the appropriateness of the period before the parenthesis, considering the journal guidelines. |
| **Lines 448-450: "For SSP5-RCP8.5 and SSP3-RCP7.0, global and regional trends disagree regarding the direction of change in grassland area, which leads to differences in forests and natural vegetation." Please explain the potential reasons behind this.** | To approach this comment, we will add the following paragraph in the discussion and conclusions section:

"A possible explanation for this behavior is the expected increase in livestock products in the SSP5-RCP8.5 and SSP3-RCP7.0 scenarios. Higher demand for meat and dairy products leads to a greater need for grasslands and crops used as animal feed. Both models account for the feed mix required to meet animal energy needs, considering factors like production systems types and feed conversion.
However, how these demands and shares of the feed mix are estimated differs between the models, which can lead to varying projections for grassland use. On the one hand, in MAgPIE, grassland intensification and reliance on crop-based feed sources reduce the need for grassland expansion in scenarios with high demand for livestock products. On the other hand, although IMAGE moves to more intensive livestock systems as well, the share of grass in the feed mix stays relatively high—especially in SSP3-RCP6.0—resulting in a grassland expansion. For information on livestock systems modeling in IMAGE, refer to Bouwman et al. (2005); Lassaletta et al. (2019), and for MAgPIE to Weindl et al. (2017a, b)" |
| **Figure B2: Did the study consider changes in pasture and forest yield in addition to crop yield?** | Changes in pasture yields and forest carbon densities under different climate change pathways (RCPs) are included in the inputs of the land-use models. Lines 137-141 will be modified to clarify this.

"Each simulation utilized biophysical data that captured the impacts of the different climate change pathways (RCPs) on cropland and pasture yields, irrigation requirements and blue water availability, and carbon stocks—changes in carbon stock data applied to natural and planted forests. The impact data was derived from internal (IMAGE) or external (MAgPIE) LPJmL computations." |

| | |
|---|---|
| Line 460: "On the one hand, for example, LUMs have been used to conduct studies focused on China, India, or the European Union, which has involved further development and validation of the models' outputs for these countries/regions (Singh et al., 2023; Wang et al., 2023; Veerkamp et al., 2020) on different resolutions." Are these popularly studied regions showing better consistency among LUMs? | This comment made us aware that the original sentence could be misunderstood. It is important to clarify that individual models have been used to conduct the studies focusing on specific regions and have been validated using databases such as FAO, national datasets, or expert knowledge. Consequently, some regions mentioned may have been studied in depth by only one model. We will revise the sentence as follows:

"LUMs have been used to conduct region-specific studies. For instance, MAgPIE has performed assessments focused on China (Wang et al., 2023) and India (Singh et al., 2023), while IMAGE has examined the European Union (Veerkamp et al., 2020). These studies have involved further development and validation of the models' outputs for these specific regions.
It is important to note that China, India, and Europe are among the largest producers of agricultural commodities—often referred to as 'breadbaskets'—and have received considerable attention from the agricultural and food system scientific community. In our study, as shown in Figure B9, the coefficient of variance in these regions, particularly for cropland area, fertilizer use, and irrigation, is relatively low compared to other areas. This remains true even under scenarios such as SSP3-7.0 and SSP5-8.5 toward the end of the century. These findings highlight the importance of expanding research to less-studied regions and land-use variables." |
| Lines 465-467: "Second-generation bioenergy crops (Figures B7, B10-B13) are generally allocated in concentrated and highly fertile areas across all scenarios. These areas primarily include the west coast of Australia, southern Brazil, the Eastern European Plain (especially in SSP1-RCP2.6), Southeast Asia, southern China, and West Africa." Please explain why these differences in bioenergy crop allocation occur. | This question addresses two aspects: (1) the allocation of second-generation bioenergy crops (lines 365–370) and (2) the difference in the peak of bioenergy demand in the SSP1-2.6 scenario (lines 466–468). We will include the following information in subsection 3.2.5 (Second-Generation Bioenergy) and section 4 (Discussion and Conclusions), respectively.

**Allocation of Second-Generation Bioenergy Crops:**
Future demand for bioenergy is expected to be high in the SSP1-2.6 scenario due to its role as a mitigation option -specifically in the second half of the century - as we can see in the analysis of global trends, leading to larger areas allocated for bioenergy crops. Unlike other land-use variables, land-use models (LUMs) do not include initial maps of second-generation bioenergy cropland for the historical period. Thus, differences in allocation arise from the absence of historical data on dedicated cropland locations and the distinct allocation rules of each LUM. Both models allocate bioenergy crops based on biophysical suitability. However, in MAgPIE, bioenergy crops must compete with other land uses and crop types, and since REMIND determines regional demand and trade flows, each region must fulfill its requirements in the land-use model. In contrast, in IMAGE, they are confined to abandoned agricultural lands or, when insufficient, to natural grasslands.

**Peak Differences in Bioenergy Crop Area:**
Previous studies, such as in Popp et al. (2014), suggest that differences among models regarding bioenergy prices, energy deployment levels and make-ups, yields, assumptions about economic and technological growth, biomass resources, and sensitivities to other variables can explain this behavior. |

| | As noted earlier, additional information is needed to understand better the effects of different parameterizations and factors on the models' outputs. Thus, to address this comment, we expand the discussion by building on insights from previous studies. The following explanation will be incorporated into the discussion, together with the sentence in lines 319-321: |
|---|---|
| **Lines 319-321: "More specifically, MAgPIE's cropland allocation is based on minimizing production costs and local biophysical constraints, while IMAGE's approach relies on a constant elasticity of transformation function, which associates land supply responsiveness with changes in yields and prices (Schmitz et al., 2014)." Could the author elaborate on how these model differences contribute to variations in the LUC results?** | "While both MAgPIE and IMAGE simulate the land-use system by accounting for future socioeconomic, biogeochemical, and biogeophysical changes, they differ in their setups. These differences may partly explain discrepancies in global projections for cropland and grassland areas under some scenarios, as well as the significant influence of the LUM factor on variance for certain variables at spatially explicit levels. A key distinction lies in the economic modeling approach. MAgPIE is a partial equilibrium model focused on the agricultural sector, whereas the IMAGE framework uses the CGE model MAGNET, which accounts for the entire economy. Additionally, MAgPIE's cropland allocation is based on minimizing production costs and local biophysical constraints, while IMAGE's approach relies on a constant elasticity of transformation function, which associates land supply responsiveness with changes in yields and prices (Schmitz et al., 2014). Previous studies, e.g., Alexander et al. (2015), have shown that CGE models often project lower cropland areas. This outcome is likely due to factors such as input substitutability, interactions between agriculture and other economic sectors, and their resulting effects on prices, demand, and supply of agricultural commodities and inputs. Another major difference involves the use of the LPJmL model. MAgPIE employs LPJmL outputs as exogenous inputs, while IMAGE integrates LPJmL dynamically. As Doelman et al. (2022) highlighted, the dynamic coupling of crop, hydrological, and vegetation models can influence estimates, leading to variations in projected biophysical conditions on the spatially explicit level under similar scenarios. Finally, the approach to technological change (TC) is another critical factor. TC directly impacts yields for cropland and grassland, which, in turn, affects land demand and competition, contributing to variations in land-use projections. For further details on the key processes modeled in IMAGE and MAgPIE, please refer to Table A1." |
| **Region Division (Figure B1): The globe is divided into five regions in the manuscript (Figure B1). Please explain the criteria for this division.** | Since MAgPIE and IMAGE perform simulations using different regions, we selected five mega-regions to illustrate regional trends. Specifically, we used the so-called SSP regions, which have been widely applied in studies involving the Shared Socioeconomic Pathways (SSPs) and climate change, e.g., in Popp et al. (2017); Bauer et al. (2017); Meinshausen et al. (2020) and Fu et al. (2021). This clarification and the references will be added to the Methods section (subsection 2.4.1) and summarized in the caption of Figure B1. |
| **Figure B2 Placement: Given the importance of this modeling protocol, I suggest moving Figure B2 into the main text.** | Figure B2 will be moved to the Methods section, specifically to Section 2.1 (Land-Use Models) where the land-use models and their inputs are described. |

---

## Author Response (AR1)

**Potsdam Institute for Climate Impact Research**
**Research Domain: Climate Resilience**

Edna J. Molina Bacca
Telegrafenberg A6
PO Box 60 12 03
D-14412 Potsdam, Germany
Tel.: +49 162-513-2396
E-mail: mbacca@pik-potsdam.de
Potsdam, January 29th, 2025

Prof. Dr. Min Chen
Earth System Dynamics
Copernicus Publications
Bahnhofsallee 1e
37081 Göttingen
Germany

Dear Prof. Dr. Min Chen,

We sincerely appreciate your and the reviewers' insightful comments and your continued consideration of our manuscript. We have carefully addressed the concerns raised and incorporated the feedback into the revised manuscript.

Responding to RC1's comments allowed us to reflect on critical points, add relevant details, and clarify our results better. These improvements will help the audience, including users of the land use and management patterns, better understand and apply our data and findings. RC1's observations were particularly valuable in identifying key gaps, such as the need for more detailed discussions on the downscaling process of the LUMs and the rationale behind observed differences in trends and patterns.

Similarly, RC2's precise and insightful feedback highlighted areas for improvement, including the interplay between policy targets for bioenergy in the scenarios presented and land-use modeling. In the discussion section, we elaborate on these aspects, clarify concepts for non-scientific audiences such as policymakers, and provide a more concise and concrete outlook of the key findings.

These revisions have significantly strengthened the manuscript, and we hope it now meets the standards for publication in *Earth System Dynamics*.

Enclosed are the following additional documents:

1. A point-by-point response to the reviewers' comments, with our responses highlighted in blue (found below).
2. The revised manuscript with all modifications and additions highlighted in yellow.
3. A clean version of the manuscript without highlights.

Thank you once again for your time and consideration.

Sincerely,

**Edna J. Molina Bacca et al.**

**Response to reviewers**

**Reviewer # 1**

**[General comment]** This study utilizes two models (IMAGE and MAgPIE) to project future land use and management under three socioeconomic-climate scenarios. It compares harmonized land-use and management trends, and analyzes uncertainties arising from socioeconomic-climate scenarios, land-use models, climate models, harmonization processes, and the interactions between these factors. The findings indicate that uncertainty in land-use variables increases with higher spatial resolution, while the choice of climate models has minimal impacts on projection variance across scales. The study highlights the need for more intercomparison exercises focused on spatially explicit projections to improve understanding of the complex interactions between human activities, climate, socioeconomic dynamics, land responses, and associated uncertainties at high-resolution levels as models continue to evolve.However, there are some explanations that need further elaboration in the manuscript.

**[Response]** We thank RC1 for reviewing our manuscript and providing thoughtful and constructive feedback. Your suggestions have been invaluable in enhancing the quality and clarity of our work.

**Major Comments**

**[Comment #1]** Clarification of Figures 7 and 8: Could the author clarify why the gridded differences are primarily influenced by the specific factors highlighted in Figures 7 and 8, rather than other potential variables? On Lines 438–441, the manuscript states: "One of the primary explanations for the effect of harmonization on forests is the different inputs regarding forests among the LUMs and LUH2 historical maps used in harmonization, especially in areas with intermediate tree cover. For example, global forest areas in 2000 range among different satellite sources and FAO between 3600 and 4300 million hectares (Ma et al., 2020)." While this provides some explanations for the effect of harmonization on forests, more detailed explanations of the dominance of the factors shown in Figures 7 and 8 would be helpful for better understanding.

**[Response]** We thank RC1 for this comment and acknowledge that more details regarding selecting these factors are necessary. We focused on these three factors for the variance analysis for specific reasons. First, the ISIMIP3b experimental design includes three scenarios and, for climate impact uncertainty analysis, five GCMs. Given that two land-use models (LUMs) reported projections, this provides three categorical variables (LUMs, scenarios, and GCMs) and an error term.

Second, the main objective of this variance analysis is to evaluate the level of agreement among LUM patterns and the effect of harmonization based on available data rather than to conduct an in-depth exploration of uncertainty sources among LUMs. If there is agreement among models, scenario assumptions should dominate the variance in

results. Conversely, for example, if LUMs have a stronger influence, this would indicate disagreements in translating narratives, inputs, or internal dynamics among the LUMs.

Incorporating additional variables from the LUMs would require further tests, inputs, and simulations, which are beyond the scope of this study and have been explored in prior research by other authors.

To address this comment, we added the following paragraph to clarify the aims of the variance analysis in the methods section and the reason why we focus on three variables. Lines **208 and 219 of the marked-up version:**

*"This analysis aims to evaluate the level of agreement among LUMs informing about the primary sources of variation of the land-use and land-use-related projections of ISIMIP3b on different scales. In other words, this analysis is used to identify the locations and variables where variations can be explained by the differences among the scenarios' assumptions rather than by differences among land-use model dynamics, impact data, or their interactions. Given the level of detail reported by the land-use models, we focus on the factors corresponding to the categorical variables available. Our central assumption is that the primary sources of variance in the data stem from (1) distinctions among scenario trajectories, (2) differences in the processes, inputs, and modeling approaches of the various land-use models, and (3) uncertainties in the GCMs used to generate the impact data. Incorporating additional variables would require re-running the models and conducting further tests. However, as the primary aim of this study is to evaluate data presented rather than to, e.g., comprehensively analyze differences among and-use models, such tests fall outside the scope of this work. Schmitz et al. (2014) and Nelson et al. (2014) delve more deeply into differences among land-use models. Table A2 in the Appendix provides additional information on initial inputs and model processes relevant for calculating the land-use types and management variables."*

The interpretation based on the aims of the variance analysis could be identified in paragraphs such as:

**Lines 458-464** (Referring to Figure 7):

*"Scenario differences contribute most significantly to the overall variance in second-generation bioenergy crop projections, both globally and regionally, especially in ASIA and the OECD, around the 2060-2070 period. Afterward, LUMs and/or the Interactions factor have a higher share of explaining the variance than the other factors. The differences among LUM models regarding second-generation bioenergy projections suggest challenges for long-term bioenergy with carbon capture and storage (BECCS) and related mitigation policy on the global and local levels since, under the same scenario, LUMs display different second-generation demand and production sites."*

Or the new paragraph in **lines 484 - 488** (referring to Figure 8):

*"The fact that the Interactions factor is significant compared to the other factors highlights the complexity of the relationships between land-use patterns and the GCMs, LUMs, and scenarios studied. Equation 2 in the methods section simplifies highly complex systems, spanning climate, crop, energy, and land-use models, as the workflow diagram shows (Figure 1). Therefore, a significant contribution from the Interaction factor*

*highlights the varying sensitivities and complexity of modeling different land-use variables and the effect that climate impacts and socioeconomic growth assumptions have on them."*

Additionally, we added a supplementary table in Annex A to further expand on the results in section 3.3.2 regarding differences among land-use models and factors—including inputs, definitions, and processes. This table details the land-use variables explored in this study, clarifying their input sources, definitions, and calculation methods for the two models. This addition provides further transparency and context to our analysis.

[revised manuscript text omitted]

**[Comment #2]** Comparison of ISIMIP3b and LUH2 Datasets: The manuscript compares the ISIMIP3b LUC with CMIP6 LUH2 data in various instances, such as on Lines 287–289: "This drop in demand for second-generation bioenergy crops is related to changes in the mitigation assumptions of SSP1-RCP2.6, which involves updated impacts on yields." This is informative, but could the authors provide a more detailed explanation of the core differences between the ISIMIP3b and LUH2 datasets, and explain how these fundamental differences contribute to the observed discrepancies? This additional context would help the reader better understand the significance of ISIMIP3b LUC and understand why it differs from CMIP6 LUH2.

**[Response]** This comment is very relevant, and responding to it helps us highlight the novelty of our work. To clarify the main differences between the ISIMIP3b and LUH2-CMIP6 data sets, we added the following text in the methods section (subsection 2.5 Land-Use Harmonization 2 (LUH2) - CMIP6 dataset). Starting in **line 255 - 268:**

*LUH2 data used for CMIP6 differs from the ISIMIP3b data in that it does not account for $CO_2$ fertilization. In crop models such as LPJmL, $CO_2$ fertilization has a positive effect (yield growth), leading, e.g., to lower required cropland areas to achieve the same production levels. Additionally, LUH2, used for CMIP6, combines outputs from multiple land-use models for different scenarios, introducing variability in dynamics based on the models used. In contrast, for ISIMIP3b, each land-use model simulated each SSP-RCP combination covered in this study. Another key difference lies in the inputs of the LUH2 harmonization algorithm, as the historical datasets used in ISIMIP3b have been updated compared to those in LUH2 for CMIP6. Additionally, a new representation of protected lands to better match the IAM assumptions was included, affecting patterns related to natural vegetation. There are also notable differences in the versions of the models employed. For MAgPIE, the version used for CMIP6 simulations was 3.0, while ISIMIP3b utilized version 4.4.0. The latter (starting from MAgPIE 4.0) introduces several enhancements, most notably, a food demand model that accounts for detailed dietary composition, food waste, and demographic characteristics. MAgPIE's current version also improves spatially explicit outputs by incorporating the accounting of capital stocks and their depreciation and a more detailed representation of the forestry sector. Similarly, for IMAGE, the version used for ISIMIP3b was 3.3, whereas version 3.0 was used for LUH2. IMAGE 3.3 includes more crop categories and advancements in bioenergy, deforestation, land-based mitigation, and water use modeling.*

**[Comment #3]** Explanation of Equation 2: Please explain how interaction is defined and how the interaction calculation is conducted.

**[Response]** Thanks for bringing this to our attention. To clarify, we added the following text to section 2.4.3 (Variance Analysis). Starting on **Line 231 - 236:**

*"The residual term—SS_Int in Equation 2 —represents the portion of variance the independent variables (GCMs, RCPs, LUMs) cannot explain. This interpretation, where*

*residuals are equivalent to the interactions, is particular to this type of study due to the deterministic nature of our data (the LUM models are deterministic). Since the total (SSt_total,v,t) and factors' variance (SS_LUM,v,t + SS_GCM,v,t + SS_Sce,v,t) can be derived from the data, the factor that reflects the effect of the interactions SS_Int can be calculated as the difference between the total and the factor's variance. This component captures the non-additive or nonlinear contributions to the variance".*

**[Comment #4]** Uncertainty from Land Use Downscaling: The land-use downscaling process could introduce uncertainty into the gridded LUC. I suggest the authors could discuss this uncertainty in the discussion section.

**[Response]** We deeply appreciate that the reviewer pointed out one key process for creating spatially explicit land-use projections that was missing in the methods and discussion. To respond to this concern, we added details regarding what the downscaling looks like for each model and some implications in the discussion section.

In the methods section **between lines 104 and 109:**
*"Regarding disaggregation of land-use patterns to the grid level, IMAGE relies on gridded potential yields from LPJmL, data from the simulation's previous time step, a regional management factor, and an empirical allocation algorithm. The process begins with calculating potential cropland and crop production data in the current time step using the patterns from the previous step. If production is insufficient to meet demand, less productive areas are abandoned, whereas cropland expansion employs the empirical allocation algorithm that evaluates cropland and grassland allocation. More information is available in (Doelman et al., 2018)."*

And **between lines 127-131:**
*"In MAgPIE, land-use disaggregation is based on the patterns of the previous time step, available cropland, and a mapping between the high and low resolutions. At each time step, starting with cropland, changes in land use from the clusters are disaggregated using expansion and reduction weights and information about land availability and suitability. Detailed information can be found in the interpolateAvlCroplandWeighted function from the R library luscale developed by the MAgPIE team (Dietrich et al., 2024)".*

In the discussion section, we also added **lines 601 to 608:**

*"Besides modeling dynamics and assumptions, another source of uncertainty in the high-resolution patterns reflected in the LUM factor of the variance analysis are the different downscaling procedures used by the models. Disaggregation of LUMs outputs to high-resolution levels is critical in determining spatially explicit land-use patterns and could contribute to uncertainty if different algorithms are used. During the harmonization process, the original gridded data reported by the LUMs is aggregated to a 2°×2° resolution and subsequently harmonized and disaggregated to 0.25°×0.25° using the approach described in Hurtt et al. (2020). However, the different algorithms the LUMs*

*use to disaggregate their outputs introduce uncertainty on where the reduction or expansion of cropland, or other land types, occurs, affecting fertilizer and irrigation patterns on the spatially explicit level. "*

**Minor Comments:**

**[Comment #5]** Lines 447-448: "However, we found some differences regarding the regional and local distribution of land-use change, specifically in cropland for the LAM region." Please explain why this difference in cropland occurs.

**[Response]** To tackle this, we expanded this sentence to explain the reason why there was this difference in cropland in LAM. In **lines 512-514:**

*"However, we found some differences regarding the regional and local distribution of land-use change, specifically in cropland for the LAM region, due to a higher demand for bioenergy crops in this area in MAgPIE compared to IMAGE."*

This is complemented by **lines 569 - 571**, where possible differences in bioenergy crop demand are mentioned:

*"Previous studies, as in Popp et al. (2014b), suggest that such differences among models on the global and regional level can be associated with bioenergy prices, energy deployment levels and make-ups, crop yields, assumptions about economic and technological growth, biomass resources, and sensitivities to other variables."*

**[Comment #6]** Lines 69-70: "... which has commonly been used for impact analyses in global and regional studies. (Yu et al., 2019; Qiu et al., 2023; Hoffmann et al., 2023)." Please check if the period before the parentheses need to be removed.

**[Response]** Thank you for pointing this out and giving a look at our work with such a level of detail. We have removed the period before the parenthesis in the citations of Yu et al., Qiu et al., and Hoffman. **(See lines 69-70)**

*"…which has commonly been used for impact analyses in global and regional studies (Yu et al., 2019; Qiu et al., 2023; Hoffmann et al., 2023)"*

**[Comment #7]** Lines 448-450: "For SSP5-RCP8.5 and SSP3-RCP7.0, global and regional trends disagree regarding the direction of change in grassland area, which leads to differences in forests and natural vegetation." Please explain the potential reasons behind this.

**[Response]** We thank RC1 for this comment. We understand that more context was missing to understand these results better. To touch on this, we added the following text to the discussion sections starting in **line 515:**

*"A possible explanation for this behavior is the expected increase in livestock products in the SSP5-RCP8.5 and SSP3-RCP7.0 scenarios. Higher demand for meat and dairy products leads to a greater need for grasslands and crops used as animal feed. Both models account for the feed mix required to meet animal energy needs, considering factors like production systems types and feed conversion. However, how these*

*demands and shares of the feed mix are estimated differs between the models, which can lead to varying projections for grassland use. On the one hand, in MAgPIE, grassland intensification and reliance on crop-based feed sources reduce the need for grassland expansion in scenarios with high demand for livestock products. On the other hand, although IMAGE moves to more intensive livestock systems as well, the share of grass in the feed mix stays relatively high—especially in SSP3-RCP6.0—resulting in a grassland expansion. For information on livestock systems modeling in IMAGE, refer to Bouwman et al. (2005), Lassaletta et al. (2019), and for MAgPIE to Weindl et al. (2017a, b)."*

**[Comment #8]** Figure B2: Did the study consider changes in pasture and forest yield in addition to crop yield?

**[Response]** This is a great question. Yes, the impacts of climate change on pastures and forests are considered. We expanded lines 136 to 137 of the original manuscript to reflect this. In the revised manuscript, in **lines 155 to 157:**

*"Each simulation utilized biophysical data that captured the impacts of the different climate change pathways (RCPs) on cropland and pasture yields, water demand and availability, and carbon stocks—changes in carbon stock data applied to natural vegetation and planted forests. The impact data was derived from internal (IMAGE) or external (MAgPIE) LPJmL computations."*

**[Comment #9]** Line 460: "On the one hand, for example, LUMs have been used to conduct studies focused on China, India, or the European Union, which has involved further development and validation of the models' outputs for these countries/regions (Singh et al., 2023; Wang et al., 2023; Veerkamp et al., 2020) on different resolutions." Are these popularly studied regions showing better consistency among LUMs?

**[Response]:** This comment made us aware that the original sentence could be misunderstood and needed to be modified. It is essential to clarify that individual models have been used to conduct the studies focusing on specific regions and have been validated using databases such as FAO, national datasets, or expert knowledge. Consequently, some regions mentioned may have been studied in depth by only one model. To clarify this, we modified the text as follows (**Lines 554-562**):

*"LUMs have been used to conduct region-specific studies. For instance, MAgPIE has performed assessments focused on China (Wang et al., 2023) and India (Singh et al., 2023), while IMAGE has examined the European Union (Veerkamp et al., 2020). These studies have involved further development and validation of the models' outputs for these regions. It is important to note that China, India, and Europe are among the largest producers of agricultural commodities—often referred to as 'breadbaskets'—and have received considerable attention from the scientific community studying the agricultural and food systems. In our study, as shown in Figures B8, B11, and B12, the coefficient of variance in these regions, particularly for cropland area, fertilizer use, and irrigation, is relatively low compared to other areas. This remains true even under scenarios such as*

*SSP3-7.0 and SSP5-8.5 toward the end of the century. These findings highlight the importance of expanding research to less-studied regions and land-use variables."*

**[Comment #10]** Lines 465-467: "Second-generation bioenergy crops (Figures B7, B10-B13) are generally allocated in concentrated and highly fertile areas across all scenarios. These areas primarily include the west coast of Australia, southern Brazil, the Eastern European Plain (especially in SSP1-RCP2.6), Southeast Asia, southern China, and West Africa." Please explain why these differences in bioenergy crop allocation occur.

**[Response]** To tackle this point, we added the following explanation between **lines 422 and 428** to address this comment.

*Unlike other land-use variables, LUMs do not include initial maps of second-generation bioenergy cropland for the historical period. Thus, differences in allocation among LUMs arise from the absence of historical data on dedicated second-generation bioenergy cropland locations and the distinct allocation rules of each LUM. Both models allocate bioenergy crops based on biophysical suitability. However, in MAgPIE, bioenergy crops must compete with other land uses and crop types. Since REMIND determines regional demand and trade flows, each region must fulfill its requirements in the land-use model. In contrast, in IMAGE, cropland dedicated to bioenergy is confined to abandoned agricultural lands or, when insufficient, to natural grasslands.*

As mentioned in Comment # 5, we also added the following sentence explaining differences in overall bioenergy demand (**lines 569-571**):

*"Previous studies, as in Popp et al. (2014b), suggest that such differences among models on the global and regional level can be associated with bioenergy prices, energy deployment levels and make-ups, crop yields, assumptions about economic and technological growth, biomass resources, and sensitivities to other variables."*

**[Comment #11]** Lines 319-321: "More specifically, MAgPIE's cropland allocation is based on minimizing production costs and local biophysical constraints, while IMAGE's approach relies on a constant elasticity of transformation function, which associates land supply responsiveness with changes in yields and prices (Schmitz et al., 2014)." Could the author elaborate on how these model differences contribute to variations in the LUC results?

**[Response]** To elaborate more on this point, we moved lines 319-321 to the discussion section and expanded on how the model difference can contribute to the results. Between **lines 534 and 550**:

*"While both MAgPIE and IMAGE simulate the land-use system by accounting for future socioeconomic, biogeochemical, and biogeophysical changes, they differ in their setups. These differences may partly explain discrepancies in global projections for cropland and grassland areas under some scenarios, as well as the significant influence of the LUM factor on variance for certain variables at spatially explicit levels. A key distinction lies in the economic modeling approach. MAgPIE is a partial equilibrium model focused on the agricultural sector, whereas the IMAGE framework uses the CGE model MAGNET, which accounts for the entire economy. Additionally, MAgPIE's cropland*

*allocation is based on minimizing production costs and local biophysical constraints, while IMAGE's approach relies on a constant elasticity of transformation function, which associates land supply responsiveness with changes in yields and prices (Schmitz et al., 2014). Previous studies, e.g., Alexander et al. (2015), have shown that CGE models often project lower cropland areas. This outcome is likely due to factors such as input substitutability, interactions between agriculture and other economic sectors, and their effects on prices, demand, and supply of agricultural commodities and inputs. Another major difference involves the use of the LPJmL model. MAgPIE employs LPJmL outputs as exogenous inputs, while IMAGE integrates LPJmL dynamically. As Doelman et al. (2022) highlighted, the dynamic coupling of crop, hydrological, and vegetation models can influence estimates, leading to variations in projected biophysical conditions on the spatially explicit level under similar scenarios. Finally, the approach to technological change (TC) is another critical factor. TC directly impacts yields for cropland and grassland, which, in turn, affects land demand and competition, contributing to variations in land-use projections. For further details on the key processes modeled in IMAGE and MAgPIE, please refer to Tables A1 and A2 in the Appendix."*

**[Comment #12]** Region Division (Figure B1): The globe is divided into five regions in the manuscript (Figure B1). Please explain the criteria for this division.

**[Response]** The following clarification was added between **lines 133 and 137** in the methods section to clarify this regional division:

*"Since MAgPIE and IMAGE perform simulations using different regions, we selected five mega-regions to illustrate regional trends. Specifically, we used the so-called SSP regions, which have been widely applied in studies involving the Shared Socioeconomic Pathways (SSPs) and climate change, e.g., in Popp et al. (2017); Bauer et al. (2017); Meinshausen et al. (2020); Fu et al. (2021) (see Appendix Figure B1 for a map of the regions)".*

**[Comment #13]** Figure B2 Placement: Given the importance of this modeling protocol, I suggest moving Figure B2 into the main text.

**[Response]** This is a great hint. Figure B2 was moved to the Methods section, specifically to subsection 2.2 (Scenarios), referenced in **lines 159-160** and found after **line 162**.

*(See Figure 1 for a graphical depiction of the modeling workflow)*

[Figure]

**Figure 1. Modeling protocol.** The flow diagram depicts the modeling workflow starting with the global climate models, which feed the crop/natural vegetation/hydrology models, which in turn generate the input data used by the land-use models (LUMs) together with the multi-region and sector models data used to build the assumptions and constraints of the different SSPx-RCPy scenarios. Black boxes represent processes (decision of scenarios and post-processing tests and steps), the purple represents the multi-region and multi-sector models, the gray the climate models, the green the crop/natural vegetation/hydrology models, and the light brown the land-use models. The dotted line represents the data transfer among models

**Reviewer # 2**

**[General comment]** The study compares the results of two land use models under different scenarios. The study covers a wide range of sections and provides information that demonstrates the robustness of the application. However, some minor and major improvements can still be made.

**[Response]** We sincerely thank RC2 for reviewing our manuscript and providing thoughtful, constructive, and highly relevant feedback.

**[Comment #1]** Line 46-48. Include more recent studies to support the frameworks and models to project and compare future land-use and land-use-related variables.

**[Response]** Thank you for highlighting that additional papers are needed for more context. We agree that additional papers to provide more resources were missing. Considering this, we added newer intercomparison papers **(now lines 48 to 51)** to tackle this point. Specifically, the sentence now looks like:

*"For this purpose, various frameworks and models have been utilized to project and compare future land use and food system-related variables focusing on crop and livestock production, food prices, use of resources, changes in land-use areas, among others under different scenarios and frameworks (Sörgel et al., 2024; Weindl et al., 2024; Doelman et al., 2022; Rose et al., 2022; Lèclere et al.,2020; Hasegawa et al., 2018; Popp et al., 2017; Nelson et al., 2014; Popp et al., 2014b)"*

**[Comment #2]** Lines 61-75 should be part of the methods section.

We agree. We simplified the text in the introduction and added details about the regions, climate models, and the modeling flow diagram used in the methods section.

The text in the introduction **(lines 63 to 74)** now reads as:

*"Specifically, we compare the land-use and land-use change patterns generated by the Integrated Model to Assess the Global Environment (IMAGE) (Stehfest et al., 2014; Van Vuuren et al.,2021), and the Model of Agricultural Production and its Impact on the Environment (MAgPIE) (Dietrich et al., 2019) under assumptions for three different socioeconomic-climate scenarios and climate impact data generated using five Coupled Model Intercomparison Project Phase 6 (CMIP6)-biased corrected global climate models (GCMs). The global trends of the LUMs projections under the different scenarios were compared to the Land-Use Harmonization 2 (LUH2) dataset (Hurtt et al., 2017) of future land-use projections, which has commonly been used for impact analyses in global and regional studies (Yu et al., 2019; Qiu et al., 2023; Hoffmann et al., 2023). In addition to comparing the projections, we consider the variance of contributing factors to identify differences in the land-use model outputs and the locations where the variation among the projections is driven by factors different from the socioeconomic-climate scenarios assumptions, e.g., where differences among land-use model dynamics, the interaction among factors, or the uncertainty from the climate impact data play a more prominent role in explaining the variance."*

The details in the methods section about the regional mapping used in the analysis are now in **lines 133 and 137:**

*"Considering that MAgPIE and IMAGE perform simulations using different regions, we selected five mega-regions to illustrate regional trends. Specifically, we used the so-called SSP regions, which have been widely applied in studies involving the Shared Socioeconomic Pathways (SSPs) and climate change, e.g., in Popp et al. (2017); Bauer et al. (2017); Meinshausen et al. (2020); Fu et al. (2021) (see Appendix Figure B1 for a map of the regions)".*

And about the details of the climate models and the modeling flow diagram in **lines 157 to 162:**

*"The impact data was generated for five general circulation models (GCMs): GFDL-ESM4 (Dunne et al., 2020), IPSL-CM6A-LR (Boucher et al., 2020), MPI-ESM1-2 (Müller et al., 2018), MRI-ESM2-0 (Yukimoto et al., 2019), and UKESM1-0-LL (Sellar et al., 2019) (See Figure 1 for a graphical depiction of the modeling workflow). These GCMs were selected based on the completeness of their data across all ISIMIP sectors, their performance during the historical period, and their representation of key processes, among other criteria (Lange, 2021)."*

**[Comment #3]** Line 100. Not clear how the demand for bioenergy production aligned with climate policies was determined.

**[Response]** Given the critical importance and interconnectedness of sustainable development pathways (including policies), land use, and bioenergy, we recognize and agree with RC2 that there is a gap in explaining how these factors interact in the modeling framework. To address this, we have added the following paragraphs.

For IMAGE between **lines 99 and 102:**

*"The TIMER energy model defines bioenergy demand based on land supply, biomass productivity, input costs, and learning dynamics, which influence bioenergy prices. Climate policies in the IMAGE framework are designed to meet long-term climate targets by establishing global emission pathways. These pathways determine carbon tax prices and mitigation costs, which, in turn, affect bioenergy prices and demand (as detailed by Doelman et al. (2018)."*

For MAgPIE between **lines 118 and 124:**

*"Specifically, REMIND provides information on GHG pricing and the demand for second-generation bioenergy crops (lignocellulosic feedstocks). REMIND determines this demand by considering the supply, trade, and conversion of biomass feedstocks through the value chain while accounting for the energy sector market conditions and regulatory frameworks in each socioeconomic growth scenario (as detailed by Merfort et al. (2023)). Since these scenarios are aligned with specific climate change pathways, bioenergy demand, for example, is intrinsically linked to the emissions budgets and carbon taxes required to achieve particular warming targets."*

**[Comment #4]** Given that the identification of uncertainties was one of the purposes of the study, authors should discuss more in detail the implications of them when these models are being used, in particular for policy making. More examples as the one presented in lines 454-456 are missing.

**[Response]** We thank RC2 for encouraging us to discuss our results and their implications from a broader point of view, including the policy-making perspective. In response to this, we added the following paragraphs (see also comment #5) to the discussion:

Between **lines 528 and 533:**

*"In general, differences in land-use projections are expected to directly affect the impact models that use this data as input. For instance, grasslands are among the ecosystems with the highest wildfire frequencies (Donovan et al., 2017). Therefore, uncertainty in LUMs×GCMs grassland projections could influence the identification of fire hotspots due to human-induced effects (Thompson and Calkin, 2011). Uncertainty propagation stemming from land-use patterns could also impact, e.g., the calculation of emissions from land-use transitions (Neuendorf et al., 2021), shifts in biomes (Alexander et al., 2017), the assessment of ecosystem services, habitat intactness, and biodiversity (Yang et al., 2024), among others."*

*"The uncertainties observed in land-use variables at different resolutions arise from error propagation throughout the modeling workflow, as well as from scenario narrative modeling approaches and other factors. These uncertainties highlight the need for conscientious use of the reported data, carefully considering its limitations and assumptions. The objective of the data is to provide a global overview of land and agricultural systems and their development under a set of socioeconomic and climate scenarios based on different assumptions."*

*"In policy and management decision-making contexts, however, the data presented here should be seen as an overview of global trends. In other words, it is not intended to replace targeted assessments and actions specific to, e.g., country, local, or regional levels that include contextual knowledge—including input from communities and experts— that should be incorporated during the assessment and planning phases to ensure that proposed actions align with the actual needs of the stakeholders (Neuendorf et al., 2021)."*

**[Comment #5]** Authors provide a rich set of results, however a summary key messages across land uses and global regions are missing, that is, messages that contextualise the value of the findings for decision making based on modelling outputs. This could be done in the abstract or in a conclusions section

**[Response]** We appreciate RC2's comment. As suggested by the reviewer, we added the following texts to the abstract and discussion sections.

In the abstract, **lines 19 to 21:**

Referring to the study's highlights, *"...It also underscores the importance of region-specific strategies to balance agricultural productivity, environmental conservation, and sustainable resource use, emphasizing adaptive capacity building, improved land-use management, and targeted conservation efforts."*

In the discussion, starting in **line 614 and finishing in line 640:**

*"While our study's primary focus is not to provide direct policy recommendations, it could offer some general insights. For example, our study suggests the need to promote sustainable grassland management practices and diversified feed mixes for livestock to balance ecological, environmental, and economic demands, particularly in regions like LAM and MAF, where grasslands are projected to grow, especially under the SSP3-RCP7.0 and SSP5-RCP8.5 scenarios in IMAGE's simulations.*
*Also, building adaptive capacity could be key to addressing uncertainties in land-use changes and management projections. It would need to prioritize region-specific strategies that reconcile agricultural and environmental priorities.*

*Key uncertainty hotspots in our study include the allocation of cropland in East Africa, Central Asia, and Australia; forest areas in the African tropical rainforest (ATR); grasslands in Central and Eastern Europe; and other natural vegetation in Southeast. South America, the Sahel, and the east coast of Australia.*

*Another key point is the decline of forests and other natural vegetation in scenarios such as SSP3-RCP7.0 and SSP5-RCP8.5, especially in LAM, MAF, and ASIA. This emphasizes the urgency of prioritizing conservation efforts, monitoring, and dedicated policies to safeguard biodiversity-rich ecosystems.*

*Likewise, the differences in second-generation bioenergy crop allocation among models call for tailored regional strategies that support sustainable bioenergy expansion while considering local suitability and market demands. Developing a common framework for bioenergy crop allocation scenarios in land-use models (LUMs) could also help reduce uncertainty and suggest better methods for the sustainable allocation of bioenergy crops.*

*Projected increases in fertilizer use, particularly in Asia (notably China and India), Brazil, and the American Corn Belt due to their critical roles in food production, highlight the need for efficient fertilizer management practices. Building regional capacity to balance food security requirements while minimizing environmental impacts is essential. Finally, in the scenario with high agricultural demand (SSP3-RCP7.0), areas around rivers such as the Ganges, Indus, and Huang or Aravand rivers consistently appear as critical locations for irrigated cropland. Strengthening water management systems in highly irrigated regions will ensure sustainable irrigation practices and support long-term agricultural productivity.*

*In policy and management decision-making contexts, however, the data presented here should be seen as an overview of global trends. In other words, it is not intended to replace targeted assessments and actions specific to, e.g., country, local, or regional levels that include contextual requirements and knowledge—including input from communities and experts— that should be incorporated during the assessment and planning phases to ensure that proposed actions align with the actual needs of the stakeholders (Neuendorf et al., 2021)."*